# SATA-Bench: Select All That Apply Benchmark for Multiple Choice Questions

## Abstract

Current large language model (LLM) evaluations primarily focus on single-answer tasks, whereas many real-world applications require identifying multiple correct answers. This capability remains underexplored due to the lack of dedicated evaluation frameworks. We introduce SATA-Bench, a benchmark for evaluating LLMs on Select All That Apply (SATA) questions spanning six domains, including reading comprehension, legal reasoning, and biomedicine. Our evaluation of 32 models demonstrates substantial limitations: the strongest model achieves only 75.3% Jaccard Index and 41.8% exact match accuracy. We identify three systematic biases underlying these failures: *(i) unselection bias:* models systematically avoid certain correct answer choices; *(ii) speculation bias:* models include incorrect answers when uncertain; and *(iii) count bias:* models consistently underpredict the number of correct answers. To address these limitations, we propose **Choice Funnel**, a decoding strategy that combines token debiasing with adaptive thresholding and abstention handling to guide models toward complete and accurate multi-answer selections. Choice funnel improves the accuracy of the exact match by up to 29% while reducing the inference cost by more than 64% compared to the existing approaches. We release SATA-Bench and Choice Funnel to encourage the development of LLMs capable of robust decision-making in realistic multi-answer scenarios.

🐙 **Data & Code:** github.com/sata-bench/sata-bench

🤗 **Data & Dataset Card:** huggingface.co/datasets/sata-bench/sata-bench

## 1 Introduction

Large Language Models (LLMs) have demonstrated remarkable capabilities across diverse natural language processing tasks, with multiple-choice question answering becoming a standard evaluation framework (Pezeshkpour & Hruschka, 2024; Zheng et al., 2024a). However, current benchmarks assume a single correct answer per question, even though many applications require multiple valid responses, and because they rely on binary scoring that does not penalize speculation, they inadvertently encourage hallucination (Kalai et al., 2025). Consider content moderation systems that must flag posts

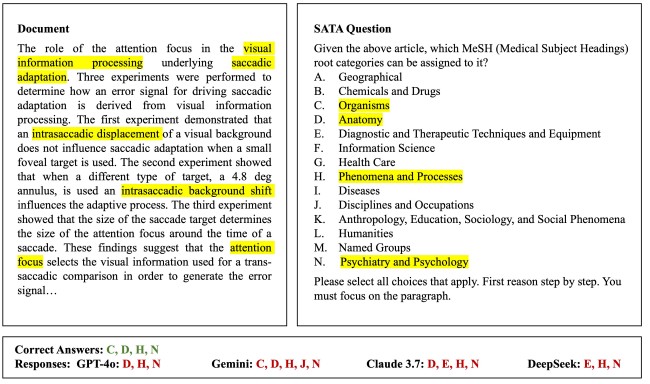

Figure 1: Representative example of an LLM failure on a SATA (Select All That Apply) question. Models often miss valid answers due to unselection, count, and speculation biases. Gemini speculates in this question while GPT-4o underselects. Other models may have unselection bias over C.

for several policy violations simultaneously, medical diagnosis tools that identify co-occurring conditions, or legal research platforms that classify documents under multiple relevant statutes. These scenarios represent Select All That Apply (SATA) tasks, where success depends not on choosing

the single best option but on accurately identifying the complete set of correct answers. Despite their prevalence in real-world applications, SATA tasks remain underexplored in LLM evaluation, leaving a gap between benchmark performance and practical utility with direct implications for trustworthiness and safety. Existing evaluations overestimate model reliability by rewarding speculation, whereas SATA-specific metrics directly penalize speculative behavior.

To address this gap, we introduce SATA-BENCH, a comprehensive benchmark containing over 10,000 human-validated questions across six domains: reading comprehension, toxicity detection, news categorization, biomedicine, legal classification, and event analysis. Unlike existing multi-label classification datasets that often include dozens of possible labels and assume bag-of-words features, SATA-BENCH provides natural-language multiple-choice questions with 3–15 options and 2–10 correct answers, together with metrics that evaluate option-order effects, abstention behavior, and other phenomena unique to LLMs.

Our evaluation of 32 state-of-the-art models (including both proprietary LLMs and open-source alternatives) reveals substantial limitations in multi-answer reasoning. Even the best-performing model achieves only 41.8% exact match accuracy, missing the full correct set in nearly 60% of questions. Figure 1 illustrates a representative failure where models correctly identify some valid answers but systematically avoid others. We identify three systematic biases[1] underlying these failures: *unselection bias*, where models consistently avoid certain answer positions regardless of content; *count bias*, where models underestimate the total number of correct answers; and *speculation bias*, where models include incorrect options when uncertain rather than abstaining (Kalai et al., 2025). To mitigate these issues, we propose *Choice Funnel*, a decoding algorithm that combines token debiasing, adaptive thresholding, and abstention handling. Beyond evaluation, SATA-BENCH serves as both a benchmark and a diagnostic platform, revealing systematic failure modes and enabling algorithmic advances such as Choice Funnel.

**Our Contributions.** The primary contributions of this paper are:

1. SATA-BENCH *Data Curation*: We curate a high-quality, diverse benchmark dataset explicitly designed to challenge LLMs on multi-answer tasks. SATA-BENCH contains more than 10K human-validated questions with multiple domains, varying difficulty levels, multiple correct answers, and carefully constructed distractors. In addition, we provide readability, confusion, and similarity analyses to ensure clarity, diversity, and task complexity across six domains.

2. *Comprehensive Evaluation*: We conduct the largest-to-date evaluation of 32 proprietary and open-source LLMs on SATA questions, revealing that even the strongest models achieve only 41.8% exact match accuracy and 75.3% Jaccard Index.

3. *Bias Diagnosis*: We identify and formalize *unselection*, *count*, and *speculation* biases as obstacles to solving SATA questions, and introduce multiple metrics to evaluate these biases.

4. *Choice Funnel Algorithm*: We introduce a decoding strategy that jointly mitigates these biases through token debiasing, adaptive thresholding, and abstention handling, improving exact match accuracy by up to 29 percentage points while reducing inference cost by 64%.

## 2 SATA-BENCH DATA CURATION

Our objective is to develop a dataset that spans diverse tasks and domains while providing sufficient challenge to reveal differences in LLM capabilities. The curation process consists of three stages: (i) selecting source datasets, (ii) transforming them into SATA format, and (iii) filtering questions for readability, diversity, human validation, and clarity (see Figure 3). We curated SATA-BENCH to include tasks in *Reading Comprehension* (Khashabi et al., 2018), *Text Classification* (`News` (Padmanabhan et al., 2016), `Events` (Event-Classification)), and *Domain Understanding* (`Toxicity` (Gehman et al., 2020), `Biomedicine` (PubMed-MeSH, 2021), `Laws` (Chalkidis et al., 2019)). Detailed dataset descriptions are provided in Appendix A.

### 2.1 SATA TRANSFORMATION

We convert each item to a SATA item by first gathering the text, gold labels, and option count. We then enforce an option-to-answer ratio of 2–3 to maintain consistency and difficulty (Thompson &

---

[1]We use the term bias to highlight systematic tendencies in prediction (See Appendix Q.2 for mathematical definitions), not socioeconomic or demographic bias

Table 1: Compared to prior benchmarks (Kalai et al., 2025), SATA-BENCH penalizes speculation, spans multiple domains, uses non-binary metrics, and includes multi-stage human annotations. Penalizing speculation means wrong answers receive lower scores than abstaining. Jaccard Index penalizes speculation: if ground truth is $A, B$ and model predicts $B, C$, $JI(JacardIndex) = 0.33$. if it does not speculate and predicts $B$, $JI = 0.5$. Thus, this scoring scheme gives a lower score to LLMs that speculate when uncertain.

| Benchmark | Scoring method | Binary grading | Penalizing speculation | Human labeling | # Domains |
|---|---|---|---|---|---|
| GPQA | Multiple-choice accuracy | Yes | None | Yes | 3 |
| MMLU-Pro | Multiple-choice accuracy | Yes | None | Yes | 57 |
| IFEval | Programmatic instruction verification | Yes[a] | None | No | 1 |
| Omni-MATH | Equivalence grading[*] | Yes | None | Yes | 1 |
| WildBench | LM-graded rubric[*] | No | Partial[c] | Partial | Varied |
| BBH | Multiple-choice / Exact Match | Yes | None | Yes | 23 |
| MATH | Equivalence grading[*] | Yes | None | Yes | 1 |
| MuSR | Multiple-choice accuracy | Yes | None | Yes | 1 |
| SWE-bench | Patch passes unit tests | Yes | None | No | 1 |
| HLE | Multiple-choice / equivalence grading[*] | Yes | None | Yes | 10+ |
| SATA-BENCH | Jaccard Index / Exact Match | Partial[b] | Yes | Yes | 6 |

[*] Grading is performed using language models, hence incorrect *bluffs* may occasionally be scored as correct.
[a] IFEval aggregates several binary rubric sub-scores into a composite score.
[b] Jaccard Index and Precision are not binary grading.
[c] Grading rubric (1-10 scale) may award hallucinated responses.

Giffin, 2021). Next, we set $k$ to the number of correct answers $c$, construct the option set with the $c$ gold choices plus $k - c$ distractors sampled from the pool, and finally shuffle the options to mitigate position and label bias.

## 2.2 QUESTION FILTERING

From the original SATA questions (characteristics shown in Table 6 in Appendix), we filter them using the following steps (see Figure 3):

**Initial Filtering.** To clean the original source data, we eliminated questions with fewer than ten words (Sanderson, 2010; Karunarathna et al., 2024). To ensure each question is understandable and solvable, we excluded those containing ambiguous, vague, or subjective terms (Moore et al., 2024). We also removed contaminated questions to reduce memorization risk, following (Li et al., 2024) (details in Appendix B.1).

**Readability.** To ensure SATA-BENCH questions are both understandable and challenging, we assessed readability using the Flesch Reading Ease (FRE) score (Flesch, 1948) and the Gunning Fog Index (GFI) (Gunning, 1952). We retained questions with an FRE score between 20–100 and a GFI score between 6–17, corresponding to $6^{th}$-grade through graduate-level difficulty (Kincaid et al., 1975; Gunning, 1952). This step removed unclear or trivial questions while preserving a broad difficulty range.[2]

**Question Similarity.** To avoid redundancy, we measured cosine similarity between TF-IDF representations (Sparck Jones, 1972) of all question pairs, following Zhu et al. (2021). Cosine similarity between each correct option and each distractor option was calculated, producing an n × m similarity matrix. The confusion score for the question is defined as the average cosine similarity across all n × m pairs. We removed questions with at least 80% similarity. We also performed statistical analysis (Appendix B) to confirm the consistency of our label design.

**Confusion Score.** SATA difficulty is closely tied to the similarity between correct answers and distractors. We quantified this by computing semantic similarity using ST5-XXL (Ni et al., 2021), which performed best in Muennighoff et al. (2022a). To balance difficulty, we binned questions into

---

[2]We additionally computed four other readability measures—Flesch-Kincaid Grade Level (FGL) (Kincaid et al., 1975), Automated Readability Index (ARI) (Kincaid et al., 1975), and Dale–Chall Readability (DCR) (Dale & Chall, 1948)—which are included in the released dataset.

10 groups by confusion score and sampled 50–300 records from each bin, ensuring SATA-BENCH covers a wide difficulty spectrum. Figures 5 and 6 show the distribution of confusion scores before and after filtering, as well as breakdowns by source dataset.

**Human Validation.** Human evaluation proceeded in two stages. First, annotators identified and removed questions containing ambiguous content (Appendix B.2) from the pre-filtering dataset, producing 9.5K pre-annotation questions, that could be used for finetuning (Appendix T). In the second stage, three annotators reviewed all sampled questions to correct labeling errors. Questions without unanimous agreement were excluded (Annotator information is mentioned in Appendix B.4). As a result, the final release includes a 1.47K evaluation set (see overall statistics in Table 5). We also validate that correct answers are consistently validated over 95% of the time.(Appendix B.5)

## 2.3 SATA-BENCH CHARACTERISTICS

SATA-BENCH has the following characteristics: (i) *granular grading:* Multiple correct answers provide a finer understanding than binary true/false; (ii) *diversity:* the dataset spans both knowledge-based and reasoning-driven tasks; (iii) *human validation:* all items are manually reviewed for clarity and correctness, and readability scores ensure coverage from $6^{th}$ grade through graduate level, with ambiguous or trivial questions removed; (iv) *challenging:* 76% of questions fall within the standard FRE range (60–70), the average GFI corresponds to $13^{th}$ grade (first-year college), and correct answers and distractors have a mean semantic similarity of 0.24 (skewness = 1.8), clustering around 0.22 with a long tail of harder items (Figure 2).

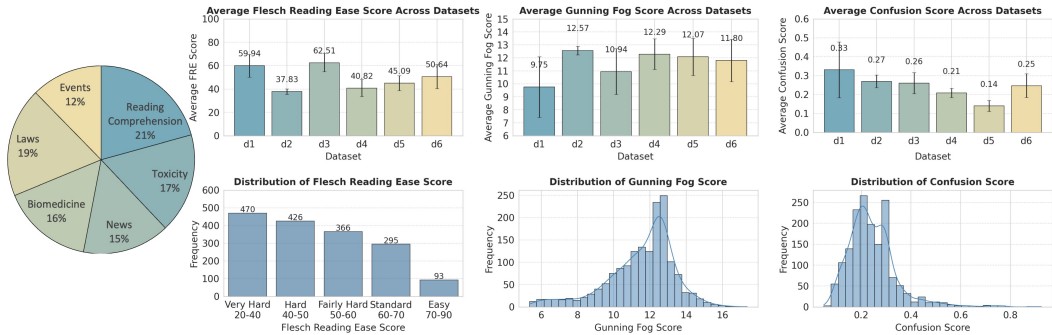

Figure 2: SATA-BENCH Evaluation Dataset Overview. SATA-BENCH covers a diverse set of topics and achieves a balance between readability and difficulty (measured by confusion score). d1: Reading Comprehension, d2: Toxicity, d3: News, d4: Biomedicine, d5: Laws, and d6: Events.

## 3 EXPERIMENTS

This section presents the experiments conducted to assess the capabilities of LLMs on SATA questions on evaluation set. Our benchmark covers 18 proprietary and 14 open-source models (see Table 8 for details). Because the benchmark spans diverse domains, we adopt a zero-shot evaluation protocol. The system prompt specifies that each question has at least two correct answers, and we instruct the LLM to output the results in JSONL format (Intelligence, 2024; Zhou et al., 2023). We benchmark different system prompt strategies in the Appendix N. Furthermore, we employ a CoT prompting strategy following OpenAI & el at (2024). We then extract answers from the JSONL output using exact and fuzzy match. For cases where JSONL extraction fails (fewer than 3%), we use Claude 3 Haiku and human labelers to recover the correct options. However, for smaller models, the JSONL extraction fails in more than 5% of the cases, making this method less reliable. In these cases, following Hendrycks et al. (2021), we omit CoT and instead rank options using the probability of the first output token. To calibrate thresholds, we hold out 100 randomly sampled instances from the benchmark and tune each model for the optimal Jaccard Index (Bogatinovski et al., 2022). We then select all options with probabilities above this threshold. This probability-based method applies only to models with accessible token likelihoods. Finally, we also evaluate the performance of non-expert humans on the benchmark (Appendix E).

Evaluation of SATA question responses requires metrics that capture partial correctness , penalize inappropriate selections, and identify bias. We organize our evaluation into three categories:

Table 2: Performance comparison of 32 different LLMs across various metrics on SATA-BENCH. We highlight the best (**bold**) and second-best (underline) values. Columns labeled [(↑)] indicate higher-is-better; columns labeled [(↓)] indicate lower-is-better. Models with explicit reasoning capabilities are highlighted in *italic*. All numeric values are rounded to two decimal places. We retrieve exact labels for models evaluated using Inference-Based Retrieval + CoT prompting. For models evaluated under Probability-Based Retrieval, we select labels based on token probability thresholds.

| Model Name | Performance | | | Selection Bias | | | Count Bias | |
|---|---|---|---|---|---|---|---|---|
| | JI↑ | FPR↓ | EM↑ | SPD↓ | RStd↓ | RSD↓ | CtDif | CtAcc↑ |
| **Inference Based Retrieval + CoT** | | | | | | | | |
| *O3* | 73.91 | 31.58 | **41.77** | 0.38 | 6.79 | 0.06 | -0.39 | 46.12 |
| GPT 4.1 | **75.23** | 40.37 | 40.49 | 0.13 | 5.98 | 0.06 | -0.04 | 45.52 |
| *GPT-OSS 120B* | 74.28 | 37.53 | 40.29 | 0.19 | 6.31 | 0.07 | -0.16 | 47.57 |
| *Grok 3 Think* | 74.40 | 43.10 | 39.71 | 0.30 | 6.26 | 0.07 | 0.06 | 44.24 |
| GPT 4 | 74.11 | 38.42 | 39.47 | 0.21 | 6.63 | 0.06 | -0.20 | 46.61 |
| *Claude 3.7 Think* | 70.96 | 35.16 | 37.92 | 0.46 | 18.77 | 0.34 | -0.32 | 44.48 |
| Claude 3.7 | 70.98 | 33.10 | 37.82 | 0.49 | 6.59 | 0.25 | -0.43 | 43.58 |
| Claude 3 Sonnet | 70.72 | 38.81 | 36.49 | 0.36 | 7.37 | 0.07 | -0.35 | **48.00** |
| *Geimini 2.5 Think* | 72.58 | 42.16 | 36.46 | **0.12** | **4.76** | 0.06 | -0.01 | 43.76 |
| Claude 3.5 Haiku | 71.12 | 50.01 | 35.89 | 0.33 | 7.31 | 0.35 | 0.18 | 42.61 |
| Claude 3 Haiku | 70.63 | 40.84 | 35.64 | 0.42 | 6.24 | 0.07 | -0.22 | 47.15 |
| Claude 3 Opus | 70.15 | 34.17 | 35.59 | 0.62 | 8.26 | 0.07 | -0.52 | 44.36 |
| Gemini 2 Flash | 70.71 | 40.79 | 34.60 | 0.17 | 6.14 | 0.06 | -0.23 | 39.94 |
| GPT 4.1 mini | 69.90 | 37.31 | 33.46 | 0.30 | 6.69 | 0.06 | -0.39 | 38.61 |
| Nova Pro | 68.92 | 31.64 | 32.95 | 0.52 | 7.92 | 0.07 | -0.55 | 39.27 |
| Claude 3.5 Sonnet | 67.15 | 34.25 | 32.22 | 0.43 | 8.41 | 0.09 | -0.46 | 38.55 |
| Llama 3.1 405B | 67.18 | 35.06 | 30.17 | 0.33 | 6.90 | 0.45 | -0.39 | 36.30 |
| Nova Lite | 63.75 | 39.88 | 29.11 | 0.52 | 9.12 | 0.45 | -0.51 | 37.39 |
| *Deepseek R1* | 64.49 | 34.89 | 28.17 | 0.94 | 17.44 | **0.03** | -0.57 | 33.52 |
| *GPT-OSS 20B* | 60.73 | 40.90 | 27.35 | 0.77 | 11.05 | 0.10 | -0.53 | 31.80 |
| Mistral Large V2 | 57.16 | 27.23 | 22.83 | 1.33 | 10.89 | 0.12 | -1.10 | 27.27 |
| Qwen Plus | 55.74 | 24.03 | 21.12 | 2.24 | 10.72 | 0.11 | -1.18 | 24.85 |
| Nova Micro | 55.77 | 29.28 | 18.37 | 1.84 | 11.10 | 0.27 | -1.09 | 24.30 |
| Llama 3.2 90B | 55.78 | **23.81** | 18.30 | 1.84 | 11.10 | 0.27 | -1.09 | 24.30 |
| Llama 3.1 70B | 55.59 | 23.92 | 17.94 | 1.81 | 10.06 | 0.10 | -1.12 | 22.12 |
| Non-expert Human | 45.02 | – | 17.93 | 1.46 | 15.32 | 1.46 | -0.6 | 34.12 |
| **Probability Based Retrieval** | | | | | | | | |
| Mistral 8B | **46.63** | 32.21 | **14.73** | **11.42** | 19.47 | 1.27 | -1.35 | 21.01 |
| Llama3 8B | 43.64 | 30.06 | 13.82 | 12.09 | 17.85 | 1.09 | -1.59 | **22.00** |
| Bloomz 7B | 41.15 | 57.76 | 11.27 | 20.62 | 29.00 | 1.51 | **-0.87** | 20.09 |
| *DeepSeek R1 Distill 8B* | 40.02 | 45.33 | 8.85 | 13.38 | 21.62 | 1.14 | -1.29 | 20.42 |
| Qwen2.5 14B | 37.58 | **17.27** | 6.30 | 21.01 | 18.02 | **1.06** | -2.24 | 11.93 |
| Phi3 7B | 34.57 | 17.64 | 2.97 | 23.22 | 18.57 | 1.22 | -2.33 | 7.22 |
| *Phi4-mini-reasoning* | 29.69 | 26.73 | 2.12 | 21.62 | **13.90** | 1.59 | -2.37 | 7.35 |

performance metrics that measure correctness and speculation, selection bias metrics that quantify positional preferences, and count bias metrics that assess quantity prediction accuracy. Detailed explanations for all metrics appear in Appendix F.

**Performance and Speculation Bias Metrics.** We employ four metrics to assess answer correctness (Tarekegn et al., 2024a). *Jaccard Index (JI)* measures the intersection-over-union between predicted and gold labels, providing credit for partial matches. A low JI also reflects limited overlap between predicted and gold labels, indicating speculation bias. *False Positive Rate (FPR)* measures the proportion of questions where models select any incorrect option, directly quantifying speculation bias. *Exact Match (EM)* requires the predicted set to exactly match the gold set, representing the most stringent evaluation criterion.

**(Un)selection Bias Metrics.** To characterize positional preferences, we measure models' tendencies to favor or avoid specific option positions. We use *RStd* (Zheng et al., 2024a) and *RSD* (Croce et al., 2020; Reif & Schwartz, 2024) to quantify selection bias toward particular option IDs. Additionally, we introduce *Selection Probability Divergence (SPD)* to measure unselection bias—the systematic tendency to avoid certain options regardless of content (detailed in Appendix G).

**Count Bias Metrics.** Models often select fewer options than warranted, necessitating specialized metrics for quantity assessment. We measure: (i) mean signed difference between selected and correct counts (*CtDif*), where negative values indicate under-selection; and (ii) percentage of cases with exact count matches (*CtAcc*) to assess quantity prediction accuracy.

## 3.1 KEY OBSERVATIONS

**SATA-BENCH is challenging and different.** 13 models achieve a JI above 70%, but none surpass 42% EM. This shows that while models often identify some correct answers, they fail to consistently recover the full set.

Proprietary models generally achieve higher JI and Precision than open-source ones. Unlike other benchmarks, no single model dominates across all metrics. Notably, larger and more recent models do not always perform better. For instance, Claude 3 Sonnet outperforms Claude 3.5 Sonnet and Claude 3 Opus in exact match, though within the Claude family, larger models consistently have higher precision (e.g., Claude 3 Opus has the highest precision among the Claude 3 variants). According to Anthropic (2024); DeepSeek-AI & el at (2024), these results contrast with performance on single-answer Multi-Choice Question (MCQ) benchmarks such as MMLU (Hendrycks et al., 2021) and ARC (Clark et al., 2018), where larger or newer models typically show clear gains. Large reasoning models (LRMs) are slightly better than their non-reasoning counterparts in JI but failed to reduce selection and count bias. We provide a case study in Appendix R to investigate LRM's behavior.

**Models choose too few answers.** Nearly all LLMs tend to select fewer answers than required. For example, Llama 3.1 70B selects, on average, one fewer option per question than the correct number. Accordingly, it achieves the highest precision but the lowest Jaccard Index (JI). The tendency to under-select increases as the number of correct answers grows (Figure 11), which in turn depresses JI for questions with many correct choices (Figure 12). Even the best model achieves a CtAcc of only 48%, predicting the correct number of answers in fewer than half of the questions. We hypothesize that this behavior stems from models being primarily trained and evaluated on benchmarks with single correct answers, making them poorly suited for SATA tasks. A t-test confirms this under-selection: the mean of CtDif is significantly below 0 (one tailed), with $p = 1.70 \times 10^{-6}$, $t = -5.82$, $DoF = 24$. However, CtAcc has improved as with better performed model (See Figure 15).

**Models speculate a lot.** LLMs also over-select, consistently choosing incorrect options, with all models exceeding a 20% FPR. More than 70% of the models predict at least one incorrect choice more often than they produce exact matches, underscoring their speculating behavior. Interestingly, stronger-performing models tend to speculate more: FPR and EM are positively correlated ($r = 0.61$, $p = 8 \times 10^{-4}$, $DoF = 23$.) (Figure 14) This dual trend suggests that as models improve in identifying correct answers, they also become more prone to speculation, highlighting the difficulty of disentangling genuine knowledge from overconfidence in LLM predictions.

Table 3: Average performance of three models. The first column shows row numbers for reference.

| | Experiment | EM | Precision | RStd | CtDif |
|---|---|---|---|---|---|
| 1 | 1/2/3/4 | 35.50 | 82.99 | 10.22 | -0.37 |
| 2 | a/b/c/d | 30.69 | 83.10 | 11.56 | -0.26 |
| 3 | default | 33.00 | 84.62 | 7.37 | -0.25 |
| 4 | few shots | 28.35 | 76.61 | 17.33 | -0.42 |
| 5 | option by option | 30.50 | 86.28 | 4.81 | -0.64 |
| 6 | option few shots | 30.87 | 85.80 | 7.93 | -0.48 |
| 7 | with avg count | 27.33 | 76.17 | 14.90 | -0.40 |
| 8 | with count number | 53.95 | 83.30 | 3.45 | -0.08 |
| 9 | single choice | 45.53 | NA | NA | NA |

**Unselection bias exists.** Some models exhibit a systematic tendency to avoid selecting certain labels, even when they are correct. When comparing Selection Probability Divergence (SPD) from our benchmark with 1,000 randomly simulated SPDs, Welch's t-test shows that LLMs' SPD is significantly higher than random ($p = 0.0467$, $DoF = 23$, $t = 1.75$). Geimini 2.5 is the best

performed model in (un)selection bias. However, it still underperforms on label M, with its recall rate 6.3% lower than its overall average recall (Figure 10).

## 3.2 ABLATION STUDIES

We conducted ablation studies to test different strategies for improving model performance. We report the average results across three models (Llama 3.1 405B, Nova Pro, Claude 3.5 Haiku) selected for diverse profiles in terms of cost, open-source availability, and overall performance. The complete prompts are provided in Appendix H.3.

We tested multiple strategies to improve performance, but none produced consistent or significant gains, suggesting that prompting alone is insufficient for enhancing SATA performance.

- **Changing option symbols.** Replacing the default option IDs (A/B/C/D) with a/b/c/d or 1/2/3/4 did not reduce selection bias. While the numeric format slightly improved exact match, it also increased selection bias and reduced precision. Overall, it's ineffective.(rows 1–3, Table 3).
- **Few-shot prompting.** Providing few-shot examples before test questions produced no meaningful improvements (row 4, Table 3).
- **Option-by-option prompting.** Inspired by survey methodology (Smyth et al., 2006; Pew Research Center, 2019), we instructed models to evaluate each option individually. However, models still under-selected and showed no overall improvement (rows 5–6, Table 3).

With additional information, two strategies improved performance and provided insight into why models struggle:

- **Providing the number of correct answers.** To assess how much error stems from uncertainty about the number of valid options, we explicitly told models how many correct answers each question contained. This increased exact match by 20.95 points and reduced selection bias (RStd). However, giving only the average number of correct answers across the dataset reduced performance (rows 7–8, Table 3).
- **Decomposing into single-choice tasks.** For a question with three correct and six incorrect options, we converted it into three separate single-choice questions (one correct + six incorrect each). We redefined exact match as the proportion of original questions where all expanded items were answered correctly. This raised performance by **12.53%** (row 9, Table 3), showing that SATA questions are much harder for LLMs than single-choice ones.

Together, these results suggest that while models can often identify individual correct answers, their lack of awareness of how many answers to select is a key failure mode, highlighting the need for specialized decoding strategies.

## 4 IMPROVING PERFORMANCE ON SATA QUESTIONS

This section focuses on improving performance in open-source models, which expose token-level logits that proprietary models do not. Section 3 shows that straightforward probability-based methods for leveraging token logits perform poorly in SATA: thresholds are hard to calibrate and lead to low exact-match accuracy, they do not correct positional bias (SPD up to 23.22 in Table 2), and they cannot abstain under uncertainty, causing high false-positive rates. These failures are closely tied to three systematic biases— speculation, unselection, and count bias—which we show further degrade LLM performance on SATA-BENCH.

To address **unselection bias**, we can draw from prior research on token debiasing methods (Choi et al., 2024; Zheng et al., 2024a) in the MCQ setting, where selection bias is attributed to the *a priori* probability mass assigned by the model to specific option IDs. These methods propose various techniques to capture and remove such biases. We hypothesize that these techniques can be adapted to mitigate unselection bias in SATA tasks. To address **speculation bias**, we want to design a mechanism to encourage LLMs to abstain rather than speculate under uncertainty. To address **count bias**, we can consider retrieving the predicted probabilities of option IDs and select options whose probabilities exceed a predefined threshold. However, because SATA-BENCH includes a large option set, the probability distribution decays rapidly, with most options receiving near-zero probability mass beyond the first few choices. This makes it challenging to establish a reliable threshold. Converting SATA questions into multiple binary classification problems helps but significantly increases inference cost.

These observations also explain why naive extensions of traditional greedy selection or fixed probability thresholding, which are common in selective prediction and multi-label output, perform

poorly in SATA. Such methods do not explicitly correct positional bias, lack a principled way to set model-specific thresholds, and offer no mechanism to abstain when the model is uncertain. As a result, they exhibit low exact-match accuracy and high false-positive rates in our experiments (see Table 3).

**Choice Funnel Algorithm.** With the above consideration, we propose a decoding method called *Choice Funnel* (Algorithm 1) tailored to solve SATA problems. This approach first adds an auxiliary option "None of the Above," then selects the option with the highest *debiased token probability* and removes it from the option set. The process repeats iteratively until one of two stopping conditions is met: (i) the model selects "None of the Above" or (ii) the probability of the next option falls below a predefined confidence threshold.

The addition of an auxiliary option is inspired by recent research that LLMs exhibit biases similar to those observed in human responses (Choi et al., 2024; Eckman et al., 2024a), aiming to reduce LLM speculation. While "I don't know" (*idk*) being the most common option used to improve survey data quality (Schuman & Presser, 1996) and have been suggested in recent LLM research (Kalai et al., 2025), *NOTA* consistently outperforms *idk* (see ablation study in Appendix M.1).

The intuition behind the second stopping condition comes from our finding that output probabilities correlate with the number of correct options the model considers: the highest token probability tends to be lower at the beginning of iterations, when the model treats multiple options as equally plausible. Later in the process, relatively higher probability is assigned to the final remaining correct option in the set. We also show that Choice Funnel achieves the best performance when both stopping conditions are used together (see ablation study in Appendix M.3).

Regarding the choice of *DebiasingFunction* in Algorithm 1, Choice Funnel is flexible and can incorporate any token debiasing method proven effective in MCQ settings. We demonstrate one such method in Section 4. See ablation study on each sub-component in Appendix M.2. Finally, the inference cost of Choice Funnel, measured by the number of model forward passes, scales linearly with the number of *correct* labels rather than the number of *total* labels. *This makes the method especially efficient when correct labels constitute only a small fraction of the option set.*

---

**Algorithm 1:** Choice Funnel

**Input** : LLM $\pi_\theta$, SATA problem $\mathcal{T}$, option set $\mathcal{O}$, $NOTA$ stop option, $\tau$ confidence threshold

# Initialize the selected option set
$\mathcal{R} \leftarrow \emptyset$
**while** $\mathcal{O} \neq \emptyset$ **do**
  # Generate prompt with available options
  $\mathbf{P} \leftarrow \text{MakeSATAPrompt}(\mathcal{T}, \mathcal{O})$
  # Get first token probability distribution and apply token debiasing
  $p \leftarrow \text{DebiasingFunction}(\pi_\theta(\cdot | \mathbf{P}))$
  # Select option with highest probability
  $o \leftarrow \arg\max_{o \in \mathcal{O}} p(o)$
  # 1. stop when "None of the above" is selected
  **if** $o = NOTA$ **then**
    | **break**
  **end**
  $\mathcal{R} \leftarrow \mathcal{R} \cup \{o\}$
  # 2. stop when the confidence threshold is reached
  **if** $p(o) > \tau$ **then**
    | **break**
  **end**
  **if** $length(\mathcal{R}) = 1$ **then**
    | $\mathcal{O} \leftarrow \mathcal{O} \cup \{\text{NOTA}\}$
  **end**
  $\mathcal{O} \leftarrow \mathcal{O} \setminus \{o\}$
**end**
**Output** : $\mathcal{R}$

---

**Experimental Setup.** In our experiments we adapted the PriDe algorithm (Zheng et al., 2024a) as *DebiasingFunction* in Algorithm 1 due to its label-free design and computational efficiency. It works by first estimating the model's prior bias toward specific option ID tokens (e.g., A, B, C) through random permutations of option contents in a small subset of test samples (10% of the data in our experiments). We then use this estimated prior to adjust the prediction distribution on the remaining samples, thereby separating the model's inherent positional and token biases from its task-specific predictions. Because the original PriDe algorithm was designed for standard single-answer MCQ tasks, we modified it to better fit the SATA setting (see Appendix K).

We evaluate the performance of Choice Funnel against **three baseline methods** that rely on first-token probabilities: (i) using the first-token probability with a fixed threshold, as defined in Section 3 (referred to as *first token*); (ii) applying PriDe debiasing on top of the first-token method (Zheng et al., 2024a), current best-performed method in terms of speed and accuracy in solving MCQs. (referred to as *first token debiasing*); and (iii) converting each option into an individual binary yes/no question (referred to as *yes/no*). Other advanced calibration methods cannot generalize to SATA or require an

extensive dataset to fine-tune the model. In this study, we use standardized prompts (Appendix H) and experiment with seven LLMs from Table 20 that fall under the Probability-Based Retrieval category (details in Appendix L). For each model, we compute the metrics reported in Table 20 and additionally report an *InfCost* metric to capture the number of model forward passes required for each method.

**Key Observations.** Choice Funnel consistently outperforms all three baselines across all seven models in EM, SPD, and CtAcc (Table 4). *Choice Funnel reduces unselection bias, speculation bias, and count bias*—compared to the *first token* baseline, it achieves an average 56.2% reduction in SPD, 36.4% improvement in JI, and 154.6% improvement in CtAcc, and a 277.5% gain in Exact Match (EM) performance. While reasoning models also show improvements with Choice Funnel, we exclude them from aggregate calculations since their exceptionally low baselines would inflate relative gains. Against the strongest baseline, the *yes/no* approach, *Choice Funnel* delivers a 29.9% improvement in EM while reducing model forward passes by 64.5% through its early stopping mechanism, demonstrating scalable inference efficiency. A t-test confirms that Choice Funnel significantly outperforms both *yes/no* and *first token debiasing* on EM and CtAcc, with a maximum p-value of 0.0079 and t statistics of 4.92. Although our models' parameter sizes (7B–14B) limit direct comparison to much larger proprietary systems, Choice Funnel's performance on the *phi3-small* model still surpasses that of larger models such as Llama-90B and Mistral-Large V2 (Table 20), underscoring the effectiveness of our method. Each component of Choice Funnel is essential (Appendix M) and it performs well across larger models (Appendix M.4) and black-box settings (Appendix M.2).

Table 4: Performance of various models on SATA-BENCH using different decoding methods. *Choice Funnel* achieves consistently stronger results, effectively reducing selection and count bias compared to three baseline methods. The best values in each column are shown in **bold**. Columns labeled [↑] indicate higher-is-better, while columns labeled [↓] indicate lower-is-better. All values are rounded to two decimal places.

| Model Name | EM↑ | Recall↑ | JI↑ | SPD↓ | CtAcc↑ | InfCost↓ |
|---|---|---|---|---|---|---|
| Mistral-8B + *first token* | 14.73 | 53.23 | 46.63 | 11.42 | 0.21 | **1650** |
| Mistral-8B + *first token debiasing* | 8.91 | 37.97 | 34.27 | 152.23 | 0.14 | 2534 |
| Mistral-8B + *yes/no* | 16.48 | **55.91** | 48.80 | 12.88 | 0.21 | 15517 |
| Mistral-8B + *choice funnel* | **20.24** | 55.78 | **52.56** | **8.50** | **0.27** | 4803 |
| Phi3-7B + *first token* | 2.97 | 35.67 | 34.57 | 23.22 | 0.07 | **1650** |
| Phi3-7B + *first token debiasing* | 1.76 | 28.24 | 27.47 | 175.24 | 0.05 | 2534 |
| Phi3-7B + *yes/no* | 25.45 | **72.40** | 60.03 | **1.39** | 0.30 | 15517 |
| Phi3-7B + *choice funnel* | **29.27** | 70.24 | **61.85** | 3.47 | **0.38** | 6339 |
| Qwen2.5-14B + *first token* | 6.30 | 38.76 | 37.58 | 21.01 | 0.12 | **1650** |
| Qwen2.5-14B + *first token debiasing* | 4.61 | 31.49 | 30.36 | 154.26 | 0.09 | 2534 |
| Qwen2.5-14B + *yes/no* | 25.64 | 60.56 | 56.18 | **2.76** | 0.31 | 15517 |
| Qwen2.5-14B + *choice funnel* | **27.82** | **67.07** | **61.12** | 3.80 | **0.35** | 6005 |
| Bloomz-7B + *first token* | 11.27 | 50.80 | 41.15 | 20.62 | 0.20 | **1650** |
| Bloomz-7B + *first token debiasing* | 7.09 | 38.41 | 32.05 | 149.17 | 0.15 | 2534 |
| Bloomz-7B + *yes/no* | 11.93 | 42.67 | 29.40 | 17.78 | 0.13 | 15517 |
| Bloomz-7B + *choice funnel* | **20.18** | **54.90** | **46.15** | 9.82 | **0.32** | 5440 |
| Llama3-8B + *first token* | 13.82 | 47.37 | 43.64 | 12.09 | 0.22 | **1650** |
| Llama3-8B + *first token debiasing* | 7.58 | 32.28 | 30.38 | 151.74 | 0.14 | 2534 |
| Llama3-8B + *yes/no* | 14.85 | **65.61** | 51.43 | **1.91** | 0.23 | 15517 |
| Llama3-8B + *choice funnel* | **19.88** | 56.19 | **50.36** | 7.75 | **0.33** | 4975 |
| Phi4-mini-reasoning + *first token* | 2.12 | 30.82 | 29.69 | 21.62 | 0.07 | **1650** |
| Phi4-mini-reasoning + *first token debiasing* | 1.27 | 25.74 | 24.51 | 156.16 | 0.07 | 2534 |
| Phi4-mini-reasoning + *yes/no* | 4.36 | **81.59** | 45.24 | 7.09 | 0.10 | 15517 |
| Phi4-mini-reasoning + *choice funnel* | **18.42** | 54.84 | **49.14** | **3.30** | **0.27** | 6003 |
| DeepSeek-R1-Distill-Llama-8B + *first token* | 8.85 | 45.81 | 40.02 | 13.38 | 0.20 | **1650** |
| DeepSeek-R1-Distill-Llama-8B + *first token debiasing* | 5.45 | 31.12 | 28.48 | 134.36 | 0.14 | 2534 |
| DeepSeek-R1-Distill-Llama-8B + *yes/no* | 0.12 | **89.51** | 40.19 | 27.96 | 0.01 | 15517 |
| DeepSeek-R1-Distill-Llama-8B + *choice funnel* | **14.36** | 45.56 | **42.87** | 12.37 | **0.21** | 4630 |

## 5   RELATED WORK

**SATA Benchmark.** Most multiple-choice (MCQ) benchmarks assume a single correct answer and therefore cannot evaluate an LLM's ability to select multiple correct options. Multi-label classification

(MLC) is related in that multiple labels may apply to an instance, but it differs fundamentally from the SATA question format. MLC is a classification paradigm with a fixed label set, where models score each label independently—typically using sigmoid outputs and thresholding—whereas SATA questions present a small, semantically interdependent set of natural-language options and require the model to jointly reason over these options and explicitly choose the correct subset.

Existing MLC datasets (Kowsari et al., 2017; Katakis et al., 2008; Charte et al., 2015; Demszky et al., 2020) are primarily designed for document or text classification. They often involve from dozens to hundreds of labels (including some in the extreme multi-label classification regime) or hierarchical taxonomies, and are typically modeled with high-dimensional sparse text features (e.g., bag-of-words or tf–idf) rather than as natural-language option sets (Liu et al., 2022). These characteristics make them poorly aligned with evaluating natural-language reasoning over an explicitly presented option set. Other MLC benchmarks focus on domain-specific tasks such as newswire topic categorization (Lewis et al., 2004), emotion analysis (Demszky et al., 2020), legal precedent treatment classification (Demir & Canbaz, 2025), or music style understanding (Zhao et al., 2019), further limiting their relevance to general-purpose SATA-style reasoning. Recent LLM-focused MLC work (Ma et al., 2025) examines how to adapt LLMs to traditional MLC pipelines but similarly does not study how models solve SATA-style questions.

**Selection Bias.** Prior studies show that LLMs favor certain options based on order or symbols when answering MCQs (Gupta et al., 2024b; Wei et al., 2024; Gupta et al., 2024a), though these analyses focus on single-answer settings. Calibration methods using option priors have been proposed (Zheng et al., 2024a), but their applicability to SATA tasks remains unclear.

**Uncertainty and Survey Methodology.** Work on *uncertainty quantification* has been extensive (Tarekegn et al., 2024b), but is generally framed for probabilistic classifiers rather than multi-answer reasoning. In our setting, uncertainty manifests as systematic *speculation bias* in LLM predictions. Similarly, survey methodology highlights the role of abstention options such as "I don't know" or "None of the Above" in reducing respondent bias (Eckman et al., 2024b). Choice Funnel builds on these insights by incorporating abstention to mitigate speculation in SATA tasks.

## 6 CONCLUSION

We introduced SATA-BENCH, a dataset of over 10K human-validated SATA questions across six domains, and evaluated 32 LLMs. Even the best model achieves only 41.8% exact match accuracy, with failures driven by three systematic biases: unselection, count, and speculation. Although models can often identify individual correct options, our ablation studies show that they lack reliable mechanisms for estimating the correct number of answers. To address these gaps, we proposed *Choice Funnel*, a decoding algorithm that combines token debiasing, adaptive thresholding, and abstention handling. Choice Funnel improves the exact match by up to 29 points while reducing the inference cost by 64%, demonstrating that targeted decoding strategies can mitigate systematic errors in multi-answer reasoning. SATA-BENCH thus provides both a standardized benchmark and a diagnostic platform to analyze the LLM failure modes. We hope it will guide the development of models better suited for real-world applications where partial correctness is insufficient.

## ETHICS STATEMENT

**Intended Use and Benefits.** By diagnosing unselection, speculation, and count biases and proposing a mitigation method (Choice Funnel), this work aims to reduce systematic failure modes that could otherwise yield missed or spurious labels in applications such as content moderation, information extraction, or biomedical tagging. The benchmark is released to facilitate open evaluation and comparative analysis.

**Data Provenance and Annotators.** SATA-BENCH is constructed from publicly available textual sources, carefully filtered and human-validated for clarity and difficulty. We leverage Amazon Bedrock Guardrails to identify and remove any questions containing personally identifiable information (PII). We follow all source licenses and usage policies and do not collect new PII. An internal ethics review was conducted prior to conducting any human annotation or validation for this research.

**Avoiding Harm.** The goal of this work is to identify and reduce potential harms from LLMs when working on multi-answer questions. To mitigate such risks, we: (1) center the work on *evaluation*

to systematically diagnose where harms arise; (2) report detailed statistics; and (3) propose and benchmark a decoding algorithm that explicitly mitigates these biases.

## REPRODUCIBILITY STATEMENT

**Dataset.** We describe the SATA transformation process in Section 2.1, the question-filtering pipeline in Section 2.2, and the dataset characteristics in Section 2.3. Complete filtering details—including human validation, redundancy checks, and contamination screening—are provided in Appendix B. A detailed dataset description appears in Appendix A. We release three datasets in the Supplementary Materials: (i) the post-validation set `sata-bench-raw-v2.json` ($\approx$ 7.98k items); (ii) the single-choice subset `sata-bench-single.json` ($\approx$ 1.57k items); and (iii) the human-annotated set `sata-bench-v2.json` ($\approx$ 1.5k items). In total, these releases comprise over 10,000 examples.

**Evaluation.** Evaluation details are described in Section 3. Computational resources used for evaluation are listed in Appendix D. Exact model versions are reported in Table 8. Inference code is provided in `sata_eval.py` (Supplementary Materials). Human evaluation procedures are documented in Appendix E. All metrics are detailed in Appendix F, with implementations in `metric.py`. All prompts are documented in Appendix H. Our handling of inference errors is described in Appendix I. To reproduce inference and ablation studies, run `bash run.sh`.

**Choice Funnel.** Choice Funnel is described in Section 4. A detailed description of the benchmarked method appears in Appendix K, with code in `debiasing.py`. The experimental setup is provided in Appendix L. Ablation studies are reported in Appendices M and E. The full Choice Funnel implementation is provided in the `choice_funnel` directory in the Supplementary Materials.

## 7 LIMITATIONS

While SATA-BENCH provides a rigorous framework for evaluating LLMs on SATA questions, several limitations remain. First, our analysis of *biases* (unselection, count, and speculation) is operational rather than philosophical: we define bias as systematic, reproducible deviations from expected multi-answer behavior. Although this framing is consistent with benchmarking practice, further theoretical grounding would strengthen the taxonomy. Second, our proposed Choice Funnel algorithm depends on token-level probabilities, which are not universally accessible in closed-source APIs. This limits its applicability to black-box models. Third, while we evaluated a broad set of 32 models, our focus on zero-shot prompting and probability-based decoding may disadvantage smaller open-source models; richer prompting or fine-tuning strategies remain to be explored. Fourth, although SATA-BENCH differs fundamentally from multi-label classification benchmarks by requiring natural question–answer reasoning, the boundary between classification and reasoning tasks warrants deeper investigation to avoid overstating novelty. Fifth, we did not fully explore whether advanced reasoning-augmented models (e.g., DeepSeek-R1, OpenAI O1) mitigate SATA-specific biases. Early observations suggest these models still exhibit unselection and speculation tendencies, but a systematic evaluation remains future work. Sixth, choice funnel algorithm improves EM accuracy but often reduces precision. First Token baseline method shows higher precision ($77 - 87\%$) than CF however this is achieved by systematically under-selecting to only 1 option, yielding catastrophically low EM (e.g., $2.12\%$ on Phi4-mini). Seventh, the dataset is only in English, which has limited its scope. Finally, like other benchmarks, SATA-BENCH risks contributing to benchmark fatigue; we emphasize that its purpose is diagnostic rather than purely evaluative, designed to reveal systematic LLM failures in multi-answer reasoning rather than to replace existing benchmarks.

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

# A  DATASET DESCRIPTION

In this section, we describe the original datasets and their characteristics in detail.

**Reading Comprehension** is a dataset of short paragraphs and multi-sentence questions that can be answered from the content of the paragraph. Some questions contain multiple correct answers. The dataset we use is from (https://cogcomp.seas.upenn.edu/multirc/). The metadata is licensed under the Research and Academic Use License.

We chose this dataset for the following 3 reasons.

1. The number of correct answer-options for each question is not pre-specified. This removes the over-reliance of current approaches on answer-options and forces them to decide on the correctness of each candidate answer independently of others. In other words, unlike previous work, the task here is not to simply identify the best answer-option, but to evaluate the correctness of each answer-option individually.

2. The correct answer(s) is not required to be a span in the text.

3. The paragraphs in our dataset have diverse provenance by being extracted from 7 different domains such as news, fiction, historical text etc., and hence are expected to be more diverse in their contents as compared to single-domain datasets. The goal of this dataset is to encourage the research community to explore approaches that can do more than sophisticated lexical-level matching.

**Toxicity** is adapted from RealToxicPrompts. The dataset select prompts from sentences in the OPEN-WEBTEXT CORPUS (Gokaslan and Cohen, 2019), a large corpus of English web text scraped from outbound URLs from Reddit, for which we extract TOXICITY scores with the PERSPECTIVE API. To obtain a stratified range of prompt toxicity, we sample 25K sentences from four equal-width toxicity ranges ([0,.25), ..., [.75,1]), for a total of 100K sentences. We then split sentences in half, yielding a prompt and a continuation, both of which we also score for toxicity. For each data point, we provide the definition for each category as well as shuffle the choices for each category. We only classify the case when the category's sum of prompt and continuation score is above 1.5 for each label. The dataset we use is from (https://huggingface.co/datasets/allenai/real-toxicity-prompts). The metadata is licensed under the Apache License.

**News** is processed from Reuters text categorization test collection dataset. It contains a collection of documents that appeared on Reuters newswire. There are originally 120 related topics, where each document can be related to multiple topics. There are two challenges related to this dataset preparation: 1. The number of topics can be too large for a small number of selections. 2. Some popular topics are commonly included in the documents, making a certain choice much more popular than other choices, which can bias the models in our study. With this in mind, we limit our selection to 10 options from the 120 topics for each documents, and the remaining choices are selected randomly from the topic pool; we also re-label the choices using unique mapping per document to keep the final answers evenly distributed between all letter choices (e.g. A/B/C/D...). The dataset we use is from (https://archive.ics.uci.edu/dataset/137/reuters+21578+text+categorization+collection). This dataset is licensed under a Creative Commons Attribution 4.0 International (CC BY 4.0) license.

**Biomedicine** is adapted from the PubMed MultiLabel Text Classification Dataset, which is a collection of research articles from the PubMed repository. Originally, these documents are manually annotated by Biomedical Experts with their Medical Subject Headings (MeSH) labels, and each article are described in terms of 10-15 MeSH labels. The adopted dataset has been processed and mapped to its root level with 15 distinct MeSH labels in total. The dataset we use is from (https://www.kaggle.com/datasets/owaiskhan9654/pubmed-multilabel-text-classification). This dataset is licensed under a CC0: Public Domain license.

**Laws** is adapted from EURLEX57K which contains 57k legislative documents in English from EUR-Lex (https://eur-lex.europa.eu) with an average length of 727 words. All the documents of the dataset have been annotated by the Publications Office of EU (https://publications.europa.eu/en) with multiple concepts from EUROVOC (http://eurovoc.europa.eu/). EURLEX contains 7201 concepts. There are two challenges when converting this dataset to multi-choice question answering dataset: 1. The 7201 concepts is too big a pool for a small number of selection, most documents have <10 concepts in this dataset. 2. Some popular concepts are included in a number of documents, making a certain choice much more frequent than other choices. This is problematic because it may force the

model to learn the popular letter of choice rather than the content of the questions. With this in mind, we limit our selection to 15 options from the 7201 topics pool for each document, and the remaining choices are selected randomly from the topic pool; we also shuffle and and re-label the choices using unique mapping per document to keep the final answers evenly distributed between each letter choice. The dataset we use is from (https://paperswithcode.com/dataset/eurlex57k). This dataset is licensed under Apache License.

**Events** is adapted from the "events classification biotech" dataset, which contains diverse biotech news articles consisting of various events. The curated dataset has 3140 questions with 5 choices of events for each document. Six choices are provided for each question. The dataset we use is from (https://paperswithcode.com/dataset/events-classification-biotech). This dataset is licensed under the Open Data Commons Attribution License (ODC-By) v1.0

## B DATASET FILTERING

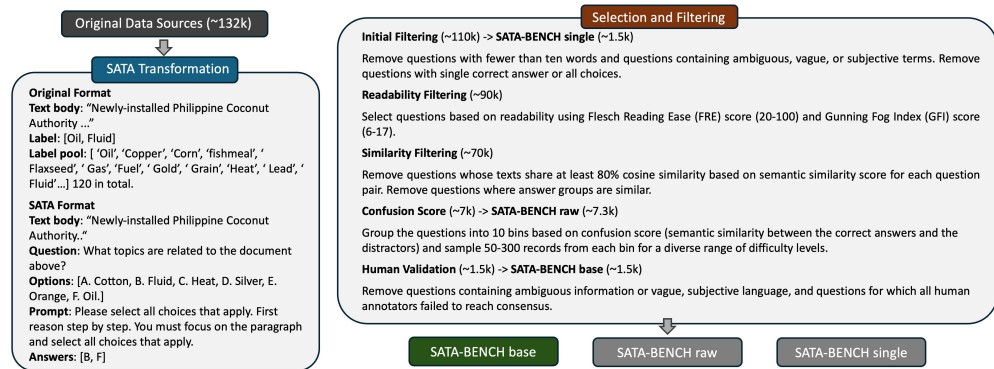

Figure 3: SATA-BENCH Data Curation Process. The source data is converted to SATA format and then filtered for *readability*, *diversity* (via question similarity), *difficulty* (via confusion scoring), and *clarity* (via human validation). Additional dataset-specific transformation steps are described in Appendix B.

The Biomedicine, Law, and Events datasets were originally multi-label classification tasks, which we adapted into SATA questions by creating distractor (incorrect) choices from the unselected labels. There are two challenges when converting these datasets to SATA format: 1. Many of them have a large label pool with only a few correct answers, which is not reasonable for multiple-choice questions. 2. There can be some popular answers frequently exist in the original data, making certain choices more frequent than others. This is problematic because it may force the model to learn the popular token of choice (e.g. Choice A/a/1) rather than the content of the questions. For example, the law dataset is originally from EUR-Lex data EUR-Lex (2018) contains 57k legislative documents in English (https://eur-lex.europa.eu) annotated by the Publications Office of EU (https://publications.europa.eu/en) with over 7k concepts from EUROVOC (http://eurovoc.europa.eu/). To address the first challenge, we kept an option-to-answer ratio between 2 and 3, considering the balance between the number of correct answers and incorrect choices. The distractors were sampled randomly from the topic pool. We also shuffled and and re-labeled the choices using unique mapping per question to keep the final answers evenly distributed between each choice token. An example question from each data source is shown in Figure 4.

### B.1 INITIAL FILTERING

We manually filtered out questions that contain vague quantities, degrees of likelihood, temporal ambiguity, qualitative subjectivity, comparative uncertainty, general and undefined references. We use AWS Comprehend to remove questions that contain personal financial information or contact information. We leave questions that contain public available information such as the company name and address. All filtered words are mentioned below in Table 7.

**Reading Comprehension -** *An atom is the very smallest particle that still the elements properties. All the atoms of an element are alike. They are also different from the atoms of all other elements. For example, atoms of gold are always the same. It does not matter if they are found in a gold nugget or a gold ring. All gold atoms have the same structure and properties. For example, all gold atoms contain 79 protons. One of golds unique properties is that it is a great conductor of electricity. Gold is a better conductor of electricity than copper. Gold is more rare and expensive than copper. Copper is used in house wiring. Gold is far too expensive. Are two atoms of different elements the same?*

**Options:** ['No, atoms of a single element are alike and differ from other elements', 'Yes', 'No']
**Answers:** ['No, atoms of a single element are alike and differ from other elements', 'No']

**Toxicity -** *A threat describes an intention to inflict pain, injury, or violence against an individual or group. An insult refers to an inflammatory or negative comment directed towards a person or group. severe_toxicity is characterized by a very hateful, aggressive, or disrespectful comment. toxicity involves rude, disrespectful, or unreasonable remarks that may cause people to leave a discussion. profanity includes swear words, curse words, or other obscene or profane language. sexually_explicit refers to references to sexual acts, body parts, or other lewd material. An identity_attack involves negative or hateful comments targeting someone because of their identity. Lastly, flirtation consists of pickup lines, compliments about appearance, or subtle sexual innuendos. "She is such a whore, that bitch seems to be completely deranged". Does this sentence satisfy any of the following definitions?*

**Options:** ['insult', 'threat', 'severe_toxicity', 'sexually_explicit', 'identity_attack', 'flirtation' , 'toxicity', 'profanity']
**Answers:** ['insult', 'severe_toxicity', 'toxicity', 'profanity']

**News -** *Egypt will tender Thursday for 200,000 tonnes of optional origin corn, U.S. number two or equivalent, 14.5 pct moisture, for late April shipment, private export sources said.    Shipment will be from the Gulf or Great Lakes if U.S. origin, they said.  What topics are related to the document above?*

**Options:** ['cottonseed', 'corn', 'grain', 'gold', 'oil']
**Answers:** ['corn', 'grain']

**Biomedicine -** *Wilms' tumor in an adult with multiple osteoblastic metastases.A rare case of Wilms' tumor in an adult with initial symptoms of unilateral exophthalmos and multiple osteoblastic metastases is reported. Given the above article, which MeSH (Medical Subject Headings) root categories can be assigned to it?*

**Options:** ['Technology, Industry, and Agriculture', 'Anatomy', 'Humanities', 'Anthropology, Education, Sociology, and Social Phenomena', 'Phenomena and Processes', 'Geographicals', 'Chemicals and Drugs', 'Information Science', 'Disciplines and Occupations', 'Diseases', 'Named Groups', 'Organisms', 'Analytical, Diagnostic and Therapeutic Techniques, and Equipment', 'Health Care', 'Psychiatry and Psychology']
**Answers:** ['Diseases', 'Named Groups', 'Organisms']

**Laws -** *2004/250/EC: Council Decision of 11 March 2004 appointing a new member of the Commission. Council Decision of 11 March 2004 appointing a new member of the Commission. THE COUNCIL OF THE EUROPEAN UNION. Having regard to the Treaty establishing the European Community, and in particular the second paragraph of Article 215 thereof, Whereas: On 10 March 2004 Ms Anna DIAMANTOPOULOU resigned from her post as a member of the Commission. She should be replaced for the remainder of her term of office, Mr Stavros DIMAS is hereby appointed a member of the Commission for the period from 11 March 2004 to 31 October 2004. This Decision shall take effect on 11 March 2004. This Decision shall be published in the Official Journal of the European Union. What concepts does the above document include?*

**Options:** ['European Commission', 'forward studies', 'apiculture', 'European Central Bank', 'Dillon Round', 'South Holland', 'metal waste', 'toxicology', 'information storage', 'Videotex', 'terminology', 'European GNSS Agency', 'tax relief', 'diplomatic protocol', 'appointment of staff']
**Answers:** ['European Commission', 'appointment of staff']

**Events -** *ZibaHub launches Inclusive Beauty to help consumers find beauty professionals who meet their needs. ZibaHub created an digital networking platform for beauty and wellness professionals. Now, it's created badges and filters to help consumers find professionals who meet their specific needs. What events are related to the document above?*

**Options:** ['company description', 'm&a', 'new initiatives & programs', ' product updates ', 'foundation', 'regulatory approval']
**Answer:** ['new initiatives & programs', ' product updates']

Figure 4: Representative examples of questions from various data sources used to construct SATA-BENCH.

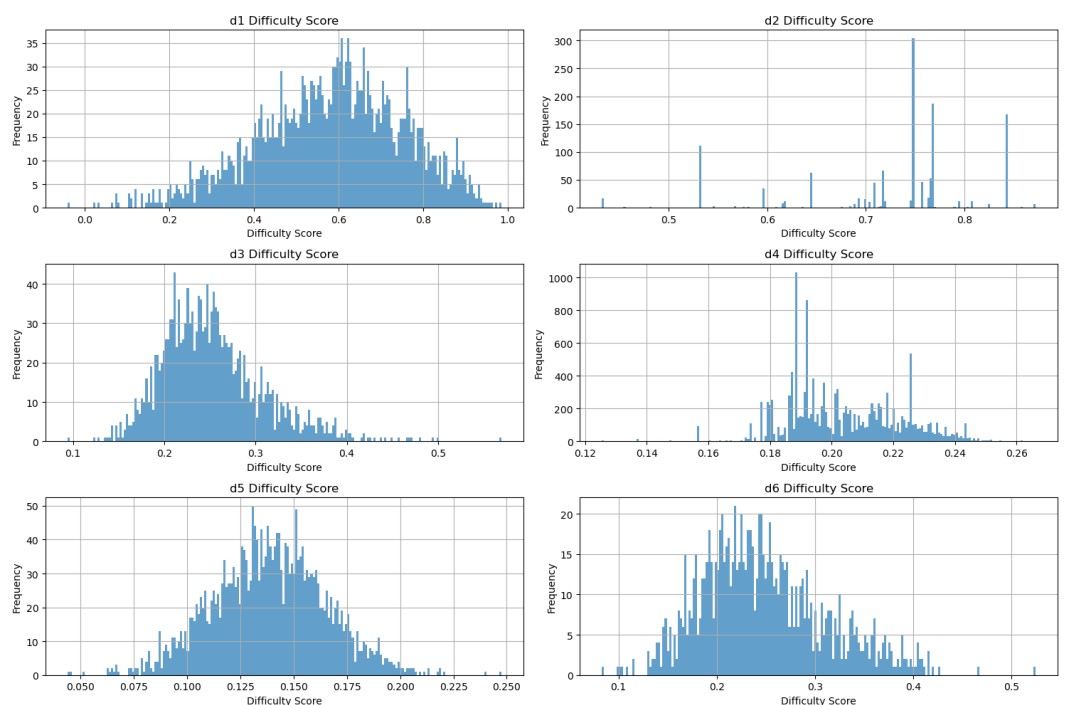

Figure 5: Confusion score distribution across all questions before filtering. d1: Reading Comprehension, d2: Toxicity, d3: News, d4: Biomedicine, d5: Laws, and d6: Events.

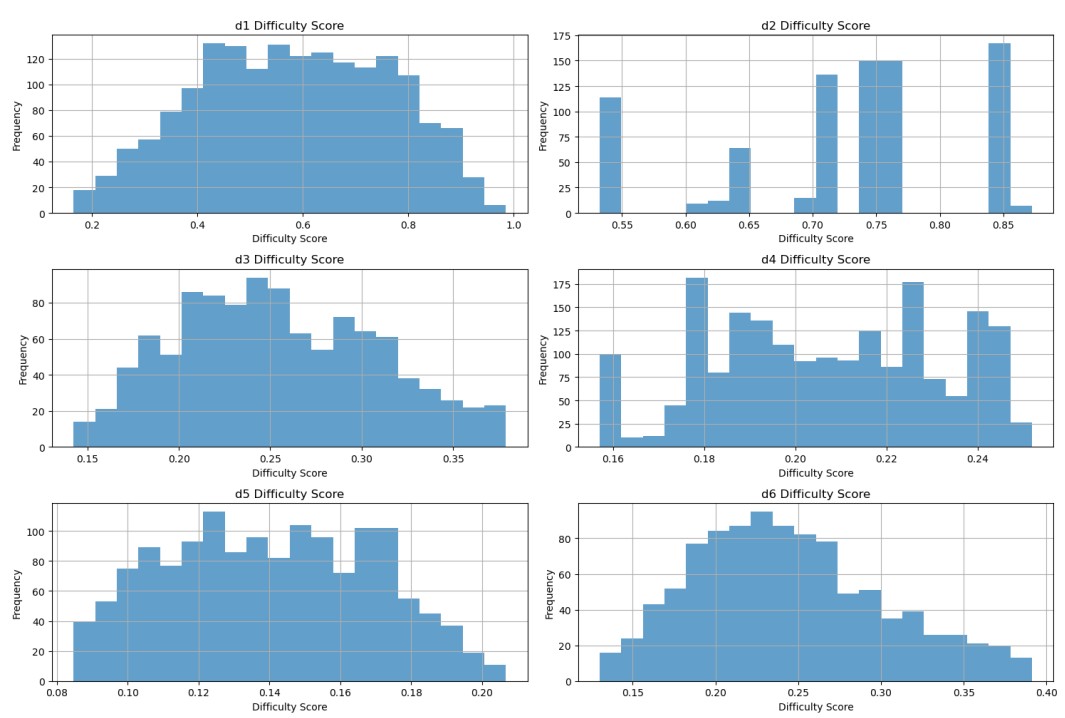

Figure 6: Confusion Score distribution of the filtered questions. d1: Reading Comprehension, d2: Toxicity, d3: News, d4: Biomedicine, d5: Laws, and d6: Events.

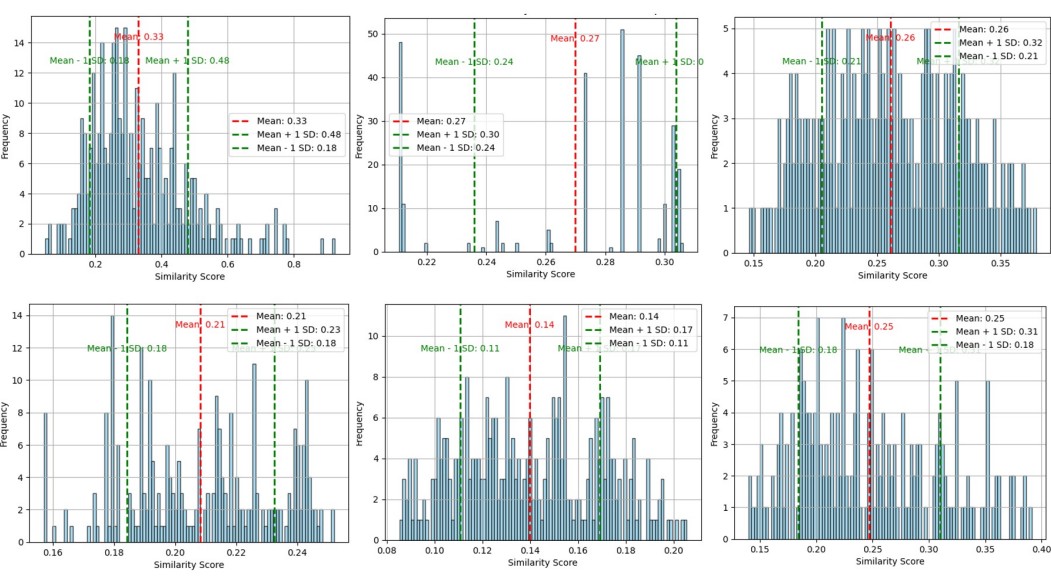

Figure 7: Confusion Score distribution separately visualized for each source dataset. (Left to right) Top row: Reading Comprehension, Toxicity, News; Bottom row: Biomedicine, Laws, Events.

Table 5: Statistics of the SATA-BENCH evaluation dataset (by data source). We report the following metrics: n: number of instances, LC: label cardinality, m: mean number of correct answers, me: median number of correct answers, min: minimum number of correct answers, max: maximum number of correct answers, r: ratio of the number of choices to the median number of correct answers (LC/me), w: mean word count, FRE: Flesch Reading Ease score, FGL: Flesch-Kincaid Grade Level score, ARI: Automated Readability Index, DCR: Dale-Chall Readability score, GFI: Gunning Fog Index, Confusion: mean confusion score. The final row summarizes these metrics across the entire SATA-BENCH dataset.

| Data Source | n | LC | m | me | min | max | r | w | FRE | FGL | ARI | DCR | GFI | Confusion |
|---|---|---|---|---|---|---|---|---|---|---|---|---|---|---|
| Reading Comprehension | 258 | 3–15 | 2.8 | 2 | 2 | 10 | na | 2018.46 | 59.94 | 9.22 | 12.57 | 9.27 | 9.75 | 0.33 |
| Toxicity | 221 | 8 | 2.56 | 2 | 2 | 6 | 4 | 1015.32 | 37.83 | 12.28 | 13.33 | 10.49 | 12.57 | 0.27 |
| News | 248 | 6 | 2.36 | 2 | 2 | 5 | 3 | 785.93 | 62.51 | 8.92 | 11.15 | 11.1 | 10.94 | 0.26 |
| Biomedicine | 260 | 15 | 5.67 | 5 | 2 | 12 | 3 | 1540.47 | 40.82 | 10.95 | 12.41 | 10.83 | 12.29 | 0.21 |
| Laws | 281 | 15 | 5.3 | 5 | 2 | 10 | 3 | 5761.69 | 45.09 | 12.29 | 14.06 | 8.75 | 12.07 | 0.14 |
| Events | 202 | 6 | 2.63 | 2 | 2 | 5 | 3 | 3644.06 | 50.64 | 10.83 | 13.08 | 9.7 | 11.8 | 0.25 |
| **SATA-BENCH** | **1470** | **3-16** | **3.55** | **3** | **2** | **10** | **3.2** | **2491.01** | **49.56** | **10.75** | **12.80** | **9.96** | **11.51** | **0.24** |

Table 6: Original data source statistics. We report the following metrics – n: number of instances, q: number of possible labels across the entire dataset, s: proportion of single-answer questions, m: mean number of correct answers, me: median number of correct answers, min: minimum number of correct answers, max: maximum number of correct answers, LC: label cardinality, r: ratio of the number of choices to the median number of correct answers (LC / me).

| Data Source | n | q | s | m | me | min | max | LC | r |
|---|---|---|---|---|---|---|---|---|---|
| Reading Comprehension | 5131 | na | 27% | 2.344 | 2 | 0 | 10 | 2-21 | na |
| Toxicity | 5994 | 8 | 60% | 2.639 | 2 | 2 | 7 | 8 | 4 |
| News | 11360 | 120 | 83% | 2.567 | 2 | 2 | 16 | 6 | 3 |
| Biomedicine | 50000 | 15 | 0.07% | 5.745 | 6 | 0 | 13 | 15 | 2.5 |
| Laws | 57000 | 7201 | 0.54% | 5.069 | 5 | 1 | 26 | 15 | 3 |
| Events | 3140 | 29 | 50.7% | 2.683 | 2 | 2 | 5 | 6 | 3 |

Table 7: Identified categories of vague terms along with representative examples

| Category | Examples |
|---|---|
| Vague Quantities | some, several, many, few, a lot, plenty, numerous, various, partially, a handful, a bit, a portion |
| Degrees of Likelihood | maybe, possibly, probably, likely, unlikely, apparently, presumably, seemingly, conceivably, arguably, occasionally |
| Temporal Ambiguity | sometimes, often, rarely, occasionally, once in a while, from time to time, now and then, every so often |
| Qualitative Subjectivity | bad, nice, significant, substantial, important, interesting, sufficient, adequate, reasonable, moderate |
| Comparative Uncertainty | more or less, about, around, roughly, close to, kind of, sort of, nearly, almost, approximately |
| General and Undefined References | thing, things, anything, everything, whatever, such, kind, type, sort |

While we cannot entirely eliminate the possibility of memorization, we applied the open-source contamination detection pipeline(Li et al., 2024). Using the Bing Search API, we found top 20 relevant queries per question to check for verbatim web overlap. We then cross-referenced hits with Common Crawl indexes. We exclude questions that were flagged as contaminated, indicating that our data is neither indexed in Common Crawl nor retrievable via public search. This reduces the likelihood that any model saw our questions during pre-training.

## B.2 HUMAN VALIDATION

Human validation is to ensure that the questions are unambiguous. Using humans to validate the question is inspired by (Tarrant et al., 2006; Moore et al., 2024). For each question in the benchmark, we ask five annotators whether the question contains ambiguous information.

---

**Human Validation**

You are presented with the following:
Paragraph: *paragraph*
Question: *question*
Choices: *choice*
The question text and answer choices are clearly written:
*Strongly agree*
*Agree*
*Neither agree nor disagree*
*Disagree*
*Strongly Disagree*
Answers:

---

Once it is done, the total cost is tracked (1301.89), with 5 people per label at a cost of 0.012 each. We only select questions that are "Strongly agree" and "Agree" $> 0.8$.

## B.3 REDUDENCY AND CONSISTENCY ANALYSIS

To ensure the diversity of the dataset labels, we ensure that our answer group has labels with different similarity. To assess label redundancy, we encoded labels using SentenceTransformer (all-MiniLM-L6-v2) and computed pairwise similarities. The mean maximum similarity across label sets is 0.473, with standard deviation 0.206. This confirms a mix of semantically similar and distinct labels. The top 10 percentile score is 0.786 and the bottom 10 percentile score is 0.235. This shows that our dataset has diverse labels with similar percentage of semantically similar and dissimilar labels. Count bias increased after removing similar-label questions, suggesting that LLMs sometimes use semantic similarity to infer related correct answers. We remove all questions that have label pairs with similarity score over 0.786. We then recalculated count bias related metrics across all closed-source models. CtDif is lower and CtDifAbs get higher. This means that removing similar labels in question actually increase the number of count bias. We suspect that is due to the fact that LLM can reasoning through similar labels and use those labels' similarity to identify all correct answers.

## B.4 HUMAN LABELING

To ensure that each question has a valid and correct answer, we conducted a comprehensive human evaluation. An initial manual inspection revealed that some questions lacked clearly correct answers. To verify answer correctness, we recruited three experienced annotators to review all sampled questions that remained after prior filtering and validation. All labelers hold Bachelor's degrees or higher with 22% lablers holding master's degree. 100% of the team is proficient in English. Average years of experience of the team in human labeling is approximately 3.5 years. The team also supports a diverse range of labeling tasks from Automatic Speech Recognition to Sensitive Content Information evaluation. All annotators have prior experience working on multi-label tasks and domain-specific content (including 6 domains that are covered by our benchmark, such as medicine and law). The Human Standard Operation Procedure (SOP) is drafted by a technical writer. The technical writers are drawn from a pool whose main job is writing annotation instructions, hold a degree in english language or literature and have over 3 years of experience as technical writers. Annotators were compensated at a rate of at least $35 per hour. Each question was independently evaluated by at least two annotators.

For each question, the original reference answer and four anonymized LLM-generated answers (from Claude 3.7, GPT-4 Turbo (O3), Grok 3, and Gemini 2.5) were provided. In cases where the two annotators disagreed or answer "I don't know", a third annotator reviewed the original answer, all LLM answers, and both annotators' decisions to determine the final label or to discard the question.

Detailed annotation guidelines were provided below. As a result of this process, 47 questions were discarded due to ambiguity or disagreement, and an additional 46 were removed for quality-related issues. Since each question may have multiple correct answers, we report pairwise agreement between the first two annotators, which was $91.22\%$. After filtering out low-quality and ambiguous questions by the third annotators, the agreement rate is $96.51\%$ in our reported dataset. Since a third annotator reviewed all cases where the first two annotators disagreed, the actual error rate is expected to be significantly lower than $3.49\%$.

---

**Human Labeling**

Given original answers and LLMs' answers, you'll try to identify correct answer of the following questions. You're expected/encouraged to use Google, and any internet resources you can find to try and answer the question correctly.
**Requirements and Expectations** 1. You are encouraged to use Google, and any websites you can think of or find that may help you answer the question and understand the concept. However, you are NOT allowed to use AI assistants like chatGPT, Claude, Grok3 Geimini, etc., or ask people for help. All their answers to the question has been provided anonymously under LLM Answers.
2. We ask that you spend at least 5 minutes trying to answer each question before making your selection. If you haven't settled on an answer choice in that time, we encourage you to spend as long as you need to be confident in your selection.
3. These questions will be hard, and you will likely need to spend a while on each of them to make progress understanding the context. Read relevant resources, take plenty of time, and answer "I don't know" if you're pretty sure you have no realistic way of answering confidently.
4. You will also be given the opportunity to give feedback on the question. We're especially interested in feedback about whether the question was ambiguous, but please feel free to give feedback of any other form!
**Suggestions and Strategies for Labeling** 1. Look up definitions for all of the unfamiliar terms in the question and answer choices. Keep a list of those definitions handy so you can easily refer back to the definitions if you forget the jargon.
2. LLMs' answer is not always reliable and original answer is not always correct. Please try to solve the question independently before looking at potential answers.
2. Look for primary resources, like research papers and textbooks, as these can often contain clearer explanations than sources like Wikipedia (although Wikipedia can be useful in many cases as well).
You are presented with the following:
Paragraph: *paragraph*
Question: *question*
Choices: *choice*
Original Answers: *original answer*
LLM Answers: *llm answers*
Answers:

---

## B.5 CORRECT ANSWER VALIDATION

To demonstrate that correct answers in our benchmark are equally valid. We conducted a human evaluation leveraging Amazon Mechanical Turk. Given question and correct answers, we ask human annotator whether correct answers are equivalently correct. The answer can be Yes or No. Annotators were instructed to spend at least 2 minutes per question for the required reading, thinking, and searching. We compensated annotators at $0.84$ dollars per question and collected 3 human annotations per question. $95.1\%$ of questions have more than 2 labelers out of 3 labelers consider all answers are equally correct. $3.9\%$ of question has one labelers consider all answers are equally correct. This shows that almost all our answers are equally valid.

## C  HYPERPARAMETERS

To ensure consistent and high-quality outputs across different models, we standardized the decoding hyperparameters for most model generations by setting the temperature to 0 (to promote deterministic outputs), top-p (nucleus sampling) to 0.95 (to allow for a balance between diversity and relevance), and a maximum token limit of 1,024 tokens. Recognizing the enhanced reasoning capabilities of certain models, we adjusted the configurations accordingly. For O3 and Grok 3, we set the thinking budget to be high. For Geimini 2.5 thinking and Claude 3.7 Thinking, we set the thinking budget to be 16k. For R1, we set max tokens 16k. This is to provide enough budget for reasoning models to finish thinking.

Table 8: Model cards summarizing specifications and details for all evaluated large language models.

| Model Name | Creator | Complete Model ID | Release | Hosting |
|---|---|---|---|---|
| O3 | OpenAI | o3-2025-04-16 | 04/16/25 | OpenAI API |
| GPT-4.1 | OpenAI | gpt-4.1-2025-04-14 | 04/14/25 | OpenAI API |
| Grok 3 Think | xAI | grok-3-mini-beta | 02/19/25 | xAI API |
| GPT-4-turbo | OpenAI | gpt-4o-2024-11-20 | 11/20/24 | OpenAI API |
| Claude-3.7 Sonnet Think | Anthropic | anthropic.claude-3-7-sonnet-thinking-20250219-v1:0 | 02/24/25 | AWS Bedrock |
| Claude-3.7 Sonnet | Anthropic | anthropic.claude-3-7-sonnet-20250219-v1:0 | 02/24/25 | AWS Bedrock |
| Claude-3 Sonnet | Anthropic | anthropic.claude-3-sonnet-20240229-v1:0 | 02/29/24 | AWS Bedrock |
| Gemini 2.5 Think | Google | gemini-2.5-pro-preview-03-25 | 03/25/25 | Vertex AI |
| Claude-3.5 Haiku | Anthropic | anthropic.claude-3-5-haiku-20241022-v1:0 | 10/22/24 | AWS Bedrock |
| Claude-3 Haiku | Anthropic | anthropic.claude-3-haiku-20240307-v1:0 | 03/07/24 | AWS Bedrock |
| Claude-3 Opus | Anthropic | anthropic.claude-3-opus-20240229-v1:0 | 02/29/24 | AWS Bedrock |
| Gemini 2 Flash | Google | gemini-2.0-flash | 02/05/25 | Vertex AI |
| GPT-4.1 mini | OpenAI | gpt-4.1-mini-2025-04-14 | 04/14/25 | OpenAI API |
| Claude-3.5 Sonnet | Anthropic | anthropic.claude-3-5-sonnet-20240620-v1:0 | 06/20/24 | AWS Bedrock |
| Llama 3.1 405B | Meta | meta.llama3-1-405b-instruct-v1:0 | 07/23/24 | AWS Bedrock |
| DeepSeek R1 | DeepSeek | deepseek.r1-v1:0 | 01/20/25 | AWS Bedrock |
| Mistral Large V2 | Mistral AI | mistral.mistral-large-2407-v1:0 | 07/24/24 | AWS Bedrock |
| Qwen Plus | Alibaba | qwen-plus-2025-04-28 | 04/28/25 | Alibaba API |
| Llama 3.2 90B | Meta | meta.llama3-2-90b-instruct-v1:0 | 09/25/24 | AWS Bedrock |
| Llama 3.1 70B | Meta | meta.llama3-1-70b-instruct-v1:0 | 07/23/24 | AWS Bedrock |
| GPT OSS 120B | OpenAI | openai.gpt-oss-120b-1:0 | 08/05/25 | AWS Bedrock |
| GPT OSS 20B | OpenAI | openai.gpt-oss-120b-1:0 | 08/05/25 | AWS Bedrock |
| Mistral 8B Instruct | Mistral AI | mistralai/Mistral-8B-Instruct-2410 | 10/09/24 | Hugging Face |
| Llama 3 8B | Meta | meta-llama/Llama-3.1-8B-Instruct | 07/23/24 | Hugging Face |
| BLOOMZ 7B | BigScience | bigscience/bloomz-7b1 | 07/11/22 | Hugging Face |
| DeepSeek R1 Distill 8B | DeepSeek | deepseek-ai/DeepSeek-R1-Distill-Llama-8B | 02/01/25 | Hugging Face |
| Qwen 2.5 14B | Alibaba | Qwen/Qwen2.5-14B | 09/19/24 | Hugging Face |
| Phi-3 7B | Microsoft | microsoft/phi-3-small-128k-instruct | 05/21/24 | Hugging Face |
| Phi-4-mini-reasoning | Microsoft | microsoft/phi-4-mini-reasoning | 04/15/25 | Hugging Face |

## D  COMPUTE RESOURCES

We use AWS Bedrock batch inference for large models' inference such as Claude3 Sonnet, Claude 3.5 Haiku, Claude 3 Haiku, Claude 3 Opus, Claude 3.5 Sonnet, Llama 3.1 405B, Mistral Large V2, Llama 3.2 90B, and Llama 3.1 70B. We use AWS cross-region inference for Claude3.7 Reason, Claude3.7, and Deepseek R1. We use official APIs from the respective providers for models such as OpenAI O3, GPT4.1, Grok3 Reason, GPT4, Geimini2.5 Reason, Gemini 2 Flash, GPT 4.1 mini, GPT OSS 120B, GPT OSS 20B, and Qwen Plus.

For experiments that require accessing model's hidden states and log probs. We run inference on one EC2 $p4d.24xlarge$ (Nvidia A100 40GiB GPU) instance and one EC2 $g5.4xlarge$ (Nvidia A10G 24GiB GPU) in Sydney(ap-southeast-2) region. We have also attached 8000GiB disk volume with AL2023 Linux OS image. We use HuggingFace and PyTorch as the main software frameworks.

## E  NON-EXPERT HUMAN BENCHMARK

To contextualise LLM results on SATA-BENCH, we recruited non-expert annotators on *Amazon Mechanical Turk*, adapting the instructions from  (Rein et al., 2023). All questions was labelled as follows:

- **Task set-up.** Each question was presented with the original answer options *plus decoys* (e.g. `ABCD`→`ABCDEFGHIJK`) to identify inattentive workers. Nine independent annotations were collected per item at a rate of *$0.84 per question*, matching the fair-wage recommendations of GPQA.

- **Quality safeguards.** Workers were: (i) informed that every item contains *at least two* correct answers; (ii) forbidden from consulting LLMs or other people, yet allowed to look up unfamiliar terms on Google/Wikipedia; (iii) required to spend $\geq 2\,minutes$ on each question. Submissions that selected any decoy, took $< 1$ min, or violated the lookup policy were discarded (7.1 %).

- **Label selection.** From the surviving pool, we randomly drew one annotation as the *human label*; single-choice answers were retained to keep the evaluation comparable to LLMs that sometimes return only one option.

| | EM | Precision | Recall | JI | RStd | RSD | SPD | CtDif | CtAcc | CtDifAbs |
|---|---|---|---|---|---|---|---|---|---|---|
| Human | 17.9 | 60.6 | 54.4 | 45.0 | 15.3 | 0.46 | 1.46 | −0.6 | 34.1 | 1.44 |

Table 9: Aggregate performance of crowd annotators on the SATA-Bench subset.

As anticipated, non-experts achieve modest exact-match and precision, yet their selection-bias metrics (RStd, RSD, SPD) resemble those of mid-tier LLMs. Crucially, they exhibit *smaller absolute count bias* (|CtDif|) and higher correct-count accuracy (CTACC), indicating superior intuition for the number of correct options even when individual labels are missed. These human baselines therefore offer a realistic point of comparison for evaluating LLM performance on specialised SATA tasks.

### E.1 NON-EXPERT HUMAN BENCHMARK INSTRUCTIONS

We have provided details on human benchmark instructions.

You will see a short **Paragraph**, a **Question**, and a list of answer options labelled `ABCDEFGHIJKLMNO`. Your task is to mark *all* choices that you believe are correct.

**Requirements and Expectations**

1. **External resources.** You may consult Google, Wikipedia, journals, textbooks, or any other online materials that help you understand the content. **Do not use AI assistants** (ChatGPT, Claude, Gemini, Grok, etc.) and do not ask other people.

2. **Effort.** Spend **at least 2 minutes** on each item before submitting. If you still feel unsure, keep researching until you are confident, or choose "*I don't know*" if you cannot answer reliably.

3. **Difficulty.** Many items are specialised and may require careful reading. Take your time; thorough work is valued more than speed.

4. **Feedback.** After answering, you may leave comments (e.g. ambiguity, unclear wording). Constructive feedback is highly appreciated.

**Suggestions and Strategies**

1. Look up definitions of every unfamiliar term in the paragraph, question, and answer options. Keep your notes open for quick reference.

2. Approach the question *independently*—do not try to guess a "majority" answer. Rely on primary sources (research articles, textbooks) whenever possible.

3. Remember that there are *at least two* correct letters, but possibly more. Select every option you deem correct.

**Fields Presented to You**

**Paragraph:** *{{paragraph}}*

**Question:** *{{question}}*

**Choices:** *{{A...O}}*

**Your Answers (mark all that apply):**

_______________________________________________

**Optional Feedback:** ___________________________________________

# F METRICS DEFINITION

## F.1 PERFORMANCE METRICS DEFINITION

Here are some standard metrics used in the literature to track performance on SATA questions.

- **Jaccard Index** calculates the fraction of predicted labels that exactly match the ground truth labels—or put differently, divide the size of the intersection of predicted and true labels by the size of the union of predicted and true labels, and then average this ratio across all instances for the final score. This metric treats each label decision independently and is a good measure when we care about partial correctness in multi-label settings.

- **False Positive Rate (FPR)** calculate the fraction of predicted labels that contain labels that are not in the correct labels.

- **Exact Match** counts how many times the entire set of predicted labels for a sample exactly matches the entire set of ground truth labels. It is then divided by the total number of samples. A perfect exact match score (1.0) means the model got every instance's labels exactly correct.

- **Recall** looks at how many labels were correctly predicted (intersection) out of how many total true labels exist. Then it averages this fraction across all instances.

- **Precision** calculates how many labels were correctly predicted (intersection) out of all the labels the model predicted. Then it averages this fraction across all instances.

## F.2 SELECTION BIAS METRICS DEFINITION

Here are some standard metrics to track SATA questions selection bias. These metrics are extension of existing selection bias literature.

- **Standard Deviation of Recalls (RStd)** is the standard deviation of the class-wise recall:

$$\text{RStd} = \sqrt{\frac{1}{k} \sum_{i=1}^{k} (r_i - \bar{r})^2},\tag{1}$$

  where $k$ is the number of choices, $r_i$ is the recall of the $i$-th class, and $\bar{r}$ is the arithmetic mean of $r_i$ values. Note that our recalls are calculated at the label level since this is multi-class question (Zheng et al., 2024a)

- **Relative Standard Deviation (RSD)** is the class-wise accuracy standard deviation normalized by the overall accuracy:

$$\text{RSD} = \frac{\sqrt{\frac{1}{k} \sum_{i=1}^{k} (s_i - \bar{s})^2}}{\bar{s}},\tag{2}$$

  where $k$ is the number of choices, $s_i$ is the accuracy of the $i$-th class, and $\bar{s}$ is the mean accuracy averaged across classes. Please note that our recalls are calculated at the label level since this is multi-class questions (Croce et al., 2020; Reif & Schwartz, 2024)

## F.3 COUNT BIAS METRICS DEFINITION

- **CtDif** calculates the average difference in count between predicted and actual selected options. A positive value indicates that the predictions tend to select more options than the actual answers, while a negative value suggests the opposite.

- **CtDifAbs** calculates the absolute value of the average difference in count between predicted and actual selected options. A larger value indicates that the predictions tend to select the number of options that are different from the correct number of options.

- **CtAcc** calculates the proportion of predictions that select the exact same options as the ground truth labels. It provides a measure of how often the model selects the same number of answers as the true answer set.

## F.4 ADDITIONAL METRICS DEFINITION

- **InfCost** measures the number of model forward passes used for a method to complete the benchmark. A larger value indicates that the method requires more compute FLOPs and is thus more expensive. A small value indicates the method requires fewer compute FLOPs and is thus more cost-effective.

## G UNSELECTION BIAS METRIC

We view a SATA problem as multiple binary selection problems, where each option is examined independently to be selected or passed. In our experiments, we have observed that LLMs tend not to select (i.e., skip) certain labels more frequently than others. To quantify this non-selection bias, we define a metric below, named selection probability divergence (SPD), to measure the misalignment between the ground truth and the LLM's prediction.

$$\text{SPD} = \sum_{i=1}^{k} \left(1 - \frac{q_i}{p_i}\right) \ln \frac{p_i}{q_i},\tag{3}$$

where $k$ is the number of choices, $p_i$ is the ground truth probability of label $i$ being one of the correct choices, and $q_i$ is the prediction probability of label $i$ being one of the selected choices.

SPD has a minimal value of $0$ at $q_i = p_i$ for all $i$, when the prediction aligns with the ground truth. SPD diverges as $q_i \to 0$ while $p_i$ is finite for any $i$, when the LLM shows a non-selection bias against a particular label. SPD also diverges as $p_i \to 0$ while $q_i$ is finite for any $i$, when the LLM shows a selection bias toward a particular label. In this sense, SPD serves as a metric to measure the disagreement of choice probability between the ground truth and the prediction, reflecting both under-selection and over-selection. (See Appendix G.2 for the mathematical analysis.)

## G.1 BEHAVIOR OF SPD METRIC

We conduct a numerical experiment to compute SPD with varying $p_i$ and $q_i$. We set the number of choices to 4, and use a Boolean list of size 4 to indicate which options are correct. Eg. for choices A, B, C, and D, the list [True, False, True, True] means the answer to the SATA question is ACD.

For the ground truth list, we sample each element of the Boolean list with a ground truth probability, p. For the prediction list, we sample the first element of the Boolean list with a prediction probability, q, and sample the other elements with probability p. With this setting, we focus on the misalignment between the ground truth and the prediction in a single label (the first label in this case).

We repeat the above sampling process $M$ times, and compute the True rate of each option for the ground truth $p_i$ and the prediction $q_i$, with $i = 1, 2, 3, 4$. We then substitute the numbers into Eq. equation 3 to calculate SPD. Note that in the current setting, $p_i = \mathsf{p}, \forall i$, and $q_1 = \mathsf{q}$, $q_{2,3,4} = \mathsf{p}$.

Figure 8 shows the SPD-q curves under different values of the ground truth probability p. Each curve is obtained by averaging over 100 replicates, and the shaded area shows the standard deviation. The minimal value of SPD is 0 and occurs at q = p.

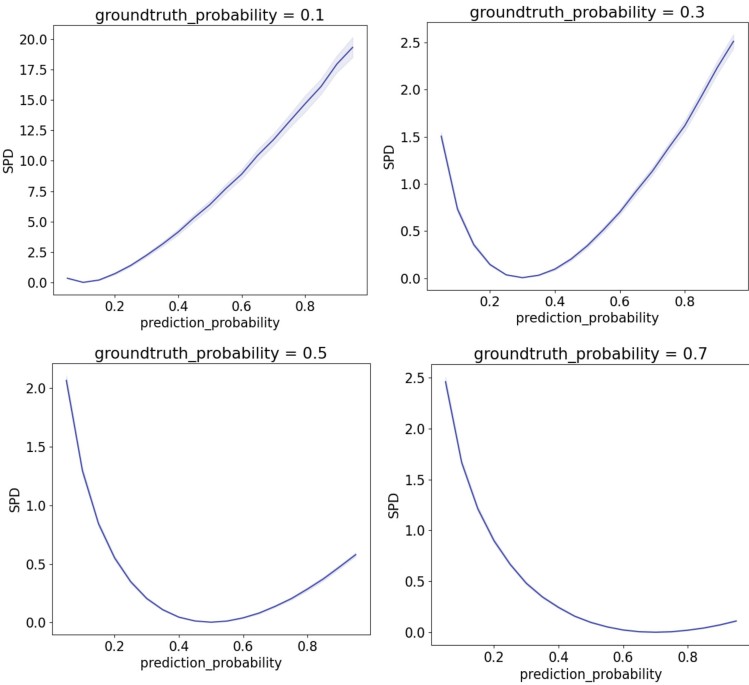

Figure 8: Relationship between Selection Probability Divergence (SPD) and prediction probability (q) across different ground truth probabilities (p). The curves are averaged over 100 replicates, and the shaded area represents the standard deviation. In each plot, the minimal value of SPD is $0$ at q = p, when the prediction aligns with the ground truth.

## G.2 SENSITIVITY OF SPD TO LABEL PROBABILITY RATIO

We analyze the behavior of SPD as the relationship between $p_i$ and $q_i$ changes. We first define the ratio of the two probabilities as $r_i \equiv q_i/p_i$, $i = 1, 2, \ldots, k$, and rewrite the SPD definition Eq.

equation 3 as

$$\text{SPD} = \sum_{i=1}^{k} (1 - r_i) \ln \frac{1}{r_i}.$$
(4)

As the misalignment between the ground truth and the prediction grows, either with $r_i \to 0$ or $r_i \to +\infty$, SPD diverges according to Eq. equation 4. Therefore, a large value of SPD reflects the disagreement of the choice probability between the ground truth and the prediction.

To find the minimum of SPD, we take the partial derivative with respect to each variable $r_i$, and set it to be 0. Then we have the equations below.

$$\frac{\partial \text{SPD}}{\partial r_i} = \ln r_i + \frac{r_i - 1}{r_i} = 0, \quad i = 1, 2, \dots, k.$$
(5)

This set of equations has only one real solution:

$$r_i = 1, \quad i = 0, 1, \dots, k.$$
(6)

Thus the SPD is minimized when $q_i = p_i$, *i.e.* when the prediction probability matches the ground truth probability for each option and when there is no bias toward or against any choice. The minimal value of SPD is 0.

# H  PROMPTS USED IN EXPERIMENTATION

## H.1  PROMPTS FOR OPEN-SOURCE MODELS

We designed simple, basic prompts without elaborate prompt engineering for all experiments with open-source models in Section 3. The main reason is that we want to avoid potential biases introduced by complex prompt engineering, thereby emphasizing the evaluation of the method itself.

### H.1.1  CHOICE FUNNEL PROMPT

This prompt is used for *Choice Funnel* as well as two baseline methods: *first token* and *first token debiasing*

---
**Open Source Prompts**

You are presented with the following:
Paragraph: *paragraph*
Question: *question*
Choices:
*Option A*
*Option B*
*Option C*
*Option D*
*Option E*
Task:
Identify and select all the correct answers based on the paragraph and the question.
Answers:

---

### H.1.2  YES/NO FOR OPEN-SOURCED MODELS

This prompt is used for *yes/no* baseline method to compare against *Choice Funnel*.

---

**Yes/No Prompts**

You are presented with the following:
Paragraph: *paragraph*
Question: *question*
Statement: *Option A |B |C |D |E*
Task:
Determine if the statement answers the question correctly and reply with "Yes" or "No" only.
Answer:

---

## H.2    PROMPTS FOR PROPRIETRY MODEL

### H.2.1    PROBLEMS FOR CURRENT MCQ PROMPTS

Existing benchmarks (Intelligence, 2024) use the following prompts for MCQ questions and then use exact match to get the correct option.

---

**Example MCQ Prompt 1**

What is the correct answer to this question: *question*
Choices: *choices*.
Let's think step by step:
Based on the above, what is the single, most likely answer choice?
Answer in the format:
correct answer is (insert answer here).

---

**Example MCQ Prompt 2**

Given the following question and four candidate answers (A, B, C, and D), choose the best answer.
Question: *question*
Your response should end with:
The best answer is [the_answer_letter]
where the [the_answer_letter] is one of A, B, C, or D.

---

However, it does not work well when there are multiple answers as LLMs are likely to repeat the content of the option before output the second choice. As an example, the below answer is A. Wheat, C. Grain instead of AC, which makes it hard to parse.

---

**Example MCQ Prompt Example 1**

What is the correct answer to this question: Bangladesh passed on its weekend tender for 100,000 tones of optional origin soft wheat, private export sources said. What topics are related to the document above?

Choices: A. Wheat, B. Barley, C. Grain, D. Retail, E. Money-supply, F. Sugar]. Let's think step by step:

Based on the above, what are all most likely answer choice?

Answer in the format:

correct answer is (insert answer here).

Answer (from GPT4o): Let's analyze step by step:

The document mentions wheat: The phrase "100,000 tonnes of optional origin soft wheat" directly refers to wheat.

Relevant choice: A. wheat. The document does not mention barley: There is no reference to barley anywhere in the text.

Not a relevant choice: B. barley. Grain is a broader category that includes wheat: Since wheat is a type of grain, it makes sense to include grain as a related topic.

Relevant choice: C. grain. The document does not mention retail: The context of the tender does not relate to retail or consumer sales.

Not a relevant choice: D. retail. The document does not mention money supply: There is no financial or monetary context provided.

Not a relevant choice: E. money-supply. The document does not mention sugar: There is no reference to sugar in the text.

Not a relevant choice: F. sugar. Final Answer:

Correct answer is **A. Wheat, C. Grain**.

---

**Example MCQ Prompt Example 2**

What is the correct answer to this question: Bangladesh passed on its weekend tender for 100,000 tones of optional origin soft wheat, private export sources said. What topics are related to the document above? Choices: A. Wheat, B. Barley, C. Grain, D. Retail, E. Money-supply, F. Sugar. Let's think step by step:

Based on the above, what are most likely answer choices?

Answer in the format:

correct answer is (insert answer here).

Answer (from GPT4o): Let's analyze the question step by step:

Key Subject of the Document: The document discusses Bangladesh passing on its tender for 100,000 tonnes of optional origin soft wheat.

Relevant Topics:

The keyword wheat clearly relates to the document. Wheat falls under the broader category of grain, making it a relevant choice. Other terms like barley, retail, money-supply, and sugar are not directly related to the subject of the document. Most Likely Answer Choices:

A. Wheat: Directly mentioned in the document. C. Grain: Relevant as wheat is a type of grain. Correct Answer:

The correct answer is **A Wheat, C Grain**.

---

We can observe that parsing the above prompts is much harder because we do not know when they will output the option ID plus answers, and we cannot blindly extract all matched option IDs. Following (Intelligence, 2024) page 38, we used the JSON format to extract answers. Closed-source LLMs have shown good instruction following capability for JSON format output (Zhou et al., 2023). Also, python has many packages that can do fuzzy JSON matching, which can be used to handle edge cases. We extract as many JSON files as possible from each output and combine the extracted answers together as the final answer to the question.

**SATA Prompt**

Given the following question where there is more than one correct answer, choose all correct answers.
Question: *question*
Choices: *choices*

Please select all choices that apply.
You must focus on the question and select all choices that apply.
Let's think step by step:
You must present your selected option IDs in the following JSON format:
$\{"choices" :< A|B|C|D|E|F|G|H|I|J|K|L|M|N|O >\}$

**SATA Prompt Example**

Given the following question where there is more than one correct answer, choose all correct answers.
Question: What is the correct answer to this question: Bangladesh passed on its weekend tender for 100,000 tones of optional origin soft wheat, private export sources said. What topics are related to the document above?Choices: A. Wheat, B. Barley, C. Grain, D. Retail, E. Money-supply, F. Sugar. Please select all choices that apply. You must focus on the question and select all choices that apply. Let's think step by step: You must present your selected option IDs in the following JSON format: $\{"choices" :< A|B|C|D|E|F|G|H|I|J|K|L|M|N|O >\}$
Answer (from GPT4o): Let's analyze the question step by step:
Understanding the question context:
The document discusses Bangladesh's weekend tender for 100,000 tonnes of "optional origin soft wheat." Analyzing the options provided:

A. Wheat: The document specifically mentions "soft wheat." This choice is correct.
B. Barley: There is no mention of barley in the document. This choice is incorrect.
C. Grain: Wheat is a type of grain. This choice is correct as the broader category includes wheat.
D. Retail: The document does not mention retail activities. This choice is incorrect.
E. Money-supply: There is no discussion of money supply or related economic terms in the document. This choice is incorrect.
F. Sugar: The document does not mention sugar. This choice is incorrect.
Conclusion: Based on the analysis, the related topics are Wheat (A) and Grain (C).

Final Answer in JSON format:
**{ "choices": ["A", "C"] }**

One can observe that our proposed prompts can easily extract the answer because they contain only the option IDs.

## H.3 ABLATION PROMPTS

### H.3.1 FEW SHOT PROMPT

We report few few-shot prompt where the number of examples is equal to 5.

---
**Few Shots Prompt**

Given the following question and four candidate answers (A, B, C, and D), choose the best answer.
Question 1: *question 1*
Option 1: *option 1*
Answer 1:*correct option json 1*
Question 2: *question 2*
Option 2: *option 2*
Answer 2: *corect option json2*
...
Question 5: *question 5*
Option 5: *option 5*
Answer 5:*correct option json 5*
Question: *question*
Option: *option*
Please select all choices that apply. You must focus on the question and select all choices that apply. Let's think step by step: You must present your selected option IDs in the following JSON format: $\{"choices" :< A|B|C|D|E|F|G|H|I|J|K|L|M|N|O >\}$

---

## H.4  THINK OPTION BY OPTION PROMPT

Inspired by Smyth et al. (2006); Pew Research Center (2019), we instruct LLM to understand each options and analyze each answer independently.

---
**Choice-by-choice Prompt**

Given the following question and four candidate answers (A, B, C, and D), choose the best answer.
Question: *question*
Option: *option*
Let's think through this step by step:
1. First, let's understand what the question is asking...
2. Now, let's evaluate each option individually...
3. Therefore, the correct answers are...
You must present your selected option IDs in the following JSON format:
$\{"choices" :< A|B|C|D|E|F|G|H|I|J|K|L|M|N|O >\}$

---

## H.4.1  FEW SHOT OPTION PROMPT

We further provide a few examples to teach LLMs how to think option by option, but it still does not improve the performance.

**Few Shots Option Prompt**

Given the following question and four candidate answers (A, B, C, and D), choose the best answer.
Question 1: *question 1*
Option 1: *option 1*
Choice by choice reasoning 1: *reason 1*
Answer 1:*correct option json 1*
Question 2: *question 2*
Option 2: *option 2*
Choice by choice reasoning 2: *reason 2*
Answer 2: *corect option json2*
...
Question 5: *question 5*
Option 5: *option 5*
Choice by choice reasoning 5: *reason 5*
Answer 5:*correct option json 5*
Question: *question*
Option: *option*
Let's think through this step by step:
1. First, let's understand what the question is asking...
2. Now, let's evaluate each option individually...
3. Therefore, the correct answers are...
You must present your selected option IDs in the following JSON format:
$\{"choices": < A|B|C|D|E|F|G|H|I|J|K|L|M|N|O >\}$

### H.4.2   PROMPT WITH AVERAGE OPTIONS COUNT

**SATA Prompt**

Given the following question where there is more than one correct answer, choose all correct answers.
Question: *question*
Choices: *choices*
Please select all choices that apply. You must focus on the question and select all choices that apply. The number of average selected options is 3.63. Let's think step by step: You must present your selected option IDs in the following JSON format:
$\{"choices": < A|B|C|D|E|F|G|H|I|J|K|L|M|N|O >\}$

### H.4.3   PROMPT WITH CORRECT NUMBER OF OPTIONS

**SATA Prompt**

Given the following question where there is more than one correct answer, choose all correct answers.
Question: *question*
Choices: *choices*
Please select all choices that apply. You must focus on the question and select all choices that apply. The number of average selected options is XX. Let's think step by step: You must present your selected option IDs in the following JSON format:
$\{"choices": < A|B|C|D|E|F|G|H|I|J|K|L|M|N|O >\}$

### H.4.4   SINGLE CHOICE PROMPT

To ensure consistency, we use a similar prompt for single choice. We use the same method to retrieve the correct choices. If there is more than one correct choice, we randomly sample from among them.

> **Single Choice Prompt**
>
> Given the following question where there is only one correct answers, choose the correct answer.
> Question: *question*
> Choices: *choices*
> Please the correct choice that apply.
> Let's think step by step: You must present your selected option IDs in the following JSON format: $\{"choice" :< A|B|C|D|E|F|G|H|I|J|K|L|M|N|O >\}$

## H.5 Prompt with Numeric Option

For numeric options, it is hard to retrieve since the number of options can be above 10, and the previous retrieving method could retrieve 12 as 1 and 2. We instruct LLMs to produce correct answers in ascending order. We start by retrieving a larger number that is above 10. For each successful retrieval, remove that number from the output. This way, we can avoid the above scenario.

> **Numeric Prompt**
>
> Given the following question where there is more than one correct answer, choose all correct answers.
> Question: *question*
> Choices: *choices*
> Please select all choices that apply. You must focus on the question and select all choices that apply. You must present your answers in ascending orders. Let's think step by step: You must present your selected option IDs in the following JSON format: $\{"choices" :< 1|2|3|4|5|6|7|8|9|10|11|12|13|14|15 >\}$

## H.6 Prompt with small alphabet Option

> **Small Alphabet Prompt**
>
> Given the following question where there is more than one correct answer, choose all correct answers.
> Question: *question*
> Choices: *choices*
> Please select all choices that apply. You must focus on the question and select all choices that apply. Let's think step by step: You must present your selected option IDs in the following JSON format: $\{"choices" :< a|b|c|d|e|f|g|h|i|j|k|l|m|n|o >\}$

# I Inference Error Handling

For 2.897% of all cases, we cannot find any match in JSON format, so we use Claude 3 Haiku to extract the final labels. To be specific, we adopt the following system prompt:



**Edge Case Handling Prompt**

Given the following text, please identify **all** valid choices. A valid choice is any single letter from A to Q, which might appear right after a colon (e.g., `choices:` `"B"`).

- If one or more valid choices are found, concatenate them and return them in the format `<answer></answer>`: For example, `<answer>BEM</answer>`

- If no valid choices are found, return `<answer></answer>`.

String to analyze: `<output>`

Please provide your answer only in the form below:
`<answer>`



For all cases below, our Claude 3 haiku is able to accurately produce the correct outcome.

Table 10: Comparison of raw LLM outputs and the extracted labeled results obtained using Claude 3 Haiku.

| LLM Output | Claude 3 Haiku Extraction |
|---|---|
| I can't fulfill that request. | NaN |
| `"choices": { "choice": "B" }` | B |
| `{{ "choice": <B \| E \| H \| J \| L`  `\| M \| O> }} }}`"'json | BEHJLMO |
| `{ "choice": [] }` "' | NaN |

We then use Amazon Groundtruth labeling to check whether Claude 3 Haiku correctly parses the answer. Of those, only 47 cases were labeled as No or Yes with confidence lower than 0.6. We manually investigated those 47 cases and found that only four were actually incorrect.

Table 11: Examples of LLM outputs and corresponding extraction results where Claude 3 Haiku produced incorrect extractions.

| LLM Output | Claude 3 Haiku Extraction | Human Corrected Answers |
|---|---|---|
| Let's analyze the text and MeSH categories step by step:...: your selected option IDs - C (Organisms), your selected option IDs - E (Phenomena and Processes), your selected option IDs - G (Chemicals and Drugs) | CE | CEG |
| `{{ "choice": <D \| E \| K \| L \| M> }} }}`"'json | DELM | DEKLM |
| `{ "choice": "choice": "N"oneyour se-lected option IDs }` "' | N | NaN |
| Let's analyze the document step by step: ... your selected option IDs your selected option IDs. Based on this analysis, the applicable choices are A, B, C, and E. | ABC | ABCE |

## J   MORE DETAILS ON KEY OBSERVATIONS

**Unselection Bias.** FP/FN means False Positive Count divided by False Negative Count. If a model has 100 False Negative cases of A, it means that the model has not predicted A in 100 cases where it should have predicted A. If a model has 20 False Positive cases of A, it means that the model has predicted A in 20 cases where it should not have. The low FP/FN rate means that out of all cases, the model tends not to predict A instead of overpredicting A. Due to Count Bias, most of the models have FP/FN rate below 1. However, almost all models has one label with an extremely low FP/FN

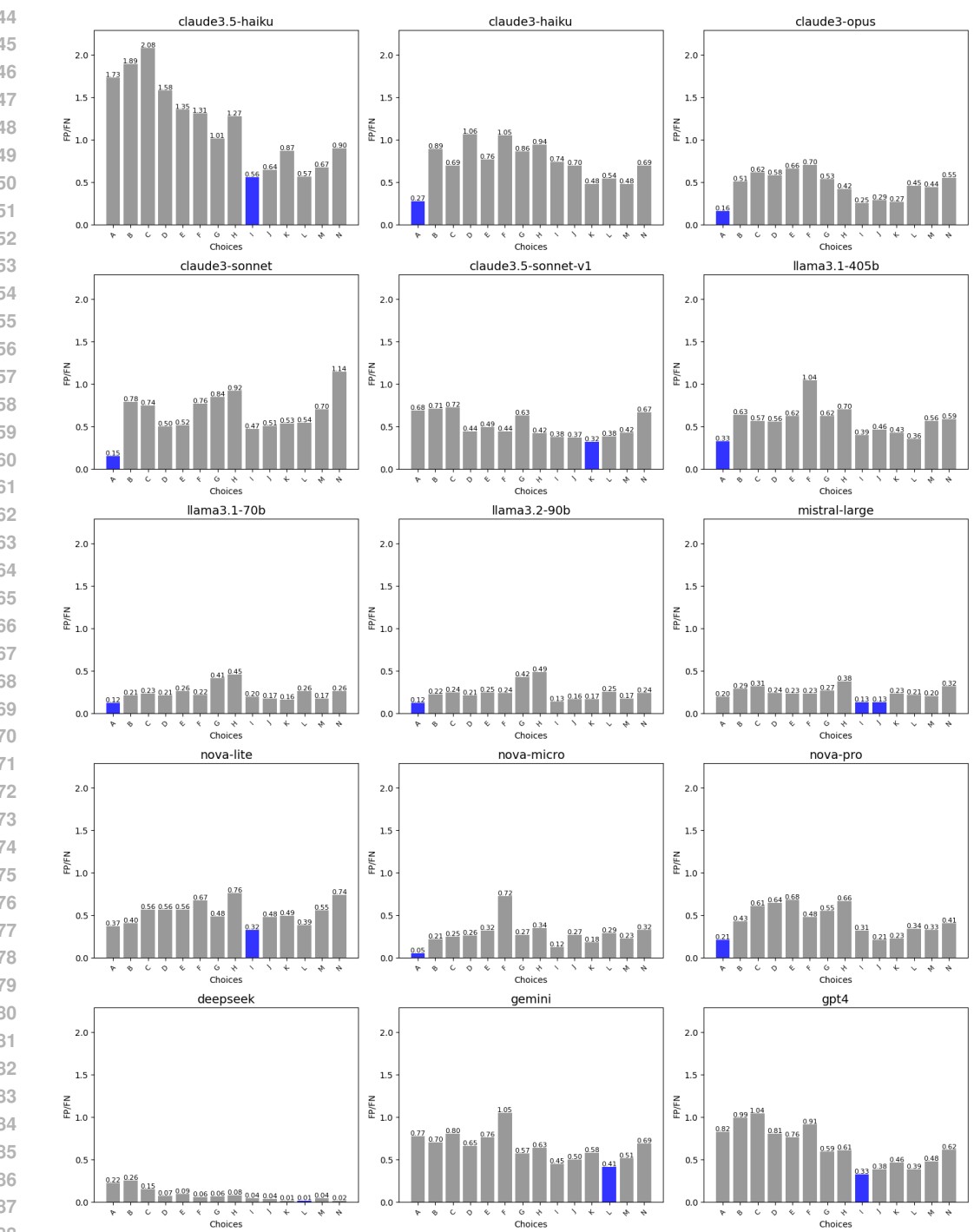

Figure 9: Ratio of false positive rate to false negative rate per label for each evaluated LLM.

rate. For example, Claude3-Haiku has a label A FP/FN rate equal to 0.27 while its second worst is 0.48 as shown in Figure 10.

Recall Difference is another metric to demonstrate unselection bias. Low recall on certain label means that LLMs' incapability of predicting certain labels correctly. As shown in Figure 9, there are many models whose worst label is more than 5% below their average performance.

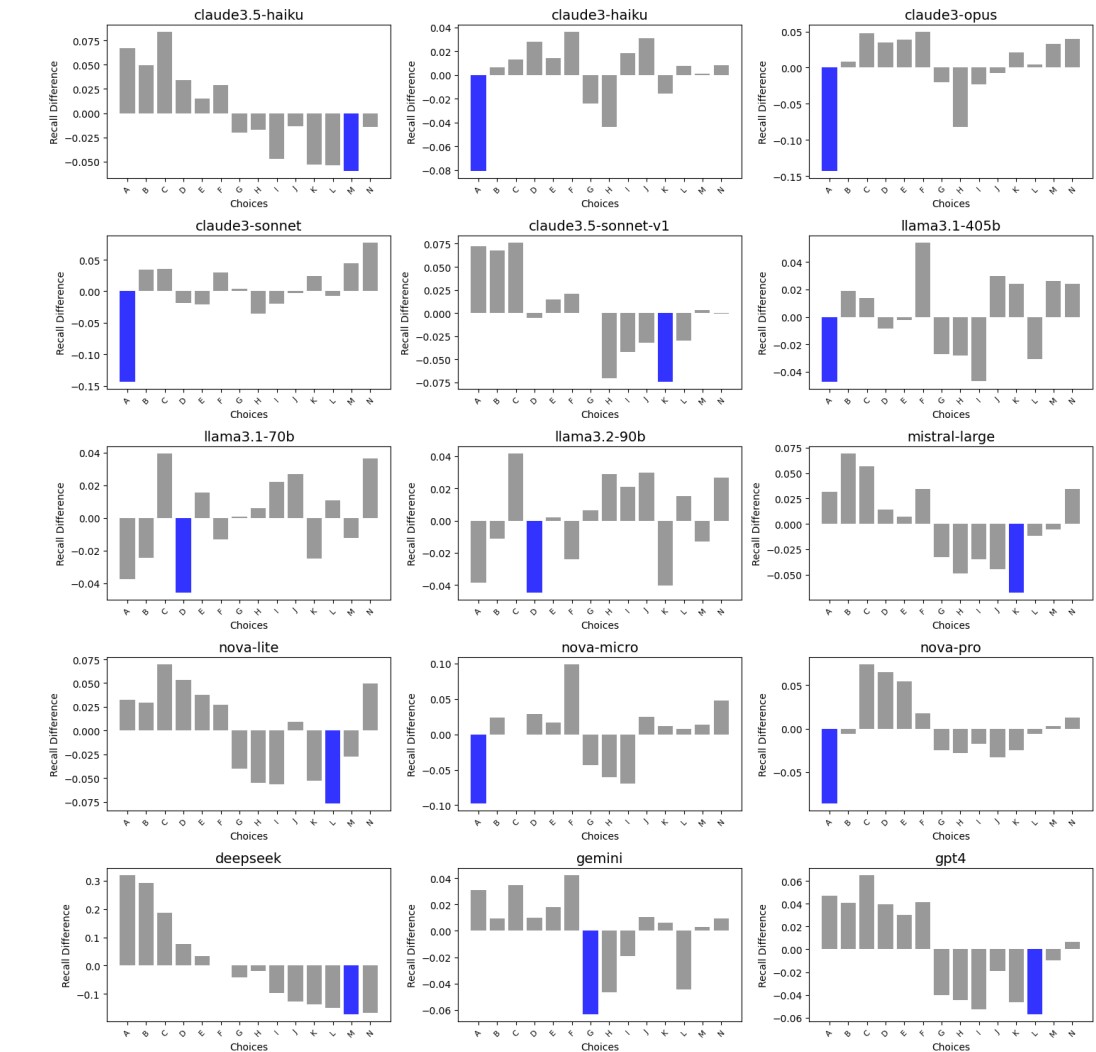

Figure 10: Recall score per label (Y-axis), normalized by subtracting the model's average recall. Most models exhibit at least one label with significantly lower recall than the rest.

**Count Bias.** Figure 11 shows that nearly all models select too few responses and that this tendency increases as the number of correct answers increases. Figure 12 shows that EM also decreases as the number of correct answers increases. This shows that LLMs tend to underpredict the number of correct choices.

## K   PRIDE DEBIASING ALGORITHM ADAPTATION FOR SATA

### K.1   PRIDE INTRODUCTION

The original PriDe algorithm (Zheng et al., 2024a) is designed for processing MCQ question sets with fixed option set length (usually 4). It works by observing the probability changes when performing permutations of option IDs for each question, and it can compute *priors*, which is known as the probabilistic mass that the model a priori assigns to option ID tokens.

Here is an example to better illustrate the process:
Given a question set with 4 options, we compute the prior of each question from 10% of the data, take the average on each option ID position and then we get:
$$P(prior) = [0.4, 0.2, 0.2, 0.2]$$

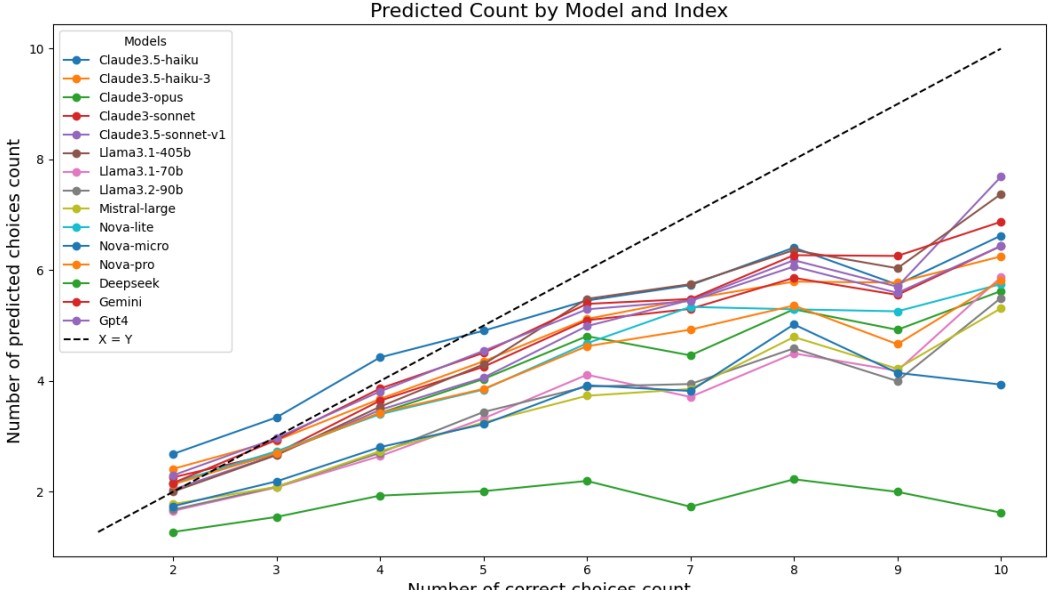

Figure 11: Relationship between predicted and actual correct choice counts across models. Models generally under-select the correct number of answer choices. Y-axis represents the average number of choices selected by the model. X-axis represents the actual number of correct choices. A perfect model would align along the diagonal where X equals Y.

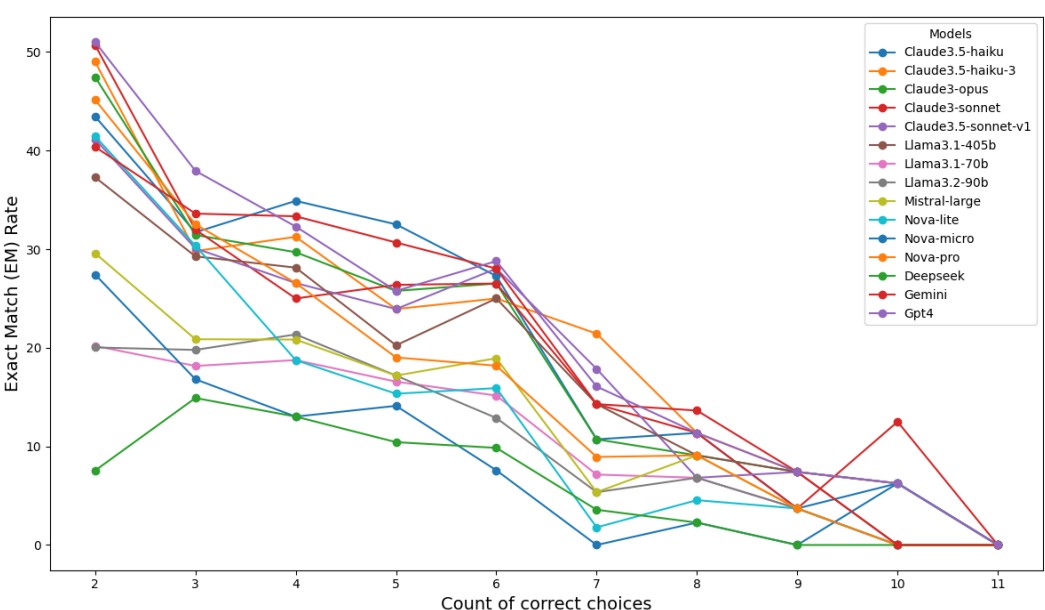

Figure 12: Relationship between Exact Match Rate and the number of correct choices. As the number of correct choices increases, the exact match rate decreases. None of the models achieve an exact match rate above 20% when the number of correct choices exceeds 7.

The list corresponds to probabilities for ABCD. In this case we can see that the model biases towards option "A". Now given a new question with probabilities computed as:

$$P(observed) = [0.5, 0.3, 0.1, 0.1]$$

Without debiasing model will select option "A" as top answer. We need to subtract prior:

$$P(debiased) = P(observed) / P(prior)$$

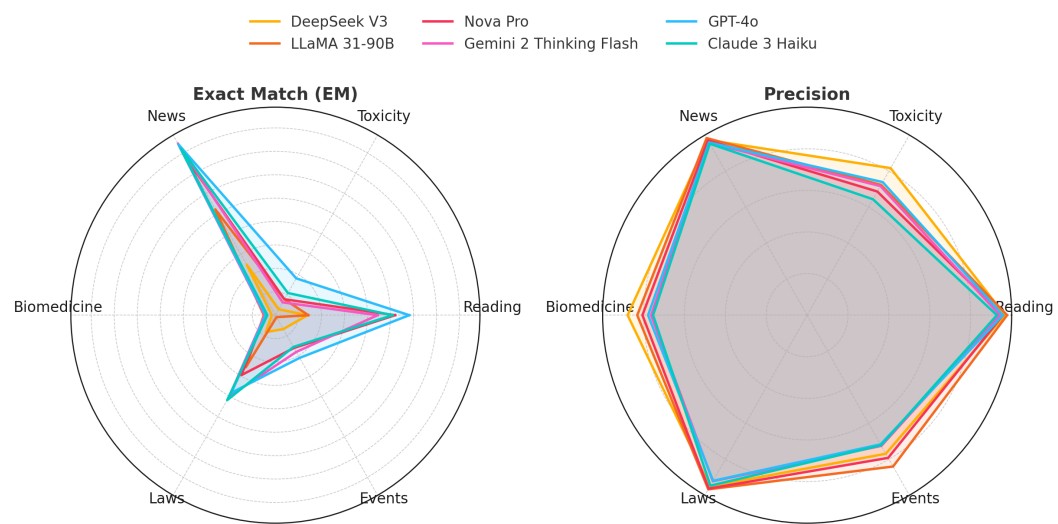

Figure 13: Performance breakdown of evaluated models across different source datasets.

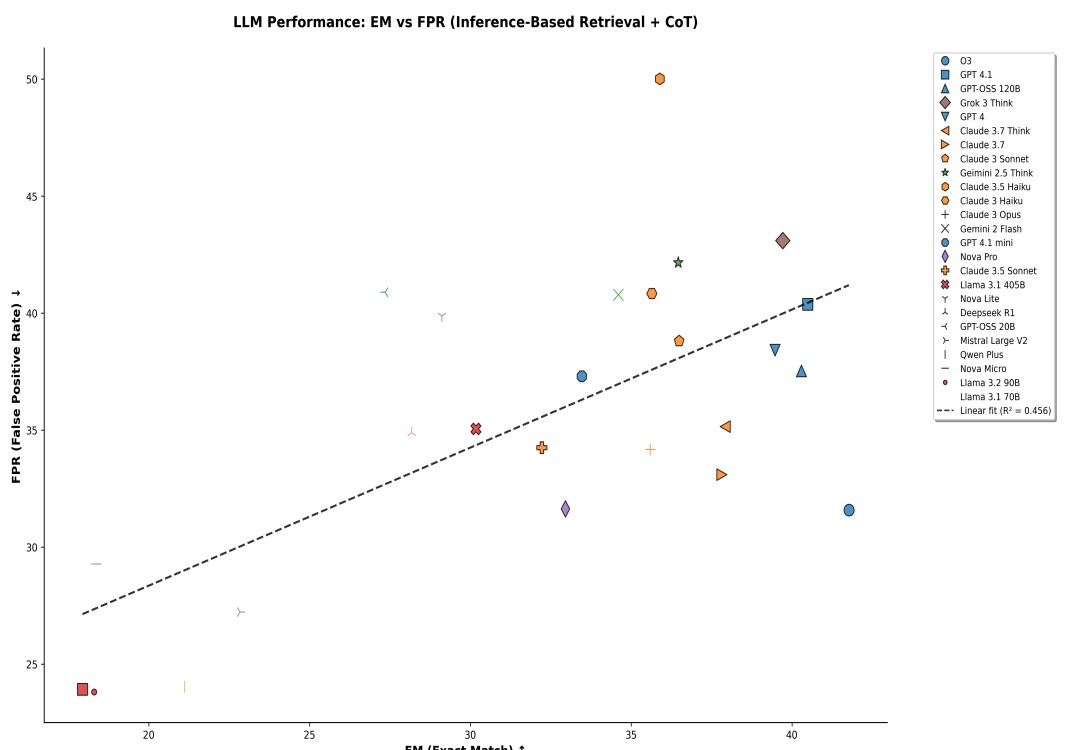

Figure 14: FPR and exact match are positively correlated (r = 0.61, p = 8 × 104, DoF = 23, Two tailed).

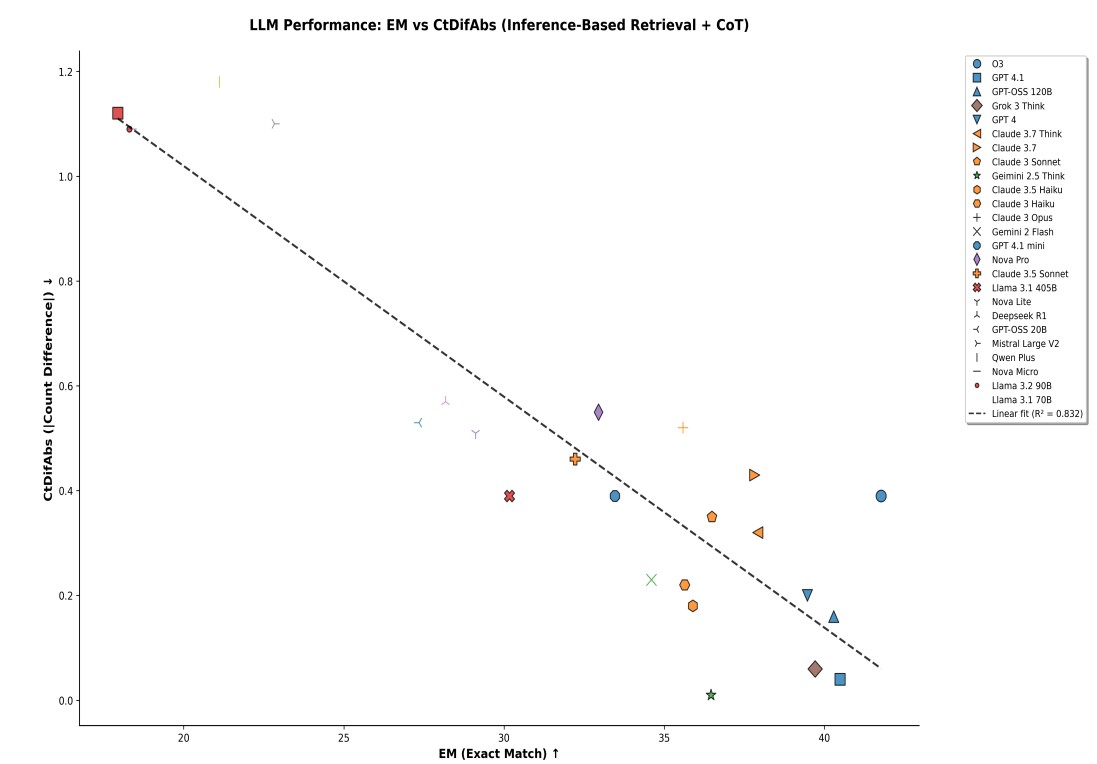

Figure 15: Stronger models Lower CtDifAbs.

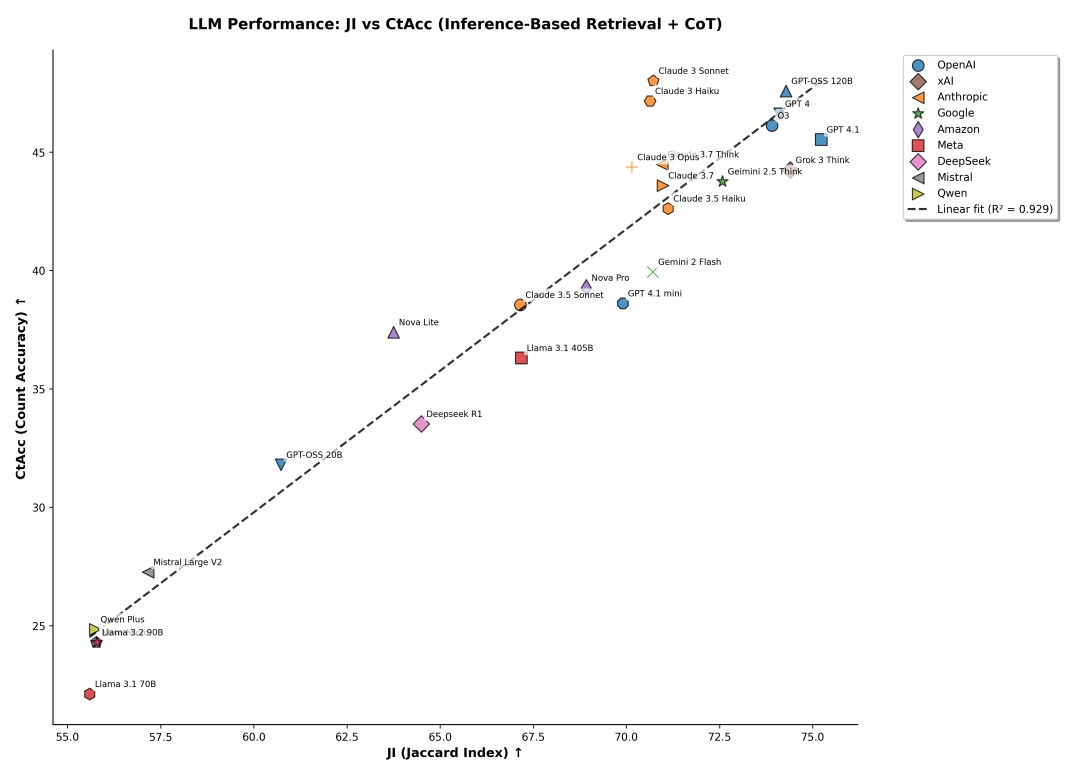

Figure 16: Stronger models better CtAcc.

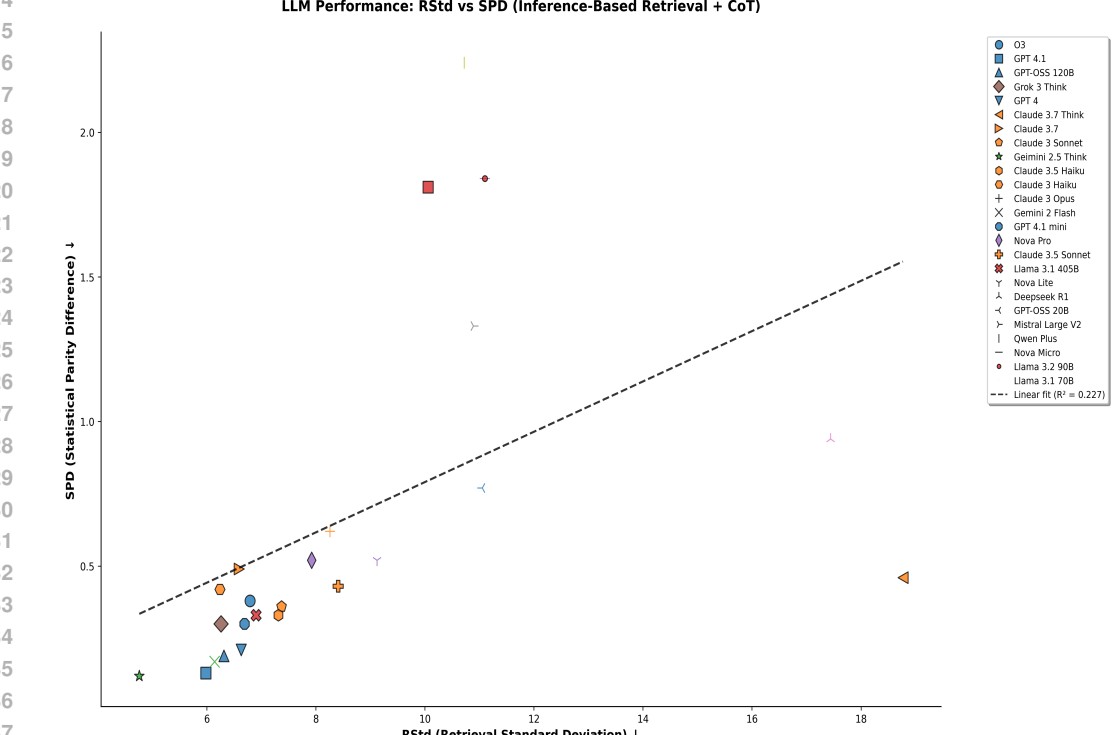

Figure 17: Geimini 2.5 Think has the lowest (un)selection bias compared to other models.

$$P(debiased) = [1.25, 1.5, 1.0, 0.5]$$

Option "B" becomes top-1 after we remove the heavy prior on "A". To learn more low-level details, please refer to the original paper (Zheng et al., 2024a)).

### K.2 LIMITATION OF ORIGINAL ALGORITHM.

However, the prior is computed on a fixed length of 4, so the prior computed for each option has its own probability distribution. For a dataset with variable lengths of option sets (3-15 options for our SATA-Bench). We can only use priors computed for their own length groups (for example, using a length-3 prior to remove bias only for questions that have 3 options). Therefore, we might not have enough data to build an accurate prior. For example, SATA-BENCH contains only 52 out of 1650 questions with 3 choices.

**Adaptation to solve SATA questions.** To solve the above problem, we first construct a dictionary with key as the lengths seen in the dataset, and value as prior computed only from questions with corresponding length, for example:

$$3: [0.5, 0.3, 0.2],$$
$$4: [0.4, 0.3, 0.1, 0.2],$$
$$N: [0.2, 0.1, 0.1, 0.04, 0.04, 0.01, ...]$$

To supplement the lengths with lower datapoint, we take prefix of the longer priors, then *normalize* to unit vector, and use as auxiliary datapoints to help computing for shorter priors, for example a 10-option prior (prior computed from 10-option question) can be used to help computing priors for 3-option question:

$$[0.12, 0.2, 0.05, 0.17, 0.04, 0.01, 0.01, 0.02, 0.3, 0.2]$$

$$\downarrow$$

$$[0.32, 0.54, 0.14]$$

We take the first 3 numbers corresponding to "ABC" of a 3-option question, then normalize it to the unit vector with the same probability distribution as the other 3-option priors. Similarly, this 10-option prior can also be used to compute priors for any shorter lengths.

Lastly, because Choice Funnel will remove the selected option from the option set, the option IDs (ABCD) would not be continuous. Because the prior vector can only work with a continuous option set, we must **rebalance the option IDs**. For example, "ACDE" ("B" is removed) will be rebalanced to "ABCD".

### K.3 Conclusion and Takeaways

Once we have done this process we should have a large enough population to compute accurate priors for most lengths. One limitation is that this adaptation does not help much if we don't have enough questions for longer lengths in our dataset, though this is not the case for SATA-Bench, which contains 21.88% data for its longest 15-option question. One potential solution is to use synthetic datasets to backfill longer-option questions, since the original work showed that the prior is transferable. We leave this for future work.

## L Experiment Setup for Choice Funnel

We chose a fixed *90%* confidence threshold as the stopping condition (ii) in Choice Funnel for **all models**. This initial parameter selection was tuned on 100 hold out data points from raw dataset instead of evaluation set and moves to the closest number that can be divided by 10. It demonstrates that the algorithm is generalizable to other models without careful calibration.

The first baseline method *first token* sets a fixed threshold so that any option with a probability above the threshold is selected, and this should be the lower bound of the performance. *First token debiasing* can be used to find out if the popular strategy used to solve the MCQ questions is transferable to the SATA questions in terms of minimizing the impact of the selection bias. Lastly, we expect *yes/no* to be a competitive baseline given that it processes each choice separately with cost of increased inference compute.

**Prompts.** To reduce the bias introduced by prompt design and emphasize the impact of the method itself, we choose prompts for all methods with minimal engineering effort and mainly capture the essential components: *paragraph, question and choices*. The complete prompts are given in Appendix H.

**Models.** Our study focuses on the causal, decoder-only LLMs since this architecture has become the dominant choice for modern LLMs. We experiment with *7 LLMs from Table 20 under Probability Based Retrieving* which are all popular open-source models on the HuggingFace website, and we can access their output probabilities: DeepSeek R1 Distilled LLAMA 8B (DeepSeek-AI et al., 2025), Qwen2.5 14B (Yang et al., 2025), Ministral 8B (Team, 2024), Phi 3 7B (Abdin et al., 2024), Phi 4 mini reasoning (Abdin et al., 2025), Bloomz 7B (Muennighoff et al., 2022b), and Llama 3.1 8B (Touvron et al., 2023).

## M Ablation Study for Choice Funnel

### M.1 "I don't know" performs worse than "None of the above"

Table 12: Performance comparison of Choice Funnel using "None of the Above" versus "I don't know" options.

| Method | EM↑ | Precision↑ | Recall↑ | JI↑ | SPD↓ | CtDifAbs↓ | CtAcc↑ | InfCost↓ |
|---|---|---|---|---|---|---|---|---|
| Phi3-7B + *nota* | **29.27** | **83.27** | 70.24 | 61.85 | 3.47 | **1.42** | **0.38** | 6339 |
| Phi3-7B + *idk* | 28.18 | 80.92 | **73.25** | **62.22** | **2.35** | 1.48 | 0.36 | 6667 |
| Llama3-8B + *nota* | **19.88** | **78.69** | 56.19 | **50.36** | **7.74** | **1.66** | **0.33** | **4975** |
| Llama3-8B + *idk* | 17.64 | 75.50 | **58.03** | 49.55 | 7.74 | 1.69 | 0.32 | 5066 |
| Bloomz-7B + *nota* | **20.18** | **66.62** | 54.90 | **46.15** | 17.78 | **1.71** | **0.32** | **5440** |
| Bloomz-7B + *idk* | 18.00 | 65.55 | **55.76** | 45.53 | **16.45** | 1.76 | 0.31 | 5528 |

We compared two commonly employed auxiliary response options in traditional survey science domain (Schuman & Presser, 1996): 'I don't know' (*IDK*) and 'None of the above' (*NOTA*), examining their effectiveness as *Choice Funnel* stopping condition. Based on an ablation study on Table 14, *NOTA* yields consistently better performance. When using *IDK*, we observe **noticeable increase in InfCost and result in worse Count Bias** (*CtDifAbs* and *CtAcc*), which means **model tends to over select number of options**, indicating that the model would rather select a wrong answer than saying "I don't know". This is potentially related to RLHF process, where the model is trained to generate answers that are more favorable to humans.

## M.2 Ablation on Choice Funnel Components

Table 13: Ablation study demonstrating that PriDe token debiasing effectively mitigates unselection bias.

| Method | EM↑ | Precision↑ | Recall↑ | JI↑ | SPD↓ | CtDifAbs↓ | CtAcc↑ | InfCost↓ |
|---|---|---|---|---|---|---|---|---|
| Phi3-7B + *debiasing only* | 1.76 | 67.92 | 28.24 | 27.47 | 175.24 | 2.50 | 0.05 | **2534** |
| Phi3-7B + *CF only* | 26.00 | 80.84 | 70.08 | 60.33 | 4.17 | 1.44 | 0.35 | 6436 |
| Phi3-7B + *CF + debiasing* | **29.27** | **83.27** | **70.24** | **61.85** | 3.47 | **1.42** | **0.38** | 6339 |
| Llama3-8B + *debiasing only* | 7.58 | 62.83 | 32.28 | 30.38 | 151.74 | 2.34 | 0.14 | **2534** |
| Llama3-8B + *CF only* | 17.45 | 76.37 | 50.84 | 46.74 | 10.12 | 1.67 | **0.34** | 4380 |
| Llama3-8B + *CF + debiasing* | **19.88** | **78.69** | **56.19** | **50.36** | 7.74 | 1.66 | 0.33 | 4975 |
| Bloomz-7B + *debiasing only* | 7.09 | 59.07 | 38.41 | 32.05 | 149.17 | 2.19 | 0.15 | **2534** |
| Bloomz-7B + *CF only* | 16.36 | 66.10 | 48.26 | 42.66 | 23.09 | **1.65** | **0.35** | 4469 |
| Bloomz-7B + *CF + debiasing* | **20.18** | **66.62** | **54.90** | **46.15** | 17.78 | 1.71 | 0.32 | 5440 |

The *CF only* setting represents scenarios where the model has no access to raw probabilities and instead relies solely on the Choice Funnel algorithm (Black-box settings). Compared to token debiasing, this approach achieves significant improvements in EM and Precision. On average, across three models—even without using token probabilities —Choice Funnel yields a 10.79% increase in Exact Match, a 20.51% increase in Jaccard Index, a 13.4 reduction in SPD, and a 0.86 reduction in CtAbsDif.

We conducted an ablation study on the two sub-components of Choice Funnel: token debiasing (*"debiasing only"*) and iterative selection (the process of iteratively selecting options until a stopping condition is met, denoted as *"CF only"*). The analysis is performed on 3 open-source models.

When comparing *"CF only"* to the complete *"CF + debiasing"*, the observed increase in SPD metric demonstrates that **token debiasing effectively mitigates unselection bias**, yielding better performance. Nevertheless, the comparison between *"debiasing only"* and *"CF only"* reveals that **our novel iterative selection component contributes more substantially to overall performance improvements.**

## M.3 Ablation on Choice Funnel Stopping Condition

Table 14: Ablation study on the two stopping conditions in Choice Funnel, showing that combining both yields the best performance.

| Method | EM↑ | Precision↑ | Recall↑ | JI↑ | SPD↓ | CtDifAbs↓ | CtAcc↑ | InfCost↓ |
|---|---|---|---|---|---|---|---|---|
| Phi3-7B + *thresholding only* | 3.82 | 65.00 | 74.84 | 48.93 | 3.37 | 2.22 | 0.13 | 7416 |
| Phi3-7B + *NOTA only* | 29.21 | 77.07 | **85.63** | **68.00** | **0.69** | **1.20** | 0.37 | 9380 |
| Phi3-7B + *thresholding + NOTA* | **29.27** | **83.27** | 70.24 | 61.85 | 3.47 | 1.42 | **0.38** | 6339 |
| Llama3-8B + *thresholding only* | 0.89 | 71.92 | 52.22 | 44.12 | 10.53 | 1.74 | 0.27 | 4564 |
| Llama3-8B + *NOTA only* | 19.51 | 69.22 | **85.77** | **60.09** | **2.24** | 1.94 | 0.25 | 10212 |
| Llama3-8B + *thresholding + NOTA* | **19.88** | **78.69** | 56.19 | 50.36 | 7.74 | **1.66** | **0.33** | 4975 |
| Bloomz-7B + *thresholding only* | 9.94 | 64.47 | 48.93 | 40.77 | 22.50 | 1.72 | 0.29 | 4506 |
| Bloomz-7B + *NOTA only* | 12.24 | 55.60 | **89.57** | **52.81** | **12.82** | 3.31 | 0.17 | 13758 |
| Bloomz-7B + *thresholding + NOTA* | **20.18** | **66.62** | 54.90 | 46.15 | 17.78 | **1.71** | **0.32** | 5440 |

We conducted an ablation study to evaluate the relative importance of our two proposed stopping conditions in Choice Funnel. The results demonstrate that Choice Funnel achieves optimal performance when both conditions are applied in combination. Notably, the "None of the above" (*NOTA*) condition emerged as the more influential factor, suggesting that models can reliably identify when no correct answers remain among the provided options.

## M.4 Scalability of Choice Funnel on Larger LLM

Table 15: Scalability demonstration with larger LLAMA3.1-70B model, showing that ChoiceFunnel improves performance across different model sizes.

| Model | EM↑ | Recall↑ | SPD↓ | CtAcc↑ |
|---|---|---|---|---|
| LLAMA3.1-70B + *prompting* | 17.94 | 60.64 | 1.81 | 0.22 |
| LLAMA3.1-7B + *ChoiceFunnel* | 19.88 | 56.19 | 7.75 | 0.33 |
| LLAMA3.1-70B + *ChoiceFunnel* | **24.43** | **68.66** | **0.37** | **0.37** |

These results show that Choice Funnel scales well with model size, and consistently outperforms prompting-only approaches while maintaining high efficiency.

## M.5 Performance on Single-Answer Questions

Although we did not include single-answer questions in our main evaluation because we specifically focus on model behavior when multiple answer paths exist, our larger 10k dataset includes approximately 750 single-answer samples. We run comparisons against the original 2+ answer dataset to evaluate ChoiceFunnel's robustness across different question types.

Table 16: Comparison of ChoiceFunnel performance on single-answer vs. multi-answer questions, demonstrating robustness across different question types.

| Model | Dataset | EM↑ | JI↑ | Precision↑ |
|---|---|---|---|---|
| Llama3-8B | Original (2+ answers) | 19.88 | 50.36 | 78.69 |
| Llama3-8B | Single answer | **55.04** | **73.18** | 70.05 |
| Mistral-8B | Original (2+ answers) | 20.24 | 52.56 | 86.03 |
| Mistral-8B | Single answer | **60.27** | **76.82** | 73.31 |
| Qwen2.5-14B | Original (2+ answers) | 27.82 | 61.12 | 85.69 |
| Qwen2.5-14B | Single answer | **63.33** | **79.56** | 76.17 |

ChoiceFunnel demonstrates improved EM and JI on single-answer questions due to reduced interference from multiple correct answers. The slight reduction in Precision compared to the original multi-answer setting likely stems from threshold miscalibration for single-answer scenarios and/or the inherently challenging distractor choices in our SATA dataset design.

## N Benchmark on System Prompt

To ensure consistency with other benchmarks, our evaluation code is following the structure in openai codebase simple-eval, where we mentioned "You are a helpful assistant. Each question below contains at least two correct answers" plus model specific system prompt. We have compared it with the system prompt "You are a helpful assistant. Please pick any candidate answer that is correct". We report 4 different models'difference in performance across 400 sata questions from evaluation set and report the difference (original system prompt performance - your suggested system prompt's performance.). All difference in all performance metrics' is less than $1.6\%$, while the average differences between 3 metrics are less than $0.2\%$. This shows that the evaluation result is not sensitive to system prompt. We suspect this is due to all questions being unambiguous as mentioned before.

| Models | EM Dif | Precision Dif | FPR Dif |
|---|---|---|---|
| GPT-OSS 20B | 0.2% | 0.9% | -1.1% |
| GPT-OSS 120B | 1.6% | 0.3% | -0.3% |
| Claude 3.5 Sonnet | -0.5% | -0.6% | 0.3% |
| QWen Plus | -0.5% | -0.4% | 0.5% |
| Average | 0.2% | 0.07% | 0.18% |

## O Positional Bias Under Randomized Answer Orderings

**Does the benchmark include randomized answer orderings?** No. In the main benchmark, each question's answer choices appear in a fixed, canonical order. To quantify the extent to which large

language models (LLMs) rely on this implicit positional cue, we ran an auxiliary study in which the answer choices for every question were *randomly permuted* (e.g. A B C → C A B). We then compared model performance on the permuted dataset to its performance on the original version.

**Setup.** All hyper-parameters, prompts, and decoding settings were kept *identical* to the main benchmark; only the answer order was shuffled once per question. Table 17 reports the *difference* (*permute–original*) for each metric, so negative values indicate a drop in performance and positive values indicate an increase. [†] **CtDif** is shown with a downward arrow even though its baseline values are negative; a more negative CtDif therefore indicates a larger absolute mismatch in option counts.

Table 17: Change in evaluation metrics after randomly reordering answer choices. Performance metrics are expected to **increase** (↑) while bias metrics are expected to **decrease** (↓).

| Model | EM ↑ | Precision ↑ | Recall ↑ | JI ↑ | RStd ↓ | RSD ↓ | SPD ↓ | CtDif[†] ↓ | CtDifAbs ↓ |
|---|---|---|---|---|---|---|---|---|---|
| Claude 3 Haiku | −24.06 | −34.69 | −34.28 | −35.31 | +6.06 | +0.17 | +0.12 | −0.07 | −0.51 |
| Llama 3.1 405B | −3.80 | −3.90 | −4.71 | −5.22 | +9.73 | −0.20 | +0.25 | −0.18 | −0.71 |

**Findings.** All three models suffer performance degradation when answer choices are shuffled, with **Claude 3 Haiku** exhibiting the sharpest decline (–24 EM, –35 JI). Selection / count-bias metrics (RStd, SPD, CtDifAbs) *increase* for every model except RSD, confirming heightened positional bias.

**Discussion.** These results suggest that current LLMs implicitly learn positional heuristics from training data in which answer orders are fixed. Breaking this assumption makes the models less certain and more prone to biased guessing. Future work should examine (i) whether fine-tuning on randomly ordered choices mitigates the effect, and (ii) how pronounced the bias is for other model families and task domains.

## P   PER-DATASET PERFORMANCE BREAKDOWN

We report detailed bias metrics for different task categories in Table 18. The News dataset has the lowest selection bias, while Reading Comprehension exhibits the highest. For count bias, Toxicity shows the smallest difference, and Biomedicine has the largest. Notably, News has significantly lower selection and count biases compared to other datasets (p-values: 0.03 for SPD and $3.8 \times 10^{-5}$ CtDifAbs, T-test). All datasets show negative count difference, confirming underprediction and the presence of count bias in SATA questions.

Table 18: Breakdown of Bias metrics by subject. Lower values are better for all metrics.

| Task | RStd ↓ | RSD ↓ | SPD ↓ | CtDif | CtDifAbs ↓ |
|---|---|---|---|---|---|
| Reading Comprehension | 19.29 ± 7.59 | 0.20 ± 0.10 | 1.53 ± 1.39 | -0.68 ± 0.42 | 0.85 ± 0.35 |
| Toxicity | 7.13 ± 2.83 | 0.11 ± 0.07 | 0.48 ± 0.56 | -0.05 ± 0.44 | 1.28 ± 0.16 |
| News | 4.32 ± 3.16 | 0.08 ± 0.19 | 0.12 ± 0.23 | -0.09 ± 0.25 | 0.32 ± 0.19 |
| Biomedicine | 6.66 ± 2.37 | 0.15 ± 0.14 | 2.90 ± 3.60 | -1.71 ± 0.96 | 2.22 ± 0.67 |
| Laws | 5.75 ± 4.17 | 0.13 ± 0.16 | 1.54 ± 3.43 | -1.00 ± 0.87 | 1.36 ± 0.75 |
| Events | 7.15 ± 4.14 | 0.13 ± 0.19 | 0.85 ± 1.02 | -0.28 ± 0.77 | 1.08 ± 0.30 |

## Q   THE CHALLENGE OF MULTI-ANSWER REASONING

### Q.1   PROBLEM SETUP

We formalize SATA questions as a subset prediction task. Given a set of $K$ candidate options $\mathcal{O} = \{o_1, \ldots, o_K\}$ and a ground-truth set $S^\star \subseteq \mathcal{O}$ of correct options, a model must output $\hat{S} \subseteq \mathcal{O}$ that matches $S^\star$. We evaluate with set-based metrics including exact match (EM), Jaccard index (JI), macro precision/recall, and count-based measures (count difference, absolute count difference, and count accuracy; see Appendix F for definitions). Unlike single-choice MCQ (where $|S^\star| = 1$), SATA requires reasoning over both *which* options are correct and *how many* should be selected.

## Q.2 Bias Definitions

Let $y_i^\star \in \{0, 1\}$ denote the ground-truth label for option $o_i$ and $\hat{y}_i \in \{0, 1\}$ the model's selection. Define the random variables $C^\star = \sum_{i=1}^K y_i^\star$ and $\hat{C} = \sum_{i=1}^K \hat{y}_i$ as the true and predicted counts.

**Count Bias.** A model exhibits *count bias* if it systematically under estimates the number of correct options: $\mathbb{E}[\hat{C}] \neq \mathbb{E}[C^\star]$ over the evaluation distribution. Empirically, we find a dominant *under-selection* pattern, $\mathbb{E}[\hat{C}] < \mathbb{E}[C^\star]$ (Sec. 3.1; Figures 12, 11), reflected in low CtAcc and negative mean CtDif.

**Selection Bias.** Let $p_i = \Pr(\hat{y}_i = 1)$ denote the marginal selection probability for option $o_i$ across the benchmark. A model exhibits *selection bias* if the dispersion of $\{p_i\}_{i=1}^K$ is larger than expected from the true label distribution, indicating preference or aversion to certain labels independent of content.[3] We quantify selection skew with RStd/RSD (Zheng et al., 2024a; Croce et al., 2020; Reif & Schwartz, 2024) and introduce *Selection Probability Divergence (SPD)* to capture *unselection* bias (Appendix F); in aggregate, observed SPD significantly exceeds random baselines (Sec. 3.1).

**Speculation Bias.** Define the per-question false-positive count $\text{FP} = \sum_{i=1}^K (1 - y_i^\star)\hat{y}_i$ and the *speculation indicator* $FPR = \mathbb{E}[\text{FP} > 0]$. A model exhibits *speculation bias* if it systematically selects options outside the gold set, especially more than the number of time it produces correct labels, $FPR > EM$. Speculation bias is reflected by higher macro *false-positive rate* and smaller JI (which penalizes any spurious selections). Note that speculation may co-occur with over-selection, but it is distinct: a model can be count-unbiased yet still speculate (high FPR).

## R Does LRM help? A case study of GPT-OSS on SATA-Bench

Reasoning model such as GPT-OSS 120B model performs on par with GPT-4.1 on SATA-Bench. GPT-OSS 20B model is much weaker than 120B but still matches Llama-3.1-405B. Despite good slightly better performance. Reasoning model does exhibit a few failure modes in SATA-Bench.

**Repetitive Reasoning.** We define a reasoning as repetitive if it repeats 100+ characters more than 10 times. This happens in 11% of reasons for 20B model. In those cases where model produce repetitive reasoning, it have much lower EM rate. As an example, GPT-OSS 20B exact match rate drop form 27.4% to 18.5% when it starts to repeat the same characters.

**Reason Answer Mismatch.** The final answer choices do not always align with the reasoning steps. We used Claude 3 Haiku to extract answers. We found that for cases where 120B is correct and 20B is wrong, 53.2% of the answers do not match the reasoning. In 45% of mismatched cases, the reasoning itself was actually correct — the model just picked the wrong a subset of correct choices as final answer. This has increase unselection bias and count bias. We provide the following as an example of GPT-OSS 120B where there is a mismatch

---

[3]Position and formatting effects can contribute; cf. Zheng et al. (2024a).

### Example: Reason–Answer Mismatch Case

**Context.** *Sunlight* Sunlight can be used to heat homes. It can also be used to produce electricity. This conversion is made possible by solar cells. However, solar energy may not always be practical. Some areas are just too cloudy. *Example* Solar panels on the roof of this house generate enough electricity to supply a family's needs. *Moving Water* Falling water can have a lot of energy. Its energy can be converted into kinetic energy. This energy can turn a turbine and generate electricity. The water may fall naturally over a waterfall or flow through a dam. A drawback of dams is that they flood land upstream. They can also reduce water flow downstream. Either effect may harm ecosystems. *Wind* is moving air. It has kinetic energy that can do work. Wind turbines change the kinetic energy of the wind to electrical energy. Only certain areas of the world get enough steady wind. Many people also think that wind turbines are noisy and not very nice to look at.

**Question.** How does moving water make electricity? *(Select all that apply)*

**Options.**

A. Water can generate electricity by turning a turbine.

B. Electricity can be made by storing the water in batteries.

C. Water's energy can turn a turbine and magically result in electricity.

D. Falling water can have a lot of energy which can be converted into kinetic energy that turns a turbine and generates electricity.

E. Water can move a turbine that generates power.

F. Electricity can be made by taking the hydrogen out of moving water.

**Model Reasoning (excerpt).** Focus on the "Moving Water" sentences: (7) Falling water can have a lot of energy; (8) its energy can be converted into kinetic energy; (9) this energy turns a turbine and generates electricity; (10) sources include waterfalls or dams. Matches: **A**, **D**, **E**. Option **C** adds "magically," which is unsupported by the passage.

**Ground Truth Answers. A, D, E**
**Model Prediction. A**
**Reason–Answer Mismatch.** The model's reasoning identifies **A, D, E** as correct, but its final selection includes only **A**. Options **B** and **F** are unsupported; **C** introduces an extraneous claim ("magically") not stated in the text.

**Overthinking is not helping** Long reasoning chains often lead to lower accuracy across GPT-OSS model family. When reasoning token is below 403, GPT OSS 120B achieve 0.65 Exact Match rate. It drops to 0.22 when the number of reasoning token is over 2.8k.

## S   LLM USAGE

We used large language models (e.g., ChatGPT) solely as assistive tools for (i) light editing of grammar and wording and table reformatting, and (ii) debugging code when running experiments (e.g., clarifying error messages, suggesting fixes). LLMs did not write any source code used in our experiments and did not generate substantive paper content beyond minor edits. All ideas, analyses, experimental designs, and final text are the authors' own. The authors reviewed and verified all model-assisted edits and take full responsibility for the contents of this paper.

## T   FINETUNING TO IMPROVE LLM PERFORMANCE ON SATA

**Training Data.** Following (Brahman et al., 2024), we use a general-purpose instruction tuning dataset to balance safety and helpfulness. We select utility data from Tulu-3-SFT-mixture (Lambert et al., 2025), a 940k-instance dataset spanning diverse tasks for training non-reasoning models. We follow a 90:10 utility-to-safety data ratio as in (Zhang et al., 2024). Due to compute limits, we sample 12,000 pairs from Tulu-3, and 1,300 from SATA raw dataset exclude evaluation set.

| Model | Tuning | MMLU-0 | GSM8K | SATA-EM | SATA-JI |
|---|---|---|---|---|---|
| Gemma-2 2B | Tulu-3 (baseline) | 31.1% | 30.5% | 0.8% | 15.7% |
| | Tulu-3 + SATA | 32.5% | 32.0% | 25.8% | 62.3% |
| Llama-3.2 1B | Tulu-3 (baseline) | 30.2% | 23.0% | 0.9% | 26.1% |
| | Tulu-3 + SATA | 26.4% | 23.6% | 29.2% | 60.1% |
| Qwen-2.5 0.5B | Tulu-3 (baseline) | 36.9% | 29.4% | 4.4% | 12.5% |
| | Tulu-3 + SATA | 31.5% | 26.2% | 23.7% | 57.0% |

Table 19: Performance of instruction-tuned models on standard benchmarks and SATA.

**Models.** We train 3 LLMs of smaller sizes, including Llama-3.2-1B, Qwen-2.5-0.5B, and Gemma-2-2B. Following prior work (Brahman et al., 2024), we conduct SFT on the base pretrained models rather than their instruction-tuned variants, to avoid confounding from built-in safety tuning. To assess the impact of our training strategy, we compare models fine-tuned on combined utility and safety data against baselines trained only on utility data.

**Evaluation Setup.** In addition to SATA-Bench, we also evaluate general language abilities and knowledge understanding assessed with the widely-used GSM8K (Cobbe et al., 2021) for grade-level math reasoning and MMLU (Hendrycks et al., 2021) for broader language comprehension.

**Implementation Details.** We use Llama Factory (Zheng et al., 2024b) as the framework for all fine-tuning experiments and perform inference using its vLLM (Kwon et al., 2023) implementation for efficient inferences. Following previous works (Brahman et al., 2024), we adopt standard fine-tuning hyperparameters: training for three epochs with a total batch size of 8. We use bfloat16 precision and a learning rate of $1 \times 10^{-5}$, which is linearly warmed up for the first $10\%$ of training steps and then decayed to zero following a cosine schedule. We use the AdamW optimizer and a standard supervised finetuning loss of next word prediction. We employ a context length of 2048 for model training. Following previous works, during inference we set the temperature 0.7 for general language ability evaluation and a maximum generation length of 1024 tokens for non-reasoning models and 8192 tokens for reasoning models. (Cui et al., 2025) All experiments are conducted on a server with 8 NVIDIA A100 40G GPUs using AWS EC2 instances. During inference, we notice that Qwen-2.5-0.5B can produce repetitive content instead of the answer. For those cases, we detect repetition by checking whether a span of content longer than 4 tokens is repeated more than 20 times, and regenerate when this occurs. All answers are extracted using GPT-OSS-120B with the following prompt.

**Findings.** SFT with SATA data hugely improve small models performance on SATA questions. While still performing worse than closed source model, we found that EM for evaluation set jump at least $18\%$ and JI at least $33\%$. This demonstrates that SATA questions could be easily resolved by adding small portion of SATA questions in the training set.

## Answer Extraction Prompt

**Multiple Choice Answer Extraction Task**

You are presented with the following:

`<question>{{question}}</question>`

`<answer>{{answer}}</answer>`

**Requirements and Expectations** 1. Carefully read and understand both the question and the provided answer text. 2. Identify which multiple-choice options are being selected in the answer (e.g., A, B, C, D, etc.). 3. Your job is **only** to extract the letter combination corresponding to the selected choices (e.g., A, B, AC, BDE). 4. Do **not** interpret or correct the answer content; simply recover the choices that the answer indicates. 5. Your final output must strictly follow the required output format described below.

**Output Format** You must respond using **exactly** the following format:

`<answer>XYZ</answer>`

where:

- `XYZ` is the letter or combination of letters representing the selected choices.

- If multiple letters are selected, they must be concatenated **without** spaces, commas, or other separators (e.g., `BDE`, not `"B, D, E"` or `"B D E"`).

**Important Requirements** 1. Output **only** the answer tags with the letter combination inside (no explanations, reasoning, or extra text). 2. Do not add any spaces, newlines, punctuation, or commentary outside the `<answer>...</answer>` tags. 3. If you cannot confidently identify the selected choice(s), leave the interior empty as `<answer></answer>`.

**Suggestions and Strategies** 1. Look for explicit mentions of answer letters (e.g., "I choose A and C") or indirect descriptions that clearly map to specific options. 2. If the answer restates option texts instead of letters, carefully match those texts back to the corresponding choice letters. 3. Double-check that you have included all and only the choices that the answer selects before producing the final `<answer>...</answer>` output.

Respond with the extracted answer in the specified format. Answer as precisely and accurately as possible.

Table 20: Performance comparison of 32 different LLMs across various metrics on SATA-BENCH. We highlight the best (**bold**) and second-best (underline) values. Columns labeled [(↑)] indicate higher-is-better; columns labeled [(↓)] indicate lower-is-better. Models with explicit reasoning capabilities are highlighted in *italic*. All numeric values are rounded to two decimal places. We retrieve exact labels for models evaluated using Inference-Based Retrieval + CoT prompting. For models evaluated under Probability-Based Retrieval, we select labels based on token probability thresholds.

| Model Name | Performance | | | | Selection Bias | | | Count Bias | | |
|---|---|---|---|---|---|---|---|---|---|---|
| | JI↑ | FPR↓ | EM↑ | Precision↑ | SPD↓ | RStd↓ | RSD↓ | CtDif | CtDifAbs↓ | CtAcc↑ |
| **Inference Based Retrieval + CoT** | | | | | | | | | | |
| *O3* | 73.91 | 31.58 | **41.77** | 87.50 | 0.38 | 6.79 | 0.06 | -0.39 | 0.94 | 46.12 |
| GPT 4.1 | **75.23** | 40.37 | 40.49 | 85.52 | 0.13 | 5.98 | 0.06 | -0.04 | 0.85 | 45.52 |
| *GPT-OSS 120B* | 74.28 | 37.53 | 40.29 | 86.28 | 0.19 | 6.31 | 0.07 | -0.16 | 0.84 | 47.57 |
| *Grok 3 Think* | 74.40 | 43.10 | 39.71 | 83.93 | 0.30 | 6.26 | 0.07 | 0.06 | 0.93 | 44.24 |
| GPT 4 | 74.11 | 38.42 | 39.47 | 85.90 | 0.21 | 6.63 | 0.06 | -0.20 | **0.82** | 46.61 |
| *Claude 3.7 Think* | 70.96 | 35.16 | 37.92 | 85.03 | 0.46 | 18.77 | 0.34 | -0.32 | 0.87 | 44.48 |
| Claude 3.7 | 70.98 | 33.10 | 37.82 | 85.35 | 0.49 | 6.59 | 0.25 | -0.43 | 0.93 | 43.58 |
| Claude 3 Sonnet | 70.72 | 38.81 | 36.49 | 84.58 | 0.36 | 7.37 | 0.07 | -0.35 | 0.83 | **48.00** |
| *Geimini 2.5 Think* | 72.58 | 42.16 | 36.46 | 84.58 | **0.12** | **4.76** | 0.06 | -0.01 | 0.88 | 43.76 |
| Claude 3.5 Haiku | 71.12 | 50.01 | 35.89 | 80.26 | 0.33 | 7.31 | 0.35 | 0.18 | 1.01 | 42.61 |
| Claude 3 Haiku | 70.63 | 40.84 | 35.64 | 83.59 | 0.42 | 6.24 | 0.07 | -0.22 | 0.85 | 47.15 |
| Claude 3 Opus | 70.15 | 34.17 | 35.59 | 86.97 | 0.62 | 8.26 | 0.07 | -0.52 | 0.93 | 44.36 |
| Gemini 2 Flash | 70.71 | 40.79 | 34.60 | 85.01 | 0.17 | 6.14 | 0.06 | -0.23 | 0.91 | 39.94 |
| GPT 4.1 mini | 69.90 | 37.31 | 33.46 | 86.05 | 0.30 | 6.69 | 0.06 | -0.39 | 0.97 | 38.61 |
| Nova Pro | 68.92 | 31.64 | 32.95 | 87.37 | 0.52 | 7.92 | 0.07 | -0.55 | 1.01 | 39.27 |
| Claude 3.5 Sonnet | 67.15 | 34.25 | 32.22 | 87.57 | 0.43 | 8.41 | 0.09 | -0.46 | 1.06 | 38.55 |
| Llama 3.1 405B | 67.18 | 35.06 | 30.17 | 86.24 | 0.33 | 6.90 | 0.45 | -0.39 | 1.02 | 36.30 |
| Nova Lite | 63.75 | 39.88 | 29.11 | 82.51 | 0.52 | 9.12 | 0.45 | -0.51 | 1.17 | 37.39 |
| *Deepseek R1* | 64.49 | 34.89 | 28.17 | 84.62 | 0.94 | 17.44 | **0.03** | -0.57 | 1.13 | 33.52 |
| *GPT-OSS 20B* | 60.73 | 40.90 | 27.35 | 80.90 | 0.77 | 11.05 | 0.10 | -0.53 | 1.45 | 31.80 |
| Mistral Large V2 | 57.16 | 27.23 | 22.83 | **88.20** | 1.33 | 10.89 | 0.12 | -1.10 | 1.47 | 27.27 |
| Qwen Plus | 55.74 | 24.03 | 21.12 | 88.54 | 2.24 | 10.72 | 0.11 | -1.18 | 1.43 | 24.85 |
| Nova Micro | 55.77 | 29.28 | 18.37 | 86.06 | 1.84 | 11.10 | 0.27 | -1.09 | 1.41 | 24.30 |
| Llama 3.2 90B | 55.78 | **23.81** | 18.30 | **89.56** | 1.84 | 11.10 | 0.27 | -1.09 | 1.41 | 24.30 |
| Llama 3.1 70B | 55.59 | 23.92 | 17.94 | **89.56** | 1.81 | 10.06 | 0.10 | -1.12 | 1.48 | 22.12 |
| Non-expert Human | 45.02 | – | 17.93 | 60.62 | 1.46 | 15.32 | 1.46 | -0.6 | 1.44 | 34.12 |
| **Probability Based Retrieval** | | | | | | | | | | |
| Mistral 8B | **46.63** | 32.21 | **14.73** | 81.46 | **11.42** | 19.47 | 1.27 | -1.35 | 1.95 | 21.01 |
| Llama3 8B | 43.64 | 30.06 | 13.82 | 80.30 | 12.09 | 17.85 | 1.09 | -1.59 | 1.88 | **22.00** |
| Bloomz 7B | 41.15 | 57.76 | 11.27 | 66.09 | 20.62 | 29.00 | 1.51 | **-0.87** | **1.71** | 20.09 |
| *DeepSeek R1 Distill 8B* | 40.02 | 45.33 | 8.85 | 72.20 | 13.38 | 21.62 | 1.14 | -1.29 | 1.75 | 20.42 |
| Qwen2.5 14B | 37.58 | **17.27** | 6.30 | 87.84 | 21.01 | 18.02 | **1.06** | -2.24 | 2.26 | 11.93 |
| Phi3 7B | 34.57 | 17.64 | 2.97 | **87.25** | 23.22 | 18.57 | 1.22 | -2.33 | 2.35 | 7.22 |
| *Phi4-mini-reasoning* | 29.69 | 26.73 | 2.12 | 77.98 | 21.62 | **13.90** | 1.59 | -2.37 | 2.39 | 7.35 |

