# OpenReview forum: "SATA-BENCH: Select All That Apply Benchmark for Multiple Choice Questions"
_ICLR.cc/2026/Conference — ICLR 2026 Conference Desk Rejected Submission_

### Official Review · Reviewer_iS7F · 2025-10-21

**Soundness:** 3
**Presentation:** 2
**Contribution:** 3
**Rating:** 4
**Confidence:** 4

**Summary:**

This paper explores multiple-choice question answering in settings where there is more than one correct answer. The authors introduce SATA-BENCH, a select-all-that-apply benchmark for evaluating model performance across domains. Across 32 models and 6 domains, they find that models tend to struggle when tasked with selecting multiple correct answers. Specifically, they find that models tend to (1) select too few options, (2) guess (or, "speculate") rather than abstaining when uncertain, and (3) consistently prefer/avoid certain option positions. To counter these three challenges, they propose "Choice Funnel", an algorithm that combines three relatively simple approaches (token debiasing methods, a none-of-the-above option, and iteratively selecting the highest probability option) to improve performance on SATA-BENCH, demonstrating positive results.

**Strengths:**

* The paper studies an important problem with real-world implications (e.g., medical diagnosis where multiple conditions could be valid options).
* The authors make a thorough, end-to-end contribution, spanning benchmark construction, evaluation and a deep-dive on failure modes, and proposing a candidate algorithm to mitigate some observed shortcomings.
* SATA-BENCH clearly had a lot of time put into producing a high-quality resource, with the authors taking numerous steps to remove too short or trivial samples, reduce contamination, eliminate redundancy, and filter out vague or ambiguous prompts. The approach for balancing difficulty using confusion scores (i.e., similarity between correct and incorrect answer options) is a particularly nice detail.
* The detailed evaluation of the three biases is informative, and in particular the tension between models that tend to select too few options while also being biased towards guessing. This result is a useful finding that can motivate future research.
* The authors also find certain counter-intuitive results, such as changing the symbol used to delimit options does not appear to help with SATA-BENCH (in contrast with previous work finding strong sensitivity to option symbol), and that scaling up the number of few-shot examples is of limited utility.

**Weaknesses:**

* Given the presentation of results in Table 2, it is hard to draw any meaningful conclusions about differences between models. In particular, there is no clear winner, and very little consistency in rank ordering of models between the different metrics. Perhaps the authors consider a clearer visualization of what this table is intended to convey.
* Relatedly, there are an awful lot of metrics deployed, and it's often unclear why multiple are required, given they appear to be different implementations for measuring the same underlying quality. For example, what do we learn by the inclusion of MAP, when we already have FPR? Or, similarly, CtDif and CtDifAbs? It would significantly improve the presentation of the results if the authors were to select and focus on the most appropriate metric, and perhaps include others as ablations in the appendices if necessary.
* Some of the contributions of the Choice Funnel are a little oversold. Describing adding a NOTA option as "abstention handling" (line 74), or taking the highest probability option until hitting a threshold as "adaptive" (line 74), obfuscates the simplicity of the algorithm itself. Similarly, the claim that Choice Funnel represents a reduction in inference cost isn't backed up by the data presented in Table 4, where Choice Funnel is the second worst approach in terms of inference cost.
* The paper is unclear on whether SATA-BENCH comprises 10k samples (as stated in the introduction) or 1.5K evaluation samples (line 166). If it's the former, if only 1.5k samples are used for evaluation, what are the other samples for?
* The text in Figures 1 and 2 is illegibly small without zooming in.

### Minor
* Statistical tests are improperly specified. If a t-test is used, the authors should declare whether it was paired or independent, one- or two-tailed, and report both the t statistic and degrees of freedom. This information is frequently missing throughout each mention of a t-test in the paper.
* Several inline text citations are accidentally in parentheses (e.g. line 150, line 155).
* Several typos in section 4. Line 431 might be missing some words "current best-performed method"; Line 446 is missing an "a" before "t-test"; etc.

**Questions:**

1. What is the k parameter on line 105? It looks like it's set to c, which would appear to lead to k - c = 0 distractors. Could the authors clarify?
2. Could the authors provide further detail on the confusion score implementation in lines 153-158? Given the sentence embeddings from ST5-XXL, then what?
3. How is "hallucination rate" (line 297) defined and calculated?
4. Why is the option set unioned with {NOTA} in the final condition of Algorithm 1?

---

> ### Author Response · Authors · 2025-11-21
> **Results interpretation and statistical test**
>
> Thank you for the thorough and constructive review. We appreciate your recognition of the **real-world importance of SATA evaluation**, **the rigor of our benchmark construction**, **the clarity of our dataset curation pipeline,(( and the motivations behind the Choice Funnel algorithm. Your identification of the three systematic biases, especially the tension between under-selection and speculative guessing, directly captures the core motivation of this work. We also appreciate your observation of counter-intuitive behaviors such as reduced sensitivity to option symbols and limited improvement from few-shot scaling, which further justified the need for an algorithmic diagnostic and mitigation approach. We address each of your comments in detail below.
>
> **Concern 1 Difficulty interpreting Table 2 and lack of consistent rank ordering**
>
> Thank you for the suggestion to clarify what Table 2 is intended to convey. Table 2 was designed to highlight that different metrics capture different failure modes, which naturally leads to variation in model rank ordering. Nevertheless, we agree that the analysis can be made clearer, so we **have added correlation analyses and visualizations** to reveal the underlying trends more explicitly.
>
> First, we observe that FPR and exact match are positively correlated (r = 0.61, p = 8 × 10⁻⁴, DoF = 23, two tailed). Similarly, FPR and JI are positively correlated (r = 0.70, p = 10⁻⁴, DoF = 23, two tailed). This shows that as **models become better at identifying correct answers, they also tend to speculate more**, which raises FPR. This trend is now highlighted in the revised manuscript (l**ines 315 to 318 and Figure 14**).
>
> Second, stronger models show better CtAcc and lower CtDifAbs. JI and CtAcc are highly correlated (r = 0.96, p = 1.1 × 10⁻¹⁴, DoF = 23, two tailed). EM and CtDifAbs are highly negatively correlated (r = -0.93, p = 1.9 × 10⁻¹¹, DoF = 23, two tailed). This reveals that **models which more accurately identify the answer set also provide more accurate estimates of the number of correct answers.** We now present these trends in the updated **Figures 15 and 16 and discuss them in lines 302 to 303**.
>
> Finally, we highlight that GeminI 2.5 Think exhibits the lowest selection and unselection bias among all models. It achieves the lowest scores on SPD and RStd and the second lowest on RSD. This model-level insight is now included in Figure 17 and lines 324 to 326.
>
> Together, these additions make the trends in Table 2 easier to interpret and show clear, consistent relationships among model behaviors across metrics.
>
> **Concern 2 Too many metrics and unclear necessity for some**
>
> Thank you for this excellent suggestion. We agree that some metrics capture highly similar underlying behaviors, and we have analyzed their correlations to determine which ones provide distinct information. In particular, FPR and MAP are strongly negatively correlated (r = -0.88), and CtDifAbs and CtAcc are also strongly negatively correlated (r = -0.94). These overlaps indicate that the additional metrics do not provide substantial new information.
>
> Following your recommendation, **we have moved MAP and CtDifAbs to the appendix** so that the main text focuses only on the most informative and interpretable measures. This change significantly improves the clarity and readability of the results. These updates are reflected in lines 216 to 259 and lines 457 to 480 of the revised manuscript.
>
> **Concern 5 Statistical test reporting details**
>
> Thank you for pointing this out. We agree that the statistical tests should be reported with full detail. In the revised manuscript we now specify, for every t-test, whether it is paired or independent, whether it is one- or two-tailed, and we **report the t statistic, degrees of freedom, and p values.** The full set of updated statistics is summarized below and has been incorporated into the corresponding locations in the paper.
>
> | Appeared Line | # tails | t statistics | degree of Freedom | Paired/Independent | p value |
> |:---:|:---:|:---:| :---:| :---:| :----:|
> | 302 | 1 | -5.82 | 24 | Independent | 1.7 * 10-6 |
> | 317 | 2 | 0.61 | 23 | Paired | 8 * 10-4 |
> | 324 | 1 | 1.75 | 23 | Independent |  0.0467 |
> | 446 | 2 | 4.92 | 4 | Paired |  0.00793 |
>
> These updates ensure that all statistical analyses follow standard reporting guidelines.
>
> **Concern 6 & 7 Citation and Typo Issues**
>
> We **corrected all instances** where inline citations were mistakenly enclosed in an extra set of parentheses. In the revised manuscript, citations appearing within the sentence now follow the Author (Year) format, and only sentence-final citations use the (Author, Year) format.The revised parts are in line 139 - 143 and line 445. Hallucination rate is a typo and we are referring to FPR. We have fixed it in line 315.

---

> ### Author Response · Authors · 2025-11-21
> **Choice Funnel Algorithm and Additional Finetuning Experiment**
>
> **C4 Clarification on dataset size and roles of the 10k samples**
>
> Thank you for pointing out this ambiguity. We clarify in the revision that SATA-BENCH contains approximately 10k total human-validated questions, of which 1.5k are reserved for the evaluation split. The remaining questions include both additional SATA items and single-answer questions that were fully validated by annotators. These additional samples serve two important purposes.
>
> First, they **allow users to construct custom evaluation sets**, for example domain-specific subsets or mixed sets that include both single-answer and multi-answer formats. Second, and more importantly, they **provide a resource for improving model behavior on SATA reasoning**. Based on our analysis, we suspect that count bias and unselection bias are strongly related to the lack of exposure to SATA-style questions in existing instruction-tuning corpora. The larger portion of the dataset therefore enables fine-tuning or continued pre-training to improve SATA performance without harming general capabilities.
>
>
> To demonstrate this, **we fine-tuned several small LLMs (Llama-3.2-1B, Qwen-2.5-0.5B, and Gemma-2-2B) using a mixture of the Tulu-3-SFT dataset (90%) and our proposed SATA training set (10%).** The additional SATA data **significantly improves performance on SATA-BENCH, while general capabilities remain stable.** All models showed significant improvements on SATA-BENCH while retaining stable performance on MMLU and GSM8K. We include all details are in Appendix T lines 2586 to 2674.
>
>
> | Model | Tuning | MMLU-0 | GSM8K | SATA-EM | SATA-JI |
> |:---:|:---:|:---:| :---:| :---:| :---:|
> |Gemma-2 2B| Tulu-3 (baseline) | 31.1% |30.5% | 0.8%| 15.7% |
> || Tulu-3 + SATA | 32.5%| 32.0%| 25.8%| 62.3% |
> |Llama-3.2 1B| Tulu-3 (baseline) | 30.2% | 23.0% | 0.9%|26.1% |
> || Tulu-3 + SATA | 26.4% | 23.6%| 29.2%| 60.1% |
> |Qwen-2.5 0.5B| Tulu-3 (baseline) | 36.9%| 29.4%| 4.4%| 12.5% |
> || Tulu-3 + SATA | 31.5%| 26.2%| 23.7%| 57% |
>
>
> **C3 Concern that Choice Funnel contributions are oversold**
>
> Thank you for raising this concern. We agree that it is important to present the contributions of Choice Funnel in a balanced way.
>
> Regarding the baselines, the two methods based on first-token selection achieve very low exact match accuracy (average EM of 9.7% for First Token and 5.2% for First Token Debiasing). These approaches are fundamentally unsuitable for SATA tasks because they systematically under-select, which explains their low inference cost but makes them inappropriate comparison points for multi-answer reasoning. We included them mainly to illustrate why methods that work in single-answer MCQ settings break down in SATA scenarios.
>
> This leaves the yes/no baseline as the only viable method for comparison in both accuracy and cost. Our statement that Choice Funnel reduces model forward passes by 64.5% was explicitly scoped to this comparison, as noted in the paper. When evaluated against yes/no, **Choice Funnel provides a clear reduction in inference cost while offering significantly higher exact match accuracy.** We now clarify this scope to avoid implying a broader cost advantage.
>
> Compared to fintuning, Choice Funnel is intended as a easy to implement, practical middle-ground method that improves exact match and joint reasoning while remaining more efficient than yes/no. Following your advice, we have **created a limitation section** where we highlight the limitation of Choice Funnel in line 569 - 571 and line 578 - 581.
>
> **Question 4 Why is the option set unioned with {NOTA} in the final condition of Algorithm 1?**
>
> Thank you for the question. The NOTA option is introduced after the model makes its first selection in order to provide an explicit mechanism for abstention. Without NOTA, the model is forced to either keep selecting options or stop based solely on a probability threshold, which can lead to speculative selections and higher false positives. By adding NOTA to the option set, the model is given a clear way to indicate that it has identified all options it considers correct and is uncertain about the remaining ones. This design reduces unnecessary speculation and improves controlled stopping behavior. We have clarified this in the revised manuscript.

---

> ### Author Response · Authors · 2025-11-21
> **Dataset Construction**
>
> **Question 1 What is the k parameter on line 105? It looks like it's set to c, which would appear to lead to k - c = 0 distractors. Could the authors clarify?**
>
> Thank you for raising this clarification question. In our formulation, k represents the total number of answer options generated for each question. It is not tied to the specific number of correct answers in that question. Instead, k is computed from the dataset-level average number of correct answers, multiplied by our chosen option-to-answer ratio (typically 2-3). This approach ensures that the number of answer choices remains consistent across questions, regardless of the individual question’s answer count.
>
> Because k is defined globally and c (the number of correct answers) is defined per question, it is indeed possible for a small number of questions to satisfy c = k, which leads to k − c = 0 distractors. In this case, all options are correct. This scenario occurred only once in the dataset and does not violate the SATA structure, since SATA questions may legitimately include cases where every presented option is correct.
>
> **Question 2 Could the authors provide further detail on the confusion score implementation in lines 153-158? Given the sentence embeddings from ST5-XXL, then what?**
>
> For each question, we start with n correct options and m distractor options. We encode every option using ST5-XXL, producing a sentence embedding for each one. We then compute cosine similarity between each correct option and each distractor option, resulting in an n by m similarity matrix. The confusion score for that question is defined as the average cosine similarity across all n × m pairs. This produces a single scalar that reflects how semantically close the distractors are to the correct options, and therefore how difficult the SATA question is likely to be.
>
> We have added this clarification to Section 2.2 in the revised manuscript.

---

### Official Review · Reviewer_Evq3 · 2025-10-26

**Soundness:** 2
**Presentation:** 3
**Contribution:** 3
**Rating:** 6
**Confidence:** 3

**Summary:**

This paper introduces SATA-BENCH, a new LLM benchmark for evaluating MCQ problems that can better reflect real-world applications of  multiple choice tasks.

This benchmark covers 10000 human validation questions across six domains, measuring not only correctness but also three systemic biases observed in LLM: speculation bias, unselection bias and count bias.

This paper further proposed a decoding method that combines a selection funnel with token debiasing, adaptive thresholding, and abstention handling. Experiments on 32 LLMs showed that compared to strong baselines, Choice Funnel improved accurate matching by 29% and reduced inference costs by 64%.

**Strengths:**

The benchmark is novel in its approach to evaluating LLMs in SATA tasks. The analysis and classification of bias (speculation bias, unselection bias and count bias)  in the execution of SATA tasks by LLM are clear and well-founded.

The process of constructing the dataset is detailed, covering data filtering, readability scoring, and multiple rounds of manual validation. The experimental design covers a wide range, including 18 closed source models and 14 open source models, all of which prove that this work is solid.

The writing is relatively clear and accompanied by very detailed and clear appendices.

The design of the core method Choice Funnel is effective, improving accuracy and inference efficiency. This benchmark has completed the evaluation task of MCQ, providing a new standard and approach for interpretable evaluation of LLM multi answer reasoning, and has a certain academic influence.

**Weaknesses:**

1. The Choice Funnel method relies on token probability and is therefore only applicable to open-source models with accessible probability distributions.
2. The method is similar to traditional greedy selection and early stopping mechanisms, with high similarity to existing selective prediction or multi label output strategies such as probability thresholding and top-k selection.
3. There may be significant differences in the threshold parameter τ between different models and tasks.
4. This paper claims a 64% reduction in inference costs (Table 4), but compared to inefficient yes/no methods that require generating independent responses for each option. Is the cost significantly reduced compared to standard generative reasoning?

**Questions:**

1. This paper uses metrics such as JI, FPR, EM and Precision to evaluate the performance of the model, but I am curious whether selecting multiple or fewer options incorrectly in certain SATA tasks in specific fields may result in different costs?
2. Would using more diverse languages to evaluate SATA tasks be more comprehensive?
3. In the task settings of the "NON-EXPERT HUMAN BENCHMARK" mentioned in the paper, it is stated that "Each question was presented with the original answer options plus decoys (e.g. ABCD→ABCDEFGHIJK) to identify inattentive workers." Is the setting of distracting options based on any principle? Is the number reasonable? Based solely on the number of distracting options in this example, I wonder if it will affect the validity of the response data?

---

> ### Author Response · Authors · 2025-11-21
>
> Thank you for your thoughtful and constructive review. We appreciate your recognition of **the novelty of our benchmark design, the rigor of our dataset curation, the breadth of our evaluation, and the effectiveness of the Choice Funnel algorithm.** Your comments on the overall writing clarity and organization are also encouraging. Below, we address your concerns in detail.
>
>
> **C1. Similarity to greedy selection, early stopping, or thresholding**
>
> We agree that Choice Funnel incorporates concepts that are broadly related to selection and stopping strategies, but SATA tasks introduce challenges that make standard approaches insufficient.
>
> Traditional probability thresholding performs poorly in SATA settings for three reasons. First, there is no principled way to set thresholds for LLMs, and static cutoffs lead to extremely low exact match accuracy. For example, the First Token baseline achieves 9.7% EM compared with Choice Funnel’s 18-29%. Second, thresholding does not correct systematic positional bias, which we observe to be substantial in SATA-BENCH (SPD up to 23.22 in Table 2). Third, thresholding cannot incorporate abstention when the model is uncertain, which results in high false positive rates for simple baselines (often above 20%).
>
> **Choice Funnel integrates three components that work together:** token debiasing to mitigate positional effects, adaptive thresholding that responds to model-specific confidence scores, and an explicit NOTA mechanism that allows abstention rather than speculation. To our knowledge, prior LLM work on multi-label or multi-answer prediction does not combine these techniques within a single decoding framework for SATA-style reasoning. We **added a clarification in Section 4(line 360-365)** to better explain how these components interact and why they are necessary for this task.
>
> **C1.2 Threshold parameter differences across models**
>
> We discuss this in Appendix L. We initially tuned the threshold using a small sample of 100 questions and observed that this value generalizes well across models. The reviewer is correct that further task-specific tuning could improve performance. Our goal is to show that Choice Funnel works reliably across different open-source models without requiring careful calibration. **The revision (line 2287-2291) highlights this point more clearly.**
>
> **C2. Comparison of inference cost to generative reasoning**
> We focus on probability-based decoding because smaller open-source models struggle with formatted response generation. Even with careful prompt engineering, the parsing success rate falls below 40%, making standard generative reasoning unreliable for SATA evaluation.
>
> Regarding inference cost, Choice Funnel produces only one token per selected option. In contrast, standard generative reasoning produces full natural language outputs. With chain-of-thought prompting, responses often exceed 100 to 300 tokens. **We clarified this in the revision. (line 209 - 210, See Algorithm 1 Comment 3)**
>
> **C3. Interpretation of JI, FPR, EM, Precision under different domains**
> Thank you for this question. The cost of selecting too many or too few answers depends heavily on application context:
>
> Medical diagnosis: Selecting too many conditions can lead to unnecessary tests or harmful drug interactions. Here, low FPR and high precision are essential.
>
> Ethical or policy violation detection: It is often safer to flag more possible violations for human review, so higher recall or slightly higher FPR is acceptable.
>
> Content moderation: Missing a violation is more costly than over-selecting; EM and recall dominate.
>
> **Q1. Use of more diverse languages**
>
> SATA-BENCH currently focuses on English. Evaluating SATA reasoning in multilingual settings is a valuable direction for future work and we have **added this to the Limitations(line 581) section** of the revised manuscript.
>
> **Q2. Human benchmark decoy design**
> When collecting non-expert human annotations, we added between one and eight decoy options per question. This approach follows established practices in instructional manipulation checks and attention validation work [1,2]. Prior studies include fictitious laws or candidates to detect inattentive respondents. In our implementation, decoy options were clearly invalid or blank so that attentive annotators would never select them. The number of decoys was chosen solely to detect inattentive responses, not to affect the difficulty of the question. We clarified this detail in Appendix B.
>
> [1] Oppenheimer, D. M., Meyvis, T., & Davidenko, N. (2009). Instructional manipulation checks: Detecting satisficing to increase statistical power. Journal of experimental social psychology, 45(4), 867-872.
> [2] Berinsky, A. J., Margolis, M. F., & Sances, M. W. (2014). Separating the shirkers from the workers? Making sure respondents pay attention on self‐administered surveys. American journal of political science, 58(3), 739-753.

---

### Official Review · Reviewer_uLiS · 2025-11-02

**Soundness:** 4
**Presentation:** 4
**Contribution:** 2
**Rating:** 6
**Confidence:** 5

**Summary:**

This paper make three primary contributions: first, it introduces SATA-BENCH, a "select-all-that-apply" benchmark that is harvested from previous benchmarks. Second, it contributes the "ChoiceFunnel" algorithm, an algorithm that iteratively selects and eliminates options until a stopping criterion is met.  A reasonably extensive empirical evaluation of the benchmark shows a wide performance spread across models; as a third contribution, the benchmark illuminates several biases in LLMs that negatively impact performance.

**Strengths:**

* The paper is well-written and easy to understand. It is easy to see the motivation for the work, and the benchmark itself seems thoughtfully constructed.

* The authors have considered a variety of evaluative methods. It feels comprehensive.

* The authors test against a number of natural baselines.

* The benchmark has two qualities that I think are important for a benchmark: (1) there is room for improvement (but it's not too hard), and (2) it provides discriminative insight across models.  The spread for this benchmark seems good, with the best performers topping out at 75% accuracy, and the lowest performers scoring around 30%.

* The insights about cardinality bias seem fresh.

* The ChoiceFunnel algorithm seems reasonable, although a bit obvious.

**Weaknesses:**

While this is a well-done evaluation of a simple idea, I have two main concerns. The first is just the significance of the work. While I appreciate the careful attention to detail and the reasonably well done empirical work, it seems like a fairly niche question.  Frankly, I don't think the authors do a good job of building the case for the need for this benchmark.

The second is that this paper seems to be missing an important set of related work, where this problem is termed "multi-label classification".  A quick google search turned up these two papers:

Large Language Models Do Multi-Label Classification Differently
Marcus Ma*, Georgios Chochlakis*, Niyantha Maruthu Pandiyan, Jesse Thomason, Shrikanth Narayanan

Validate Your Authority: Benchmarking LLMs on Multi-Label Precedent Treatment Classification
M. Mikail Demir and M. Abdullah Canbaz

which both cite many papers that don't seem to appear in the present paper.  I worry therefore, that this paper's claim to novelty isn't a strong as it appears.

--- Additional concerns

I am also worried about the performance of Choice Funnel in the case where the number of correct options is 1.  That is: in the "real world", it may not be guaranteed that there is more than one correct answer, and yet as far as I can tell, all of the benchmark questions have at least 2 correct choices.  The ideal algorithm would tolerate both single-answer and multi-answer questions.

I wish the text were more balanced on discussing drawbacks of the proposed ChoiceFunnel algorithm. For example, in the "Key Observations" section, the authors extol the positive effects that their algorithm has, without a balanced discussion of negative aspects of the algorithm.  (In other conferences, perhaps these would be included in a "Limitations" section).  For example, in my reading of Table 4, the CF algorithm often improves EM accuracy (good) but often reduces precision (bad).  It's not uniformly positive, but the authors don't transparently acknowledge that.

(Along those lines, it feels fair to point out that the yes/no algorithm has the advantage that it can assign independent confidences to each answer, while the CF algorithm cannot).

I'm also not sure how I feel about this sentence: "Existing SATA datasets, such as (Lewis et al., 2004; Kowsari et al., 2017; Aly et al., 2019; Katakis et al., 2008; Charte et al., 2015), often include over 30 labels per question, making exhaustive prediction impractical for LLMs."  I think it's unfair to dismiss this because of computational limitations -- in reality, all of these benchmarks are *tiny* and run very quickly, and I think the point of benchmarks is to assess capabilities and limitations of models, and rarely truly compute bound.  I think it's a bit presumptuous to judge whether the computational disadvantages of an algorithm outweigh its other advantages; that decision should be to the person who actually cares about the tradeoff.

**Questions:**

Can you more strongly distinguish this work from previous work on multi-label classification?

How does choice funnel perform when there is only one right answer?

---

> ### Author Response · Authors · 2025-11-21
> **Manuscripts Clarifications.**
>
> Thank you for your thorough and constructive review. We appreciate your recognition of **the benchmark construction, the breadth of evaluation methods, the insights about cardinality bias, and the design of the Choice Funnel algorithm.** Thank you for highlighting the **two most important qualities of our benchmark: room for improvement, not too hard and providing discriminative insight.** Your comments helped refine the clarity and positioning of our contributions. Below, we address your concerns in detail.
>
> **Concern 1 Significance and scope of the work**
>
> Thank you for raising this point. SATA reasoning is central to many deployed LLM applications, not a niche capability. For example, (1) legal/compliance systems must identify all applicable issues in a contract (e.g., data privacy, IP, indemnification); (2) clinical decision-support tools must surface all plausible conditions given overlapping symptoms; and (3) content-moderation systems must flag all violated policies in a single post. Missing even one correct category leads to concrete safety, legal, or medical risks.
>
> Across all these domains, accurate multi-answer reasoning requires models to (1) identify all correct options, (2) avoid selecting irrelevant ones, and (3) correctly infer the number of applicable answers. Existing single-answer benchmarks (e.g., MMLU, ARC, GPQA) do not evaluate these capabilities, and our results show that models achieving >85% accuracy on those benchmarks still fail to recover the complete answer set in 58% of SATA questions. This gap indicates that **single-answer evaluations significantly overestimate LLM reliability in real multi-label decision-making settings.**
>
> **Other reviewers explicitly highlighted the real-world importance** of this problem and viewed the benchmark as meaningful. To avoid any misunderstanding, we plan to revise the introduction to make these practical motivations more prominent and better grounded.
>
> **Concern 2 & Question 1 Related work on multi-label classification**
>
> Thank you for this insightful observation. We agree that multi-label classification (MLC) provides a related perspective, and **we updated Section 5(line 485-502) to include the two suggested LLM–MLC papers and their cited literature.** At the same time, SATA-BENCH addresses a substantially different problem setting, and we clarify this distinction below.
>
> **Task formulation:**
> Traditional MLC assumes a fixed, global label set and predicts each label independently using sigmoid outputs and thresholding. In contrast, SATA questions require reasoning over a dynamic, natural-language option set where options are semantically interdependent, phrased as full sentences, and intentionally constructed to be confusable. This shifts the problem from *label-wise scoring to contextual multi-step reasoning over candidate answers*, which classical MLC methods are not designed to capture.
>
> **LLM behavior under MLC vs. SATA:**
> Recent LLM–MLC work studies how LLMs can be adapted into the MLC pipeline by converting labels into binary decisions. Our work instead analyzes LLMs’ native reasoning behavior in SATA settings, revealing count, speculation, and unselection biases that do not surface when labels are treated independently.
>
> **Benchmark scope:**
> Existing MLC datasets are domain-specific (e.g., legal tagging, emotions) and not formatted as select-all-that-apply questions. SATA-BENCH is the first multi-domain benchmark built entirely from genuine SATA questions with curated distractors, enabling evaluation of multi-answer reasoning rather than multi-label classification.
>
> **Concern 5 Comparison with multi-label datasets and computational claims**
> Thank you for pointing this out. We agree that the original phrasing regarding computational impracticality was not well calibrated, and we removed this claim in the revision. Our intention was not to suggest that multi-label datasets are unusable due to compute constraints.
>
> The point we intended to make is that traditional multi-label datasets differ from SATA questions in their task formulation. MLC datasets assume a large global label vocabulary, evaluated independently using sigmoid outputs and thresholding. This setting does not involve contextual reasoning over a shared set of candidate options.
>
> SATA questions present a small, question-specific set of natural-language options whose correctness must be evaluated jointly. This structure produces reasoning behaviors that do not appear in MLC tasks, including count bias, unselection bias, and speculation. These phenomena are at the core of our diagnostic analysis. We updated Section 5(line 485-502) accordingly to better reflect above perspective.

---

> ### Author Response · Authors · 2025-11-21
> **Additional Choice Funnel Experiments and Clarification**
>
> **Concern 3 & Question 2 Choice Funnel behavior when only one answer is correct**
>
> Thank you for raising this point. SATA-BENCH intentionally focuses on multi-answer questions, because these are the cases where standard MCQ evaluation fails and where LLMs exhibit the three systematic biases we study (count, speculation, unselection). All benchmark questions therefore have ≥2 correct answers by design. That said, Choice Funnel naturally handles single-answer cases: when only one correct option exists, CF simply selects the top-probability choice in the first iteration and terminates, behaving identically to standard greedy decoding. To verify this empirically, our full 10k dataset includes ~750 single-answer questions of similar difficulty and from same domains, and we now report results in Appendix M.5.
>
> **Summary of our new empirical findings (Appendix M.5):**
> Across Llama3-8B, Mistral-8B, and Qwen2.5-14B, CF has high EM and JI. **CF shows no pathological behavior on single-answer tasks and remains consistent with its expected top-1 behavior.**
>
> We clarified this in the revision and included the single-answer comparison in Appendix M.5.
>
> | Model | Dataset | EM↑ | JI↑ | Precision↑ |
> |-------|---------|-----|-----|------------|
> | Llama3-8B | Original (2+ answers) | 19.88 | 50.36 | 78.69 |
> | Llama3-8B | Single answer | **55.04** | **73.18** | 70.05 |
> | Mistral-8B | Original (2+ answers) | 20.24 | 52.56 | 86.03 |
> | Mistral-8B | Single answer | **60.27** | **76.82** | 73.31 |
> | Qwen2.5-14B | Original (2+ answers) | 27.82 | 61.12 | 85.69 |
> | Qwen2.5-14B | Single answer | **63.33** | **79.56** | 76.17 |
>
>
> **Concern 4 Need for a more balanced discussion of Choice Funnel’s drawbacks**
>
> Thank you for highlighting this important point. We agree that the paper should be more explicit about the limitations and tradeoffs of ChoiceFunnel. We **added a dedicated Limitations subsection **clarifying that CF, while improving EM and JI consistently, can reduce precision relative to certain baselines.
>
> As the reviewer correctly observed, the “First Token” baseline often achieves higher precision (77–87%), but this behavior stems from systematic under-selection the model defaults to selecting only one option almost always. This leads to extremely low exact match (e.g., 2.12% EM on Phi4-mini) despite high precision. In contrast, CF intentionally increases recall to recover full answer sets in SATA tasks, which naturally leads to a precision–recall tradeoff. The goal of CF is therefore not to maximize precision alone, but to correct the count and unselection biases inherent in multi-answer reasoning.
>
> We made these tradeoffs more explicit, added a new "Limitations" subsection following reviewer's suggestion (line 578 - 582), and clarified in the main text and Table 4 discussion that CF is not uniformly better on all metrics, but is designed to improve completeness (EM/JI) while accepting the known precision tradeoff.

---

### Official Review · Reviewer_1sgr · 2025-11-02

**Soundness:** 2
**Presentation:** 2
**Contribution:** 1
**Rating:** 2
**Confidence:** 4

**Summary:**

The authors introduce SATA-BENCH, a new benchmark for evaluating Large Language Models (LLMs) on "Select All That Apply" (SATA) questions, where multiple correct answers are possible. This work addresses a gap in current evaluations, which primarily focus on single-answer tasks. The benchmark includes over 10,000 human-validated questions (but only 1.47K where annotators consensus is reached) across six domains.
Their evaluation of 32 models reveals that even the best-performing models struggle, with the top model achieving only 41.8% exact match accuracy. The paper diagnoses three systematic failure modes: unselection bias (avoiding certain correct choices), speculation bias (including incorrect choices), and count bias (underpredicting the number of correct answers).
To address these issues, the authors propose Choice Funnel, a decoding algorithm for open-source models that combines token debiasing, adaptive thresholding, and abstention handling. This method is shown to improve exact match accuracy by up to 29 percentage points and reduce inference costs by 64% compared to a "yes/no" baseline.

**Strengths:**

1)	The paper addresses a valid or a good gap. Authors point out that most real-world applications (e.g., content moderation, medical diagnosis, legal research) involve identifying multiple valid "answers". Current single-answer benchmarks fail to capture this capability. The creation of a large-scale, human-validated benchmark for this task could be a good resource
2)	The curation of SATA-BENCH is explained clearly, involving a multi-stage process of transformation and filtering based on readability, similarity, confusion score, and human validation . The proposed evaluations of the three biases (unselection, speculation, and count) is also informative.
3)	Papaer proposes choice funnel, a decoding strategy that directly targets the identified biases. The ablation studies are insightful, such as the finding that explicitly providing the correct number of answers dramatically improves performance (20.95 point EM increase), which supports their focus on "count bias."

**Weaknesses:**

•  SATA questions can have multiple answers, but it is debatable whether a ranking could exist among the correct answers. A binary prediction of whether a candidate option is correct or not doesn't capture this. This raises a slight concern about the positioning of the multiple-answers task within SATA.


•  Further, one of the other important concerns regarding the curation of this dataset is that very little information is given on the annotators. What was their background? Did they have expertise in the domains considered within SATA?


• Following up on shortcomings within the annotation process, disagreement issues between annotators are not discussed. It would be important to know what happened in cases where two out of three annotators agreed and the third disagreed. Lastly, in the annotation process, key details (e.g., distractor sampling, domain balancing, and overlap with training data) are not fully explained.

• The non-expert human benchmark is a valuable inclusion, but the results are quite low: 17.9% Exact Match (EM) and 45.0% Jaccard Index (JI). This performance is not dramatically better than that of many of the open-source models. This low score could suggest that many questions in the benchmark are inherently ambiguous or extremely difficult even for humans, which complicates the paper's narrative that the models have "substantial limitations."

• System prompts are not discussed in detail, and there are intuitively several possibilities. For example, instead of “The system prompt specifies that each question has at least two correct answers,” the system prompt could be more generic, instructing LLMs to pick any candidate answer that is correct. A similar gap exists for the CoT prompt.

**Questions:**

Please see the questions in weakness points.

---

> ### Author Response · Authors · 2025-11-21
> **Additional Human Evaluation and Evaluation details.**
>
> Thank you for your detailed review. We appreciate your recognition that **SATA-BENCH fills an evaluation gap that single-answer benchmarks cannot capture**, as well as your **positive remarks on our multi-stage curation pipeline, our analysis of three systematic biases, and the Choice Funnel algorithm.** Below we address your specific concerns.
>
> **Concern 1: Concern about ranking among multiple correct answers in SATA tasks**
>
> Answer: Thank you for this observation. SATA tasks in real applications involve selection, not ranking, and our benchmark is intentionally designed around this structure. In many real-world scenarios, correct answers are equally valid and not meaningfully ordered, for example:
>
> - *Content moderation*: A post may simultaneously violate hate speech and misinformation policies. The order is irrelevant; completeness matters.
> - *Legal document analysis*: A contract may fall under multiple statutes or legal categories; all must be identified.
> - *Medical triage*: Symptoms may be consistent with several plausible conditions; all candidates must be surfaced.
>
> In these settings, ranking correct answers is neither required nor well-defined, and selection is the appropriate formulation. This is what differentiates SATA from ranked retrieval, preference learning, or multi-label ranking tasks.
>
> To validate that SATA-BENCH correct answers are equivalently valid, we **conducted a additional dedicated human study** using MTurk. Given a question and its ground-truth answer set, annotators were asked whether all correct answers were equally correct. Our Key results are as follows:
> - 95.1% of questions received at least 2 of 3 votes indicating that all correct answers are equivalently valid
> - An additional 3.9% received 1 of 3 votes indicating equivalence.
> - Only 1% of questions raised any concern.
>
> This **confirms that the correct answers in our dataset rarely exhibit meaningful internal ranking.** We have incorporated these findings into lines 171 and 1232–1241 of the revised manuscript.
>
> **Concern 2 Lack of information about annotator background**
>
> Answer: Thank you for being interested in human labeling process. While we cannot share individual annotator identities, we **provide aggregated information about the annotation team we use for answer correctness evaluation**. All annotators hold at least a Bachelor’s degree, and 22% additionally hold Master’s degrees, and all are fluent English speakers. The team has an average of **3.5 years of professional labeling experience** and routinely works on a broad range of tasks, including general question answering, sensitive-content evaluation, and multi-label classification. Importantly, every annotator selected from annotation team **has prior experience working on domain-specific tasks aligned with the six SATA-BENCH domains** (reading comprehension, toxicity, news, biomedical, legal, and events), ensuring familiarity with the linguistic and conceptual nuances required for accurate annotation. The annotation guidelines were prepared by trained technical writers whose primary responsibility is developing annotation Standard Operating Procedures (SOPs); they hold degrees in English language or literature and have over three years of experience drafting labeling instructions. Both the SOP and the simplified annotator instructions used in this project are included in Appendix B.3. The manuscript has been updated in lines 1170–1230 to reflect these details.
>
> **Concern 3 Annotation disagreements, distractor sampling, domain balancing, and contamination checks**
> Answer: Thank you for highlighting these aspects. We now clarify them:
>
> **Disagreements:** Each question is labeled independently by two annotators. If they disagree, a third adjudicator reviews the question, model outputs, and annotator decisions. **Pairwise agreement among the first two annotators: 91.2%.** After adjudication and filtering: 96.5% agreement. Since all disagreements were adjudicated, the true error rate is significantly below 3.49%. These details are now included in Appendix B.4 and lines 1188–1194.
>
> **Distractor sampling:** Distractors were drawn from class labels in the original source datasets to maintain semantic exclusivity and plausibility. This is described in lines 950-964.
>
> **Domain balancing:** We applied confusion-score stratified sampling within each domain to maintain a balanced difficulty distribution across domains (detailed in section 2.2).
>
> **Training-data contamination:** We applied an open-source contamination detection pipeline [5], as **detailed in Appendix B.1** (line 1129 - 1134). For each question, we automatically generated twenty Bing search queries, checked for verbatim matches in publicly available web results, and cross-referenced all hits against Common Crawl indices.This multi-step procedure ensures that SATA-BENCH items do not appear in Common Crawl and are therefore **unlikely to be contaminated.**

---

> ### Author Response · Authors · 2025-11-21
> **Human benchmark and additional experiments**
>
> **Concern 4 Low non-expert human benchmark scores and concerns about ambiguity**
>
> Answer: The low human scores do not reflect dataset ambiguity. We address this in two ways:
>
> **1. Dataset filtering eliminates ambiguous items.**
> In Appendix B.2 (lines 1080–1100), we report that **we removed all questions where fewer than 5 out of 5 annotators judged the question to be clear.** This filtering step was **highlighted as a strength by other reviewers.**
>
>
> **2. Non-experts systematically underperform on SATA tasks.**
> Prior work shows that humans tend to select too few options in SATA formats, even in simple survey settings [1,2,3]. In our benchmark:
> - Questions span law, medicine, and technical domains
> - Crowdworkers lack domain expertise (Appendix E)
> - This mirrors setups in prior benchmarks (e.g., GPQA), where non-expert humans significantly underperform LLMs
>
> For example, GPQA [4] reports 33.9% human accuracy versus 87.7% for LLMs, despite being designed for human evaluation.
>
> Thus, the human difficulty arises from domain expertise gaps, not ambiguity, and does not contradict the paper’s narrative about model limitations.
>
> **Concern 5 System prompts and sensitivity analysis**
>
>
> Thank you for pointing out that alternative prompting strategies may be possible. To ensure consistency with other benchmark, our evaluation code is following the structure in openai codebase simple-eval[6]. To evaluate this, we **conducted an additional ablation comparing.** We change part of our default system prompt “Each question has at least two correct answers.” to “Please pick any candidate answer that is correct.”.
>
> Using four representative models evaluated on 400 SATA questions (original system prompt performance - your suggested system prompt's performance), we found:
> - EM differences: all within ±1.6%
> - Precision and FPR differences: all within ±1%
> - Average difference across all metrics: below 0.2%
>
> This indicates that SATA-BENCH performance is **not sensitive to system prompt phrasing.** We added this study in lines 204 and 2409-2425.
>
>
> | Models | EM Dif | Precision Dif | FPR Dif |
> |:---:|:---:|:---:| :---:|
> |GPT-OSS 20B | 0.2% | 0.9% | -1.1% |
> |GPT-OSS 120B| 1.6% | 0.3% | -0.3% |
> |Claude 3.5 Sonnet| -0.5% | -0.6% | 0.3% |
> |QWen Plus | -0.5% | -0.4% | 0.5% |
> |Average | 0.2% | 0.07% | 0.18% |
>
> [1]Pew Research Center (2019): “When Online Survey Respondents Only Select Some That Apply”
>
> [2]Smyth et al. (2006): “Comparing check-all and forced-choice question formats in web surveys”
>
> [3]Callegaro, M., Murakami, M. H., Tepman, Z., & Henderson, V. (2015). Yes–no answers versus check-all in self-administered modes: A systematic review and analyses. International Journal of Market Research, 57(2), 203-224.
>
> [4] Rein, D., Hou, B. L., Stickland, A. C., Petty, J., Pang, R. Y., Dirani, J., ... & Bowman, S. R. (2024, July). Gpqa: A graduate-level google-proof q&a benchmark. In First Conference on Language Modeling.
>
> [5] Li, Y., Guo, Y., Guerin, F., & Lin, C. (2024, November). An open-source data contamination report for large language models. In Findings of the Association for Computational Linguistics: EMNLP 2024 (pp. 528-541).
>
> [6] Simple eval codebase: https://github.com/openai/simple-evals
>
> [7] Huang, O., Fleisig, E., & Klein, D. (2023). Incorporating worker perspectives into MTurk annotation practices for NLP.

---

### Note · Program_Chairs · 2026-01-17
**Submission Desk Rejected by Program Chairs**

The following references in this submission do not refer to real documents and/or have major errors in bibliographic information:

 Francisco Charte, Antonio J. Rivera, María J. del Jesus, and Francisco Herrera. Stackex: A collection of multi-label datasets from stack exchange forums. Journal of Multiple-Valued Logic and Soft Computing, 25(5):429-448, 2015. URL https://github.com/yourURL/ stackex-dataset